# Extending Path-Dependent NJ-ODEs to Noisy Observations and a Dependent Observation Framework

**William Andersson**  *anwillia@ethz.ch*
*Department of Computer Science*
*ETH Zurich, Switzerland*

**Jakob Heiss**  *jakob.heiss@math.ethz.ch*
*Department of Mathematics*
*ETH Zurich, Switzerland*

**Florian Krach**  *florian.krach@math.ethz.ch*
*Department of Mathematics*
*ETH Zurich, Switzerland*

**Josef Teichmann**  *josef.teichmann@math.ethz.ch*
*Department of Mathematics*
*ETH Zurich, Switzerland*

**Reviewed on OpenReview:** *https://openreview.net/forum?id=0T2OTVCCC1*

## Abstract

The *Path-Dependent Neural Jump Ordinary Differential Equation (PD-NJ-ODE)* (Krach et al., 2022) is a model for predicting continuous-time stochastic processes with irregular and incomplete observations. In particular, the method learns optimal forecasts given irregularly sampled time series of incomplete past observations. So far the process itself and the coordinate-wise observation times were assumed to be independent and observations were assumed to be noiseless. In this work we discuss two extensions to lift these restrictions and provide theoretical guarantees as well as empirical examples for them. In particular, we can lift the assumption of independence by extending the theory to much more realistic settings of conditional independence without any need to change the algorithm. Moreover, we introduce a new loss function, which allows us to deal with noisy observations and explain why the previously used loss function did not lead to a consistent estimator.

## 1 Introduction

While the online prediction[1] of regularly observed or sampled time series is a classical machine learning problem that can be solved with recurrent neural networks (RNNs) as proven e.g. by Schäfer & Zimmermann (2006), the forecasting of continuous-time processes with irregular observation has long been an unsolved problem. The Neural Jump ODE (NJ-ODE) (Herrera et al., 2021) was the first framework with theoretical guarantees to converge to the optimal prediction in this setting. However, it was restricted to Markovian Itô-diffusions with irregular but complete (i.e., all coordinates are observed at the same time) observations. This was heavily generalised with the Path-Dependent NJ-ODE (PD-NJ-ODE) (Krach et al., 2022), where the convergence guarantees hold for very general (non-Markovian) stochastic processes with irregular and incomplete observations. Still, the process itself and the observation framework were assumed to be independent and observations were assumed to be noisefree. In practice both of these assumptions are often unrealistic.

---

[1]With *online* prediction we mean that we use the currently available information to predict until we get new information. As soon as new information becomes available, it is part of the available information and therefore taken into account for subsequent predictions.

E.g., for medical patient data collected at a hospital irregularly over time such as Goldberger et al. (2000), measurements are never noise-free and the decision whether to make a measurement depends on the status of the patient. Therefore, the focus of this work is to lift those two restrictions. A detailed outline is given below.

## 1.1 Related Work

GRU-ODE-Bayes (Brouwer et al., 2019) and the latent ODE Rubanova et al. (2019) both use a model very similar to the NJ-ODE model, however, with different training frameworks. While the latent ODE can only be used for offline forecasting, GRU-ODE-Bayes is applicable to online forecasting as the NJ-ODE. However, in comparison to NJ-ODE, no theoretical guarantees exist for GRU-ODE-Bayes.

Neural controlled differential equations (NCDE) (Kidger et al., 2020; Morrill et al., 2022) and neural rough differential equations (Morrill et al., 2021) also use similar model frameworks, but their primary objective are labelling problems, i.e., the prediction a classification or regression label for the input of an irregularly sampled time series. For example, based on health parameters of a patient these models try to decide whether the patient will develop a certain disease in the future.

As explained in Krach et al. (2022), PD-NJ-ODEs can be used for stochastic filtering. Another well known model class for this problem are particle filters, also called sequential Monte Carlo methods (Maddison et al., 2017; Le et al., 2017; Corenflos et al., 2021; Lai et al., 2022). Particle filtering methods are applied in the context of state-space models (SSM), which are characterized by a discrete latent Markov process $(X_t)_{t=1}^T$ and a discrete observation process $(Z_t)_{t=1}^T$ defined on a fixed time-grid. Particle filters are used to approximate e.g. the conditional distribution of $X_t$ given the observations $(Z_s)_{1 \le s \le t}$, or the joint distribution of $(X_s, Z_s)_{1 \le s \le t}$, for any $t \ge 1$, using weighted sequential Monte Carlo samples. In our work, we allow for a much more general setting than the SSM. In particular, we allow for a continuous-time (instead of discrete-time), non-Markovian stochastic process. Since our setting allows for jumps of the process, this also includes the SSM case of a discrete-time Markov process. Moreover, in our setting, the underlying process can be observed at random, irregularly sampled, discrete observation times and the framework allows for incomplete observations, where some coordinates might not be observed. The primary goal of the PD-NJ-ODE method is to make optimal forecasts for $X_t$ given all observations of $X$ prior to time $t$. As a special case (since the framework can deal with incomplete observations), this includes the filtering problem, however, in a more general setting, allowing for example to predict $X_t$ while only having (discrete) observations of $Z$ at randomly sampled observation times until time $s < t$.

For further related work we refer the interested reader to the respective sections in Herrera et al. (2021) and Krach et al. (2022).

## 1.2 Outline of the Work

We introduce two extensions of the PD-NJ-ODE (Krach et al., 2022) that can be used separately or jointly. To highlight the needed adjustments for each of the extensions, we first recall the setup, model and results from Krach et al. (2022) (Section 2) and then introduce the respective changes in the assumptions and proofs for noisy observations (Section 3) and dependence between the underlying process and the observation framework (Section 4) separately. We focus on re-proving the main results (Krach et al., 2022, Theorem 4.1 and Theorem 4.4) in the new settings, by giving the arguments which need to be adjusted while skipping those which remain unchanged. In Appendix C we give the full proof for the most general result with both extensions, making the paper self-contained. We remark here that also the results for the conditional variance and for stochastic filtering (Krach et al., 2022, Section 5 and 6) follow in these extended settings similarly as the main results. Due to the similarity, we do not elaborate on this but leave the details to the interested reader. In Section 5 we discuss the practical implications of our main convergence result. Finally, in Section 6 we show empirically that the PD-NJ-ODE performs well in these generalised settings.

## 2 Recall: the PD-NJ-ODE

In this section we recall the PD-NJ-ODE framework together with the main result from Krach et al. (2022). The PD-NJ-ODE is a prediction model that can be used to learn the $L^2$-optimal prediction of a stochastic process (continuously in time), given its discrete, irregular and incomplete observations in the past. This means that the PD-NJ-ODE learns to compute the conditional expectation, which is the $L^2$-optimal prediction. Importantly, only data samples are needed to train the model. In particular, no knowledge about the dynamics of the underlying process is needed. We first given an intuitive example for the application of this model, which will be reused throughout the paper, and then discuss its technical details.

### 2.1 Intuitive Example Application

Suppose we have a dataset of $N$ patients $1 \leq j \leq N$. Any patient $j$ has $d_X$ medical values $\left( X_{t,k}^{(j)} \right)_{1 \leq k \leq d_X}$ (such as body temperature or blood pressure) at any time $t \in [0,T]$. However, we only have (noisy) measurements of some of these coordinates at some irregular times $t_i^{(j)}$ (e.g., we measure the body temperature on one day and the blood pressure on another day and in-between we do not measure anything). Our training dataset consists of all these (noisy) measurements including their time-stamps. Based on this training dataset we train a PD-NJ-ODE which then allows us to make online forecasts for new patients based on their noisy incomplete irregularly observed measurements. For example the Physionet dataset (Goldberger et al., 2000) is exactly such a dataset, but of course our method can also be applied in many other situations.

### 2.2 Technical Background and Mathematical Notation

We start by recalling the most relevant parts of the problem setting of the PD-NJ-ODE. For more details, please refer to Krach et al. (2022).

For $d_X \in \mathbb{N}$ and $T > 0$ we consider a filtered probability space $(\Omega, \mathcal{F}, \mathbb{F} := (\mathcal{F}_t)_{0 \leq t \leq T}, \mathbb{P})$ with an adapted càdlàg stochastic process $X := (X_t)_{t \in [0,T]}$ taking values in $\mathbb{R}^{d_X}$. The main goal in this paper is the predict $X$ optimally for future times, based on discrete observations of it in the past. We denote its running maximum process by $X^\star$ (i.e., $X_t^\star := \sup_{s \in [0,t]} |X_s|_1$) and the random set of its jump times by $\mathcal{J}$. The *random observation framework* is defined independently of $X$ on another filtered probability space $(\tilde{\Omega}, \tilde{\mathcal{F}}, \tilde{\mathbb{F}} := (\tilde{\mathcal{F}}_t)_{0 \leq t \leq T}, \tilde{\mathbb{P}})$ by

- $n : \tilde{\Omega} \to \mathbb{N}_{\geq 0}$, an $\tilde{\mathcal{F}}$-measurable random variable, the random number of observations,

- $K := \sup \left\{ k \in \mathbb{N} \mid \tilde{\mathbb{P}}(n \geq k) > 0 \right\} \in \mathbb{N} \cup \{\infty\}$, the maximal value of $n$,

- $t_i : \tilde{\Omega} \to [0,T] \cup \{\infty\}$ for $0 \leq i \leq K$, sorted stopping times, which are the random observation times, with $t_i(\tilde{\omega}) := \infty$ if $n(\tilde{\omega}) < i$ and $t_i < t_{i+1}$ for all $0 \leq i < n$,

- $\tau : [0,T] \times \tilde{\Omega} \to [0,T]$, $(t, \tilde{\omega}) \mapsto \tau(t, \tilde{\omega}) := \max\{t_i(\tilde{\omega}) | 0 \leq i \leq n(\tilde{\omega}), t_i(\tilde{\omega}) \leq t\}$, the last observation time before a certain time $t$, and

- $M = (M_k)_{0 \leq k \leq K}$, the observation mask, which is a sequence of random variables on $(\tilde{\Omega}, \tilde{\mathcal{F}}, \tilde{\mathbb{P}})$ taking values in $\{0,1\}^{d_X}$ such that $M_k$ is $\tilde{\mathcal{F}}_{t_k}$-measurable. The $j$-th coordinate of the $k$-th element of the sequence $M$, i.e., $M_{k,j}$, signals whether $X_{t_k,j}$, denoting the $j$-th coordinate of the stochastic process at observation time $t_k$ is observed. By abuse of notation we also write $M_{t_k} := M_k$.

In the following we consider the filtered product probability space $(\Omega \times \tilde{\Omega}, \mathcal{F} \otimes \tilde{\mathcal{F}}, \mathbb{F} \otimes \tilde{\mathbb{F}}, \mathbb{P} \times \tilde{\mathbb{P}})$ and the filtration of the currently available information $\mathbb{A} := (\mathcal{A}_t)_{t \in [0,T]}$ defined by

$$\mathcal{A}_t := \boldsymbol{\sigma} \left( X_{t_i,j}, t_i, M_{t_i} | t_i \leq t, j \in \{1 \leq l \leq d_X | M_{t_i,l} = 1\} \right),$$

where $\boldsymbol{\sigma}(\cdot)$ denotes the generated $\sigma$-algebra. We note that $\mathcal{A}_t = \mathcal{A}_{\tau(t)}$ for all $t \in [0,T]$. The conditional expectation process of $X$, which is its $L^2$-optimal prediction (Krach et al., 2022, Proposition 2.5) and therefore the process we would like to compute, is defined as $\hat{X} = (\hat{X}_t)_{0 \leq t \leq T}$, with $\hat{X}_t := \mathbb{E}_{\mathbb{P} \times \tilde{\mathbb{P}}}[X_t | \mathcal{A}_t]$. Even though

this solution to our prediction problem is abstractly well understood, it is not clear how to actually compute it, especially if one only observes data samples without knowing the distribution of $X$. The PD-NJ-ODE is designed to do exactly this. To compute the conditional expectation, it uses the powerful framework of signatures to convert all information available from the past observations into a tractable feature-vector. This is needed, since the underlying process is not assumed to be Markovian. Since the signature is defined for (continuous) paths, the first step is to construct a continuous interpolation of the discrete observations, the *interpolated observation process*, in such a way that it carries the same information as the discrete observations. In particular, for any $0 \le t \le T$ the $j$-th coordinate of the interpolated observation process $\tilde{X}^{\le t} \in \mathbb{R}^{2d_X+1}$ at time $0 \le s \le T$ is defined by

$$\tilde{X}^{\le t}_{s,j} := \begin{cases} X_{t_{a(s,t)},j} \frac{t_{b(s,t)}-s}{t_{b(s,t)}-t_{b(s,t)-1}} + X_{t_{b(s,t)},j} \frac{s-t_{b(s,t)-1}}{t_{b(s,t)}-t_{b(s,t)-1}}, & \text{if } t_{b(s,t)-1} < s \le t_{b(s,t)} \text{ and} \\ & 1 \le j \le d_X, \\ X_{t_{a(s,t)},j}, & \text{if } s \le t_{b(s,t)-1} \text{ and } 1 \le j \le d_X, \\ \tilde{u}_{t_{a(s,t)},j-d_X} + \frac{s-t_{b(s,t)-1}}{t_{b(s,t)}-t_{b(s,t)-1}}, & \text{if } t_{b(s,t)-1} < s \le t_{b(s,t)} \text{ and} \\ & d_X < j \le 2d_X, \\ \tilde{u}_{t_{a(s,t)},j-d_X}, & \text{if } s \le t_{b(s,t)-1} \text{ and } d_X < j \le 2d_X, \\ s, & \text{if } j = 2d_X+1, \end{cases}$$

where $\tilde{u}_{t,j} := \sum_{k=0}^{K} M_{k,j} \mathbb{1}_{t_k \le t}$ is the jump process that counts the coordinate-wise observations and

$$a(s,t) := a(s,t,j) := \max\{0 \le a \le n | t_a \le \min(s,t), M_{t_a,j} = 1\},$$
$$b(s,t) := b(s,t,j) := \inf\{1 \le b \le n | s \le t_b \le t, M_{t_b,j} = 1\},$$

with $t_\infty := \infty$. Simply put, $\tilde{X}^{\le t}$ is a continuous version (without information leakage and with time-consistency) of the rectilinear interpolation of the observations of $X$ and of $\tilde{u}$. The paths of $\tilde{X}^{\le t}$ belong to $BV^c([0,T])$, the set of continuous $\mathbb{R}^{d_X}$-valued paths of bounded variation on $[0,T]$. Most importantly, the way it is defined ensures that no information about the next observation is leaked (through the forward-looking interpolation) until after the last observation time prior to it and that $\tilde{X}^{\le t}$ is $\mathcal{A}_t$-measurable. Moreover, it is time-consistent in the sense that for all $r \ge t$ and $s \le \tau(t)$ we have $\tilde{X}^{\le t}_s = \tilde{X}^{\le r}_s$. Clearly, $X_{t_i,j}, t_i, M_{t_i}$ can be reconstructed from the coordinates of $\tilde{X}^{\le t}$ for all $t_i \le t$ and $j \in \{1 \le l \le d_X | M_{t_i,l} = 1\}$, hence, $\mathcal{A}_t = \sigma(\tilde{X}^{\le t})$. Moreover, it is easy to see that $\tilde{X}^{\le t} = \tilde{X}^{\le \tau(t)}$, consequently $\hat{X}_t$ is $\sigma(\tilde{X}^{\le \tau(t)})$-measurable. Therefore, the Doob-Dynkin Lemma (Taraldsen, 2018, Lemma 2) implies the existence of measurable functions $F_j : [0,T] \times [0,T] \times BV^c([0,T]) \to \mathbb{R}$ such that $\hat{X}_{t,j} = F_j(t, \tau(t), \tilde{X}^{\le \tau(t)})$. In Krach et al. (2022) convergence of the PD-NJ-ODE model to the conditional expectation $\hat{X}$ was shown under the following assumptions.

**Assumption 2.1.** *We assume that:*

(i) *For every $1 \le k,l \le K$, $M_k$ is independent of $t_l$ and of $n$, $\tilde{\mathbb{P}}(M_{k,j} = 1) > 0$ and $M_{0,j} = 1$ for all $1 \le j \le d_X$ (every coordinate can be observed at any observation time and $X$ is completely observed at 0) and $|M_k|_1 > 0$ for every $1 \le k \le K$ $\tilde{\mathbb{P}}$-almost surely (at every observation time at least one coordinate is observed).*

(ii) *The probability that any two observation times are closer than $\epsilon > 0$ converges to 0 when $\epsilon$ does, i.e., if $\delta(\tilde{\omega}) := \min_{0 \le i < n(\tilde{\omega})} |t_{i+1}(\tilde{\omega}) - t_i(\tilde{\omega})|$ then $\lim_{\epsilon \to 0} \tilde{\mathbb{P}}(\delta < \epsilon) = 0$.*

(iii) *Almost surely $X$ is not observed at a jump, i.e., $(\mathbb{P} \times \tilde{\mathbb{P}})(t_i \in \mathcal{J} | i \le n) = (\mathbb{P} \times \tilde{\mathbb{P}})(\Delta X_{t_i} \ne 0 | i \le n) = 0$ for all $1 \le i \le K$.*

(iv) *$F_j$ are continuous and differentiable in their first coordinate $t$ such that their partial derivatives with respect to $t$, denoted by $f_j$, are again continuous and there exists a $B > 0$ and $p \in \mathbb{N}_{\ge 1}$ such that for every $t \in [0,T]$ the functions $f_j, F_j$ are polynomially bounded in $X^\star$, i.e.,*

$$|F_j(\tau(t), \tau(t), \tilde{X}^{\le \tau(t)})| + |f_j(t, \tau(t), \tilde{X}^{\le \tau(t)})| \le B(X_t^\star + 1)^p.$$

    *(v) $X^\star$ is $L^{2p}$-integrable, i.e., $\mathbb{E}[(X_T^\star)^{2p}] < \infty$.*

    *(vi) The random number of observations $n$ is integrable, i.e., $\mathbb{E}_{\tilde{\mathbb{P}}}[n] < \infty$.*

**Remark 2.2.** *The assumption that $|M_k|_1 > 0$ for every $1 \le k \le K$ $\tilde{\mathbb{P}}$-almost surely is not needed in the proof. It is just added such that the denotation "observation time" is meaningful. When removing it, some of the observation times might be "pseudo" observation times, where no coordinate is actually observed.*

**Remark 2.3.** *It turns out that Assumption 2.1 (ii) is always satisfied in the described setting, hence, we can also leave it away. Indeed, $\lim_{\epsilon \to 0} \tilde{\mathbb{P}}(\delta < \epsilon) = 0$ holds, if $\tilde{\mathbb{P}}(\delta = 0) = 0$, which is satisfied by definition of the stopping times to be in strictly increasing order.*

Moreover, relaxations on the assumption of observing $X_0$ completely were discussed in Krach et al. (2022, Remark 2.2). Convergence is defined with respect to the following distance.

**Definition 2.4.** *Let $c_0 := c_0(k) := (\tilde{\mathbb{P}}(n \ge k))^{-1}$. A distance between càdlàg $\mathbb{A}$-adapted processes $Z, \xi : [0, T] \times (\Omega \times \tilde{\Omega}) \to \mathbb{R}^r$ is defined through the pseudo metrics*

$$d_k(Z, \xi) = c_0(k) \, \mathbb{E}_{\mathbb{P} \times \tilde{\mathbb{P}}} \left[ \mathbb{1}_{\{n \ge k\}} |Z_{t_k -} - \xi_{t_k -}|_2 \right], \tag{1}$$

*for $1 \le k \le K$ and two processes are called indistinguishable, if $d_k(Z, \xi) = 0$ for all $1 \le k \le K$.*

In particular, this distance compares two càdlàg processes at their left limits[2] at the observation times $t_k$.

The path-dependent generalisation of the Neural Jump ODE model (Herrera et al., 2021) uses the truncated signature transformation $\pi_m$ (Krach et al., 2022, Definition 3.4). The signature of a path is an (infinite) tuple of features, which allows to approximate any continuous function of the path by a linear combination of the signature terms. Therefore, it is a very powerful framework in learning theory. It is defined as follows.

**Definition 2.5.** *Let $J$ be a closed interval in $\mathbb{R}$ and $\mathbf{X} : J \to \mathbb{R}^d$ be a continuous path with finite variation. The signature of $\mathbf{X}$ is defined as the collection of iterated integrals*

$$S(\mathbf{X}) = \left( 1, \mathbf{X}_J^1, \mathbf{X}_J^2, \dots \right),$$

*where, for each $m \ge 1$,*

$$\mathbf{X}_J^m = \int_{\substack{u_1 < \cdots < u_m \\ u_1, \dots, u_m \in J}} d\mathbf{X}_{u_1} \otimes \cdots \otimes d\mathbf{X}_{u_m} \in (\mathbb{R}^d)^{\otimes m}.$$

*The truncated signature of $\mathbf{X}$ of order $m \in \mathbb{N}$ is defined as*

$$\pi_m(\mathbf{X}) = \left( 1, \mathbf{X}_J^1, \mathbf{X}_J^2, \dots, \mathbf{X}_J^m \right),$$

*i.e., the first $m + 1$ terms (levels) of the signature of $\mathbf{X}$. For a path of dimension $d$, the dimension of the truncated signature of order $m$ is*

$$\begin{cases} m + 1, & \text{if } d = 1, \\ \frac{d^{m+1} - 1}{d - 1}, & \text{if } d > 1. \end{cases}$$

Moreover, PD-NJ-ODE uses bounded output neural networks $f_{(\tilde{\theta}, \gamma)}$, where $\tilde{\theta}$ are the weights of the standard neural network and $\gamma > 0$ is the trainable parameter of the bounded output activation function (Krach et al., 2022, Definition 3.12)

$$\Gamma_\gamma : \mathbb{R}^d \to \mathbb{R}^d, x \mapsto x \cdot \min\left( 1, \frac{\gamma}{|x|_2} \right),$$

applied to the output of the standard neural network. By $\mathcal{N}$ we denote the set of all bounded output neural networks based on a set $\tilde{\mathcal{N}}$ of standard neural networks. In the following we assume that $\tilde{\mathcal{N}}$ is a set of standard feedforward neural networks with Lipschitz continuous activation functions, with id $\in \tilde{\mathcal{N}}$ that satisfies the standard universal approximation theorem with respect to the supremum-norm on compact sets, see for example Hornik (1991, Theorem 2).

---

[2]For a càdlàg process $Z$ we have (by right-continuity) that $Z_t = \lim_{\epsilon \downarrow 0} Z_{t+\epsilon}$, while the left limit $Z_{t-} = \lim_{\epsilon \downarrow 0} Z_{t-\epsilon}$ exists but does not need to coincide with $Z_t$. In particular, $Z_t = Z_{t-}$ if and only if the jump of the process at $t$, $\Delta Z_t = Z_t - Z_{t-}$, is 0.

**Definition 2.6.** *The Path-Dependent Neural Jump ODE (PD-NJ-ODE) model is given by*

$$
\begin{aligned}
H_0 &= \rho_{\theta_2}\left(0, 0, \pi_m(0), X_0\right), \\
dH_t &= f_{\theta_1}\left(H_{t-}, t, \tau(t), \pi_m(\tilde{X}^{\leq \tau(t)} - X_0), X_0\right) dt \\
&\quad + \left(\rho_{\theta_2}\left(H_{t-}, t, \pi_m(\tilde{X}^{\leq \tau(t)} - X_0), X_0\right) - H_{t-}\right) du_t, \\
Y_t &= \tilde{g}_{\tilde{\theta}_3}(H_t).
\end{aligned}
\tag{2}
$$

*The functions $f_{\theta_1}, \rho_{\theta_2} \in \mathcal{N}$ are bounded output feedforward neural networks and $\tilde{g}_{\tilde{\theta}_3} \in \tilde{\mathcal{N}}$ is a feedforward neural network with trainable parameters $\theta = (\theta_1, \theta_2, \tilde{\theta}_3) \in \Theta$, where $\theta_i = (\tilde{\theta}_i, \gamma_i)$ for $i \in \{1, 2\}$ and $\Theta$ is the set of all possible weights for the PD-NJ-ODE; $m \in \mathbb{N}$ is the signature truncation level and $u$ is the jump process counting the observations defined as $u_t := \sum_{k=1}^{K} \mathbb{1}_{t_k \leq t}$.*

In the PD-NJ-ODE model, $H_t$ has the role of a latent process and $Y_t$ is the model output (similar to the latent variable and model output of an RNN). The goal of the model output $Y_t$ is to approximate the true conditional expectation $\hat{X}_t$. $H_t$ evolves continuously between any two observations, according to the *neural ODE* defined by $f_{\theta_1}$. At an observation time, when there is a "jump" in the available information, $H_t$ jumps according to $\rho_{\theta_2}$, which can be interpreted as an *RNN-cell*. The latent process $H_t$ is mapped to the output process $Y_t$ through the *readout map* $\tilde{g}_{\tilde{\theta}_3}$. The existence and uniqueness of a solution $(H, Y)$ of (2) for a fixed $\theta$ is implied by Cohen & Elliott (2015, Theorem 16.3.11)[3]. To emphasize the dependence of the PD-NJ-ODE output $Y$ on $\theta$ and $X, (t_i)_{1 \leq i \leq K}$ and $M$ we write $Y^\theta(\tilde{X}^{\leq \tau(\cdot)})$ (since $\tilde{X}^{\leq \tau(\cdot)}$ summarizes $X, (t_i)_{1 \leq i \leq K}$ and $M$).

The objective function (cf. *equivalent objective function* from Remark 4.8 & Appendix D.1.5 of Krach et al. (2022)) for the training of the PD-NJ-ODE is defined as

$$
\Psi : \mathbb{D} \to \mathbb{R}, \ Z \mapsto \Psi(Z) := \mathbb{E}_{\mathbb{P} \times \tilde{\mathbb{P}}}\left[\frac{1}{n} \sum_{i=1}^{n} \left(|M_i \odot (X_{t_i} - Z_{t_i})|_2 + |M_i \odot (X_{t_i} - Z_{t_i-})|_2\right)^2\right], \tag{3}
$$

$$
\Phi : \Theta \to \mathbb{R}, \ \theta \mapsto \Phi(\theta) := \Psi(Y^\theta(\tilde{X}^{\leq \tau(\cdot)})), \tag{4}
$$

where $\odot$ is the element-wise multiplication (Hadamard product) and $\mathbb{D}$ the set of all càdlàg $\mathbb{R}^{d_X}$-valued $\mathbb{A}$-adapted processes on the product probability space $\Omega \times \tilde{\Omega}$. The two terms in (4) compare the distances between $X_{t_k}$ and $Z$ after and before its jump at observation times $t_k$. Intuitively speaking, to minimize the second term, $Z_{t_k-}$ has to be the best possible prediction of $X_{t_k}$ given the information available before the new observation becomes available; and to minimize the first term, $Z$ has to jump to the new observation $X_{t_k}$ after it became available.

For $N \in \mathbb{N}$ the number of training paths and for every $1 \leq j \leq N$, let $(X^{(j)}, M^{(j)}, n^{(j)}, t_1^{(j)}, \ldots, t_{n^{(j)}}^{(j)}) \sim (X, M, n, t_1, \ldots, t_n)$ be independent and identically distributed (i.i.d.) random processes and variables with the same distribution as our initially introduced process $X$ together with its random observation framework. Then the Monte Carlo approximation of (4) is

$$
\hat{\Phi}_N(\theta) := \frac{1}{N} \sum_{j=1}^{N} \frac{1}{n^{(j)}} \sum_{i=1}^{n^{(j)}} \left(\left|M_i^{(j)} \odot \left(X_{t_i^{(j)}}^{(j)} - Y_{t_i^{(j)}}^{\theta,j}\right)\right|_2 + \left|M_i^{(j)} \odot \left(X_{t_i^{(j)}}^{(j)} - Y_{t_i^{(j)}-}^{\theta,j}\right)\right|_2\right)^2, \tag{5}
$$

where $Y^{\theta,j} := Y^\theta(\tilde{X}^{\leq \tau(\cdot),(j)})$.

Based on these loss functions, the following convergence guarantees can be derived, where $\Theta_m \subset \Theta$ is defined as the set of possible parameters for the 3 (bounded output) neural networks, such that their widths and

---

[3]$f_{\theta_1}$ and $\rho_{\theta_2}$ are Lipschitz continuous as neural networks with Lipschitz continuous activation functions, hence, the stochastic functions $((\omega, \tilde{\omega}), t, H) \mapsto f_{\theta_1}(H_{t-}, t, \tau(t, \tilde{\omega}), \pi_m(\tilde{X}^{\leq \tau(t)} - X_0)(\omega, \tilde{\omega}), X_0(\omega))$, and similarly for $\rho_{\theta_2}$, are uniformly Lipschitz according to Cohen & Elliott (2015, Definition 16.3.2). Moreover, it is immediate to see that these functions are *coefficients* according to Cohen & Elliott (2015, Definition 16.0.3), since they are continuous, hence predictable, and since we integrate with respect to finite variation processes. In particular, integrability (Cohen & Elliott, 2015, Definition 12.3.10) is trivially satisfied (path-wise), because any continuous bounded function is Stieltjes-integrable.

depths are at most $m$ and such that the truncated signature of level $m$ or smaller is used and such that the norms of the weights $\tilde{\theta}_i$ and the bounds $\gamma_i$ are bounded by $m$ (i.e., $|\tilde{\theta}_i|_2 \le m, \gamma_i \le m$). Thus, $\Theta_m$ is a compact subset of $\Theta$.

**Theorem 2.7.** *Let $\theta_m^{\min} \in \Theta_m^{\min} := \operatorname{argmin}_{\theta \in \Theta_m}\{\Phi(\theta)\}$ for every $m \in \mathbb{N}$. If Assumption 2.1 is satisfied, then, for $m \to \infty$, the value of the loss function $\Phi$ converges to the minimal value of $\Psi$ which is uniquely achieved by $\hat{X}$ up to indistinguishability, i.e.,*

$$\Phi(\theta_m^{\min}) \xrightarrow{m \to \infty} \min_{Z \in \mathbb{D}} \Psi(Z) = \Psi(\hat{X}).$$

*Furthermore, for every $1 \le k \le K$ we have that $Y^{\theta_m^{\min}}$ converges to $\hat{X}$ in the metric $d_k$ as $m \to \infty$.*

*Let $\theta_{m,N}^{\min} \in \Theta_{m,N}^{\min} := \operatorname{argmin}_{\theta \in \Theta_m}\{\hat{\Phi}_N(\theta)\}$ for every $m, N \in \mathbb{N}$. Then, for every $m \in \mathbb{N}$, $(\mathbb{P} \times \tilde{\mathbb{P}})$-a.s.*

$$\hat{\Phi}_N \xrightarrow{N \to \infty} \Phi \quad \textit{uniformly on } \Theta_m.$$

*Moreover, for every $m \in \mathbb{N}$, we have a.s.,*

$$\Phi(\theta_{m,N}^{\min}) \xrightarrow{N \to \infty} \Phi(\theta_m^{\min}) \quad \textit{and} \quad \hat{\Phi}_N(\theta_{m,N}^{\min}) \xrightarrow{N \to \infty} \Phi(\theta_m^{\min}).$$

*In particular, one can define an increasing random sequence $(N_m)_{m \in \mathbb{N}}$ in $\mathbb{N}$ such that for every $1 \le k \le K$ we have a.s. that $Y^{\theta_{m,N_m}^{\min}}$ converges to $\hat{X}$ in the metric $d_k$ as $m \to \infty$.*

This result shows that, given we find the minimizer $\theta_{m,N_m}^{\min}$ of the loss function, the output $Y^{\theta_{m,N_m}^{\min}}$ of the PD-NJ-ODE model converges to the optimal prediction, i.e., to the true conditional expectation $\hat{X}$, in the metrics $d_k$. This convergence holds for almost every realization of the training data, which is used to derive the minimizer $\theta_{m,N_m}^{\min}$, when evaluated on independent test data. In particular, this result verifies that the model can approximate $\hat{X}$ arbitrarily well and that minimizing the loss function is expedient to find such an approximation of $\hat{X}$. In Section 5, we further discuss the practical implication of this convergence result. In this work, we do not focus on the task of finding the minimizer for the loss function, which is an independent and well studied problem on its own. Different optimization schemes exists, which yield convergence to global or local optima. These can be combined with our results as was further discussed in Herrera et al. (2021, Appendix E.2). In our experiments we use Adam (Kingma & Ba, 2014), an stochastic gradient descent (SGD) version, which yields good empirical results.

## 3 PD-NJ-ODE with Noisy Observations

So far the PD-NJ-ODE model was only applicable when having noise-free observations of $X$. In particular, we demonstrate that the PD-NJ-ODE with the original loss yields incorrect predictions in the presence of measurement noise, i.e., i.i.d. noise terms that are added to each observation of $X$. Using the stochastic filtering approach described in Krach et al. (2022, Section 6) would be a possibility to include (discrete or continuous) noise. However, this requires strong additional assumptions (i.e., the knowledge of the joint distribution of the process and the noise or equivalently training samples split up into the noise-free observations and the noise terms) which are not satisfied easily. Therefore, we want to adapt our framework, such that it can be applied to noisy observations, while only imposing weak assumptions that are easily satisfied. In particular, we introduce a method that can deal with noisy observations, even if we have never seen any noise-free observation during training. In contrast to classical filtering, there is no need to have a strong prior knowledge of the underlying dynamics.

In this section, we introduce observation noise (e.g., measurement-noise), i.e., i.i.d. noise terms $\epsilon_i$ that are added to the process $X$ at each observation time $t_i$, leading to the noisy observations $O_{t_i} := X_{t_i} + \epsilon_i$. Even though we only observe the $O_{t_i}$, the goal still is to predict $X$, in particular to compute $\mathbb{E}[X_t \mid O_{t_0}, \ldots, O_{\tau(t)}]$.

Inspecting the PD-NJ-ODE model and its loss function (replacing $X$ by $O$), we notice two things. First, that nothing in the model's architecture prevents it from learning this modified objective, which should still have the same properties. Second, that the loss function needs to be modified. Indeed, the first term of the

loss function would train the model to jump to the noisy observation $O_{t_i}$. This would be incorrect, since in general $\mathbb{E}[X_{t_i} \,|\, O_{t_0}, \ldots, O_{t_i}] \neq O_{t_i}$ because the conditional expectation of $X$ filters out the noise as well as possible. We therefore drop the first term of the loss function.

On the other hand, it is easy to see that the conditional expectations of $X$ and $O$ coincide in between observation times[4] if the observation noise $\epsilon_i$ is independent of the observations and has mean 0. Therefore, the second term of the loss function is minimised if the model learns the conditional expectation $\mathbb{E}[X_t \,|\, O_{t_0}, \ldots, O_{\tau(t)}]$ between observation times.

Along these lines, it turns out that it suffices to omit the first term of the loss function to recover the original results of Theorem 2.7 under noisy observations. In particular, to optimize the loss, the model learns to jump to the conditional expectation of $X$ at observation times even without the respective loss term. Indeed, since it evolves continuously after an observation, it would otherwise be different from the optimal prediction right after the observation time and therefore would not optimize the loss. In the following this is formalised.

### 3.1 Setting with Noisy Observations

The process $X$ as well as the $n, K, t_i, \tau, M$ are defined as in Section 2. Additionally, we define

- $(\epsilon_k)_{0 \leq k \leq K}$, the observation noise, which is a sequence of i.i.d. random variables on $(\tilde{\Omega}, \tilde{\mathcal{F}}, \tilde{\mathbb{P}})$ taking values in $\mathbb{R}^{d_X}$,

- $O_{t_k} := X_{t_k} + \epsilon_k$ for $0 \leq k \leq n$, the noisy observation sequence.

Since the goal is to predict $X$ given the observations $O_{t_i}$, we redefine the filtration of the currently available information via
$$\mathcal{A}_t := \boldsymbol{\sigma}\left(O_{t_i,j}, t_i, M_{t_i} | t_i \leq t, \, j \in \{1 \leq l \leq d_X | M_{t_i,l} = 1\}\right),$$
such that $\hat{X}_t = \mathbb{E}_{\mathbb{P} \times \tilde{\mathbb{P}}}[X_t | \mathcal{A}_t]$ is the conditional expectation of $X$ given the noisy observations. We define $\tilde{O}^{\leq t}$ in the same way as $\tilde{X}^{\leq t}$ and note that similarly as before there exist measurable functions $F_j$ such that $\hat{X}_{t,j} = F_j(t, \tau(t), \tilde{O}^{\leq \tau(t)})$. We need the following slight modification of Assumption 2.1.

**Assumption 3.1.** *We assume that Assumption 2.1 items (i) to (iii) and (v) hold and additionally that:*

(iv) *$F_j$ are continuous and differentiable in their first coordinate $t$ such that their partial derivatives with respect to $t$, denoted by $f_j$, are again continuous and there exists a $B > 0$ and $p \in \mathbb{N}$ such that for every $t \in [0, T]$ the functions $f_j, F_j$ are polynomially bounded in $X^\star$, i.e.,*

$$|F_j(\tau(t), \tau(t), \tilde{O}^{\leq \tau(t)})| + |f_j(t, \tau(t), \tilde{O}^{\leq \tau(t)})| \leq B(X_t^\star + 1)^p + B \sum_{i=0}^{n} |\epsilon_i|.$$

(vi) *$n$ is square-integrable, i.e., $\mathbb{E}_{\tilde{\mathbb{P}}}[|n|^2] < \infty$.*

(vii) *The i.i.d. random noise variables $\epsilon_k$ are independent of $X, n, M, (t_i)_{1 \leq i \leq K}$, are centered and square-integrable, i.e., $\mathbb{E}_{\tilde{\mathbb{P}}}[\epsilon_k] = 0$ and $\mathbb{E}_{\tilde{\mathbb{P}}}[|\epsilon_k|^2] < \infty$.*

**Remark 3.2.** *The relaxations on the assumption of observing $X_0$ completely discussed in Krach et al. (2022, Remark 2.2) can equivalently be applied in this setting here.*

In this setting, the PD-NJ-ODE uses the noisy observations $O_{t_i}$ and $\tilde{O}^{\leq \tau(t)}$ as inputs instead of $X_{t_i}$ and $\tilde{X}^{\leq \tau(t)}$. Moreover, we define the new *noise-adapted* objective function as described before as

$$\Psi : \mathbb{D} \to \mathbb{R}, \, Z \mapsto \Psi(Z) := \mathbb{E}_{\mathbb{P} \times \tilde{\mathbb{P}}}\left[\frac{1}{n}\sum_{i=1}^{n} |M_i \odot (O_{t_i} - Z_{t_i-})|_2^2\right], \quad (6)$$

$$\Phi : \Theta \to \mathbb{R}, \, \theta \mapsto \Phi(\theta) := \Psi(Y^\theta(X)), \quad (7)$$

and its Monte Carlo approximation $\hat{\Phi}_N$ accordingly.

---

[4]Note that, in general we have $\mathbb{E}[X_{t_i} \,|\, O_{t_0}, \ldots, O_{t_i}] \neq \mathbb{E}[O_{t_i} \,|\, O_{t_0}, \ldots, O_{t_i}] = O_{t_i}$ at observation times, in contrast to their equality between observation times.

### 3.2 Convergence Theorem with Noisy Observations

In the setting defined in Section 3.1, Theorem 2.7 holds equivalently as before.

**Theorem 3.3.** *If Assumption 3.1 is satisfied and using the definitions of Section 3.1, the claims of Theorem 2.7 hold equivalently, upon replacing the original loss functions and their Monte Carlo approximations by their noise-adapted versions. In particular, we obtain convergence of our estimator $Y^{\theta_{m,N_m}^{\min}}$ to the true conditional expectation $\hat{X}$ in $d_k$.*

To prove this, we first need to adjust the orthogonal projection result (Krach et al., 2022, Lemma 4.2) for this setting. At this point we want to highlight the relevance of this result as the backbone of the proof of Theorem 3.3, which is first used to show that $\hat{X}$ is a minimizer of $\Psi$, then to show that it is unique and finally to bound the distance between $\hat{X}$ and $Y^{\theta_m^{\min}}$ in $d_k$ through the difference of their loss function values.

**Lemma 3.4.** *For any $\mathbb{A}$-adapted process $Z$ it holds that*

$$\mathbb{E}_{\mathbb{P}\times\tilde{\mathbb{P}}}\left[\frac{1}{n}\sum_{i=1}^{n}|M_{t_i}\odot(O_{t_i}-Z_{t_i-})|_2^2\right]$$
$$= \mathbb{E}_{\mathbb{P}\times\tilde{\mathbb{P}}}\left[\frac{1}{n}\sum_{i=1}^{n}\left|M_{t_i}\odot(O_{t_i}-\hat{X}_{t_i-})\right|_2^2\right] + \mathbb{E}_{\mathbb{P}\times\tilde{\mathbb{P}}}\left[\frac{1}{n}\sum_{i=1}^{n}\left|M_{t_i}\odot(\hat{X}_{t_i-}-Z_{t_i-})\right|_2^2\right].$$

*Proof.* First note that by Assumption 3.1 point (iii) we have that $X_{t_i}=X_{t_i-}$ almost surely and when defining $O_{t_i-}:=X_{t_i-}+\epsilon_i$ we therefore also have that $O_{t_i}=O_{t_i-}$ almost surely. Similarly as in Krach et al. (2022, Lemma 4.2), we can derive for $\hat{O}_{t_i-}:=\mathbb{E}_{\mathbb{P}\times\tilde{\mathbb{P}}}[O_{t_i-}|\mathcal{A}_{t_i-}]$ that

$$\mathbb{E}_{\mathbb{P}\times\tilde{\mathbb{P}}}\left[\frac{1}{n}\sum_{i=1}^{n}|M_{t_i}\odot(O_{t_i-}-Z_{t_i-})|_2^2\right]$$
$$= \mathbb{E}_{\mathbb{P}\times\tilde{\mathbb{P}}}\left[\frac{1}{n}\sum_{i=1}^{n}\left|M_{t_i}\odot(O_{t_i-}-\hat{O}_{t_i-})\right|_2^2\right] + \mathbb{E}_{\mathbb{P}\times\tilde{\mathbb{P}}}\left[\frac{1}{n}\sum_{i=1}^{n}\left|M_{t_i}\odot(\hat{O}_{t_i-}-Z_{t_i-})\right|_2^2\right].$$

To conclude the proof, it is enough to note that

$$\hat{O}_{t_i-}=\hat{X}_{t_i-}+\mathbb{E}[\epsilon_i|\mathcal{A}_{t_i-}]=\hat{X}_{t_i-}+\mathbb{E}[\epsilon_i]=\hat{X}_{t_i-}, \tag{8}$$

using that $\epsilon_i$ has expectation 0 and is independent of $\mathcal{A}_{t_i-}$. □

In the following we sketch the proof of Theorem 3.3, by only outlining those parts of it that need to be changed in comparison with the original proof of Theorem 2.7 in Krach et al. (2022). The main differences are that the loss function needs to be adjusted whenever used, and when showing integrability, we additionally have to account for the noise terms $\epsilon_k$. A full proof is given in Appendix C.

*Sketch of Proof of Theorem 3.3.* First, it follows directly from Lemma 3.4 that $\Psi(\hat{X})=\min_{Z\in\mathbb{D}}\Psi(Z)$, i.e., that $\hat{X}$ is a minimizer of the redefined objective function $\Psi$. Secondly, again by Lemma 3.4, we have for any process $Z\in\mathbb{D}$ that

$$\Psi(Z)=\Psi(\hat{X})+\mathbb{E}_{\mathbb{P}\times\tilde{\mathbb{P}}}\left[\frac{1}{n}\sum_{i=1}^{n}\left|M_{t_i}\odot(\hat{X}_{t_i-}-Z_{t_i-})\right|_2^2\right],$$

hence $\Psi(Z)>\Psi(\hat{X})$ follows as before if $Z$ is not indistinguishable from $\hat{X}$, meaning that $\hat{X}$ is the unique minimizer of $\Psi$.

Under Assumption 3.1 the approximation of the functions $f_j, F_j$ by bounded output feedforward neural networks works similarly as before, with the slight adjustment that their differences are now upper bounded by

$$U := 3B\left((X_T^\star+1)^p+\sum_{i=0}^{n}|\epsilon_i|\right).$$

Defining

$$c_m := c\,\varepsilon(T+1)d_X + c(T+1)d_X U\left(\mathbb{1}_{\{X_T^\star \geq 1/\varepsilon\}} + \mathbb{1}_{\{n \geq 1/\varepsilon\}} + \mathbb{1}_{\{\delta \leq \varepsilon\}}\right)$$

it follows that there exists $\theta_m^* \in \Theta_m$ such that $\left|Y_t^{\theta_m^*} - \hat{X}_t\right|_2 \leq c_m$ for all $t \in [0,T]$. Convergence of $\Phi(\theta_m^\star)$ to $\Psi(\hat{X})$ then follows similarly as before, when noting that by Assumption 3.1

$$\mathbb{E}\left[(X_t^\star + 1)^{2p} + \left(\sum_{i=0}^{n}|\epsilon_i|\right)^2\right] \leq \mathbb{E}\left[(X_t^\star + 1)^{2p}\right] + \mathbb{E}[n^2]\mathbb{E}\left[|\epsilon_0|^2\right] < \infty, \tag{9}$$

using Cauchy–Schwarz and that the $\epsilon_i$ are i.i.d. and independent of $n$ for the first step and the integrability of $X^\star$, $\epsilon_0$ and $n^2$ for the upper bound. Moreover, the convergence of $d_k(\hat{X}, Y^{\theta_m^\star}) \to 0$ follows as before.

Finally, the remaining claims of the theorem (including the Monte Carlo convergence) also hold similarly as before upon replacing $X_{t_i}$ by $O_{t_i}$ and noting that the integrability of $\sup_\theta h(\theta, \xi_j)$ follows from (9). $\qquad\square$

### 3.3 More General Noise Structure & Conditional Moments

Revisiting the proof in Section 3.2, we see that the noise terms need neither be independent nor centered. If we assume that the conditional bias of the noise,

$$\beta_i(\tilde{O}^{\leq \tau(t)}) := \mathbb{E}[\epsilon_i | \mathcal{A}_{t_i-}],$$

is a *known* function of the observations (using Doob-Dynkin Lemma (Taraldsen, 2018, Lemma 2) for its existence), then we can modify the objective function by subtracting it. This leads to

$$\Psi : \mathbb{D} \to \mathbb{R},\ Z \mapsto \Psi(Z) := \mathbb{E}_{\mathbb{P} \times \tilde{\mathbb{P}}}\left[\frac{1}{n}\sum_{i=1}^{n}\left|M_i \odot \left((O_{t_i} - \beta_i(\tilde{O}^{\leq t_{i-1}})) - Z_{t_i-}\right)\right|_2^2\right]. \tag{10}$$

Revisiting (8), which is the only part of the proof where we needed the noise terms to be centered, we see that

$$\mathbb{E}\left[(O_{t_i-} - \beta_i(\tilde{O}^{\leq t_{i-1}}))|\mathcal{A}_{t_i-}\right] = \hat{X}_{t_i-} + \mathbb{E}[\epsilon_i \mid \mathcal{A}_{t_i-}] - \beta_i = \hat{X}_{t_i-}. \tag{11}$$

This implies that the statement of Lemma 3.4 holds equivalently under the reduced assumption of a known conditional bias function, when using the adjusted loss (10). Additionally assuming that $\mathbb{E}\left[\left(\sum_{i=0}^{n}|\epsilon_i|\right)^2\right] < \infty$, the following result follows as before.

**Corollary 3.5.** *In the setting described in this sub-section (i.e., arbitrary known mean of the noise and no independence assumption on the noise), which is a generalisation of the setting in Section 3.1, Theorem 2.7 holds equivalently as before when using the objective function* (10).

The following remark explains how Corollary 3.5 can be used to predict conditional higher moments (instead of only the conditional expectation) under certain assumptions.

**Remark 3.6.** *This result makes it possible to compute the conditional moments of $X$ given the noisy observations, which doesn't work in the setting of Section 3.1. In particular, we consider observations $O_{t_i-} = X_{t_i-} + \epsilon_i$, where we assume that*

- *$\epsilon_i$ is independent of $\mathcal{A}_{t_i-}$,*
- *$\epsilon_i$ is conditionally independent of $X_{t_i-}$ given $\mathcal{A}_{t_i-}$*
- *and $\epsilon_i$ have known finite moments.*

*Remark that Proposition A.8 implies that the first two assumptions are in particular satisfied if $\epsilon_i$ is independent of $\sigma(\mathcal{A}_{t_i-}, X_{t_i-})$. The binomial theorem implies for any $q \in \mathbb{N}$*

$$O_{t_i-}^q = X_{t_i-}^q + \sum_{j=1}^{q}\binom{q}{j}X_{t_i-}^{q-j}\epsilon_i^j.^{[5]}$$

---

[5]Note that $q$, $j$ and $q-j$ denote exponents here rather than superscripts.

We interpret the entire sum as the observation noise and accordingly define the conditional bias of the observation noise of the $q$-th moment as

$$\beta_i^q := \mathbb{E}\left[\sum_{j=1}^q \binom{q}{j} X_{t_i-}^{q-j} \epsilon_i^j \mid \mathcal{A}_{t_i-}\right] = \sum_{j=1}^q \binom{q}{j} \mathbb{E}[X_{t_i-}^{q-j}|\mathcal{A}_{t_i-}]\mathbb{E}[\epsilon_i^j],$$

where we use the assumptions on $\epsilon_i$ together with Proposition A.7 for the second equality.

Then an inductive argument shows that $\beta_i^q$ is a known function of the observations, using the assumption that the moments of $\epsilon_i$ are known. Indeed, to compute $\beta_i^q$ the conditional expectations of smaller moments $\mathbb{E}[X_{t_i-}^{q-j}|\mathcal{A}_{t_i-}]$ need to be computed, which can be done according to the induction hypothesis (note that the base case follows directly from Corollary 3.5 and the assumptions on $\epsilon_i$). Therefore, Corollary 3.5 implies that we can compute $\mathbb{E}[X_{t_i-}^q|\mathcal{A}_{t_i-}]$ (assuming that we reach the limit where the PD-NJ-ODE output equals the conditional expectation). In case of an exponential moment assumption $\mathbb{E}[\exp(\lambda|X_{t_i-}|)] < \infty$ for some $\lambda > 0$ we can therefore infer the conditional law of $X_{t_i-}$.

In the remainder of this section, we discuss the cases of linear and non-linear (noisy) observation models.

**Remark 3.7** (Linear Observation Model)**.** *Let us consider a linear observation process*

$$O_{t_k} := AX_{t_k} + \epsilon_k,$$

*for some fixed matrix $A \in \mathbb{R}^{d \times d_X}$, for $d \in \mathbb{N}$, where we have observations of $O$, but are ultimately interested in predicting $X$ without having access to any observations of $X$. For this to be suitable for our framework, we need that Assumption 3.1 is satisfied for $Z := AX$.[6] Then we can apply our method to approximate $\hat{Z}_t = \mathbb{E}[Z_t \mid \mathcal{A}_t]$, as discussed before. To be able to infer predictions for $X$ from $\hat{Z}$, we need to further assume that $A$ defines an injective map, hence, a left inverse $A^{-1}$ with $A^{-1}A = \mathrm{id}$ exists. Under this assumption, $\mathbb{E}[X_t \mid \mathcal{A}_t] = A^{-1}\hat{Z}_t$, by linearity. It is important to note that in an incomplete observation setting, it would in general not be possible to apply our method to directly approximate $X$, since (noisy) observations of $X$ cannot be computed from (noisy) observations of $Z$, if they are incomplete (i.e., $A^{-1}O$ might not be well defined). Therefore, the detour via first computing $\hat{Z}$ is essential. Moreover, it should be apparent that in the case the map defined by $A$ is not injective (or at least injective on the support of $X$), information about $X$ will be lost when only observing $O$. Nevertheless, any process of the form $BZ$ for another matrix $B$ can be predicted similarly as described above when observing $O$.*

**Remark 3.8** (Non-linear Observation Model & Beyond)**.** *Let us consider the observation process*

$$O_{t_k} := \phi(X_{t_k}) + \epsilon_k,$$

*for some measurable function $\phi$, where we again have observations of $O$ (in particular we assume that $\phi(X)$ satisfies the assumptions), but are ultimately interested in predicting $X$. Then, even if a left inverse $\phi^{-1}$ with $\phi^{-1} \circ \phi = \mathrm{id}$ exists, the technique from Remark 3.7 does not work, since $\mathbb{E}[X_t \mid \mathcal{A}_t] \neq \phi^{-1}(\mathbb{E}[\phi(X)_t \mid \mathcal{A}_t])$ in general. However, under the stronger assumption that we have access to (noisy) samples of the joint process $\xi := (X, \phi(X))$ for training and that this $\xi$ satisfies Assumption 3.1, we can compute $\hat{\xi} = (\hat{X}, \widehat{\phi(X)})$, which can then also be evaluated for samples where only (noisy) observations of the $\phi(X)$-coordinates are available (cf. Krach et al. (2022, Corollary 6.2)). Moreover, we note that the same is true, when replacing $\phi(X)$ by a general process $Z$.*

### 3.4 Examples of Processes Satisfying the Assumptions

In principle, all the examples presented in Krach et al. (2022, Section 7) are valid examples for this setting when adding some type of i.i.d. observation noise satisfying our assumptions, as e.g. Gaussian or uniform noise. However, it is important to note that the (true) conditional expectation is not the same, since we now condition on the noisy observations $O_{t_i}$ instead of the original observations $X_{t_i}$. Therefore, we give one explicit example where we compute the conditional expectation in the noisy observation setting.

---

[6]We note that by linearity of the (conditional) expectation we have $\mathbb{E}[Z_t \mid \mathcal{A}_t] = A\,\mathbb{E}[X_t \mid \mathcal{A}_t]$, hence, $Z$ satisfies Assumptions 3.1 (iii) to (v), if $X$ does.

### 3.4.1 Brownian Motion with Gaussian Observation Noise

Let $X := W$ be a standard Brownian motion and let $\epsilon_0 = 0$, $\epsilon_i \sim \mathcal{N}(0, \sigma^2)$ for $i \geq 1$ be the i.i.d. noise terms for some $\sigma > 0$. Then $O_{t_i} = X_{t_i} + \epsilon_i$ are the observations. Clearly, all integrability assumptions are satisfied by $X$ and $\epsilon_i$ (cf. Krach et al. (2022, Section 7.5)). To compute the true conditional expectations we first note that the independent increments property of the Brownian motion imply for $t_k \leq t < t_{k+1}$

$$\mathbb{E}[X_t|\mathcal{A}_t] = \mathbb{E}[W_t - W_{t_k}|\mathcal{A}_{t_k}] + \mathbb{E}[W_{t_k}|\mathcal{A}_{t_k}] = \mathbb{E}[W_{t_k}|\mathcal{A}_{t_k}] = \mathbb{E}[W_{t_k}|O_{t_1}, \ldots, O_{t_k}],$$

and therefore, $f(s, \tau(t), \tilde{O}^{\leq \tau(t)}) = 0$. Since $W$ is a Brownian motion and $\epsilon_i$ are independent i.i.d. Gaussian noise terms, we know that

$$v := (O_{t_1}, \ldots, O_{t_k}, W_{t_k})^\top \sim \mathcal{N}(0, \Sigma)$$

where

$$\Sigma = \begin{pmatrix} \Sigma_{11} & \Sigma_{12} \\ \Sigma_{21} & \Sigma_{22} \end{pmatrix} \in \mathbb{R}^{(k+1)\times(k+1)},$$

with $\Sigma_{11} \in \mathbb{R}^{k \times k}$ and $(\Sigma_{11})_{i,j} = \min(t_i, t_j) + \sigma^2 \mathbb{1}_{\{i=j\}}$, $\Sigma_{12}^\top = \Sigma_{21} = (t_1, \ldots, t_k) \in \mathbb{R}^{1 \times k}$ and $\Sigma_{22} = t_k$. Then the conditional distribution of $(W_{t_k}|O_{t_1}, \ldots, O_{t_k})$ is again normal with mean $\hat{\mu} := \Sigma_{21}\Sigma_{11}^{-1}(O_{t_1}, \ldots, O_{t_k})^\top$ and variance $\hat{\Sigma} := \Sigma_{22} - \Sigma_{21}\Sigma_{11}^{-1}\Sigma_{12}$ (Eaton, 2007, Proposition 3.13). In particular we have

$$\mathbb{E}[X_t|\mathcal{A}_t] = \mathbb{E}[W_{t_k}|O_{t_1}, \ldots, O_{t_k}] = \hat{\mu}.$$

### 3.5 A Practical Note on Using the Noise-Adapted Loss Function

We have seen in Section 3.2 and 3.3, considering the limit where network size $m$ and training samples $N$ go to infinity, that there is no disadvantage of the noise-adapted loss function (10) compared to the original loss function (3). In particular, the noise-adapted training framework yields the optimal solution with noisy observations, but also if there is no observation noise. Indeed, setting $\epsilon_k = 0$ for all $k$, we have $O_{t_k} = X_{t_k}$ and therefore the original result also holds with the noise-adapted loss function. Hence, the question arises, why one should use the original loss function (3) at all anymore?

To answer this question, we first note that in practice we are not in the limit case, but we only have access to a finite training set (i.e., we only observe a finite amount of observations $n^{(j)}$ from finite number of paths $N$). Hence, the inductive bias when training the model becomes more important. See Appendix B for more details on the inductive bias. In particular, using the original loss function (3) in a setting without observation noise is preferable, since it directly penalizes the model for not jumping to observations and therefore making it easier for the model to learn this behaviour. Indeed, in the noise-adapted loss function, the penalization for not jumping to an observation is more indirect, since it will only be punished at the following observation time until which the output of the model further evolves through the neural ODE. Therefore, the feedback signal is weaker, i.e., the respective gradients are smaller.

In a setting with observation noise, the practical relevance of the noise-adapted loss function depends on the size of the variance of the noise, $\text{Var}_{noise}$, compared to the size of the variance of the data, $\text{Var}_{data}$. Clearly, if the variance ratio $\frac{\text{Var}_{noise}}{\text{Var}_{data}}$ is small, the noise does not have a large impact on the observations and therefore neither on the training of the model when using the original loss function (3). Due to the better inductive bias during training, it might therefore be beneficial to use (3) instead of the noise-adapted loss (10). On the other hand, if the variance ratio is large, the impact of the noise on the observations is substantial and therefore the noise-adapted training framework leads to better results. This behaviour is well visible in the experiment presented in Section 6.3. The turning point, where training with the noise-adapted loss function becomes better, is problem specific (e.g., the number of training samples has a big influence) and, if in doubt, we suggest to train the PD-NJ-ODE model with both loss functions and compare their results.

## 4 PD-NJ-ODE with Dependence between $X$ and the Observation Framework

Recall that the observation mask process is given by $M$ and the underlying process by $X$. In this section, we remove the assumptions that the observation times are independent of $X$, and that $M$ is independent

of the observation times and of $X$. In essence, the model is now defined on only one probability space $\mathbb{P}$ and no independence assumptions between the random variables are made. Instead, we need some weaker conditional independence assumptions to recover the results of Theorem 2.7.

### 4.1 Intuition on independence assumptions

In many real-world applications, the independence of the process $X$ and the observation framework (i.e., $(t_i)_{i \in \{1, \dots, n\}}$ and $M$) is heavily violated. For example, consider the irregular measurements of a patient's health parameters. A nurse or doctor will only take (expensive) measurements if information on the patient's state, $X$, hints that the measurement is necessary. In practice, different measurements are taken from different patients depending on observations of their state. This motivates the crucial importance of lifting the independence assumption for real-world applications.

However, even in this paper, we cannot completely remove any independence assumption; we still need conditional independence of the process $X$ and the observation framework, given all past observations (as we make precise in Assumption 4.1 items (vii) and (ix) and Proposition 4.9). This is a realistic assumption, which we show by continuing our hospital example. If we assume that *every* piece of information the hospital gets from the patient is immediately logged as an observation (perhaps noisy or incomplete) of $X$, then all information about $X$ that the hospital has is contained entirely within these observations. Thus measurement decisions are conditionally independent of $X$, given the past observations (if there are no further hidden confounders).

On the other hand, it is easy to violate this assumption. Imagine that the patient tells the nurse and doctor she feels feverish and they subsequently take her temperature. If they log her temperature, but not her feeling, the assumption is violated, as the hospital has a piece of information that is not logged as an observation of $X$. In such cases where information is not logged as an observation, our model (and most other classical forecasting methods) would learn a biased forecast of the body temperature. In the extreme case that body temperature is only measured if patients feel very feverish, measurements will record a high temperature, which leads our model to always predict a high body temperature (i.e., the expected body temperature conditioned on feeling feverish and all other past observations) even if the patient does not feel feverish.

There are often ways to mitigate such issues. In our case, for example, by first logging the feverish feeling and then the actual measurement at a later time stamp. Moreover, a non-feverish feeling has to be logged whenever no feverish feeling is reported during a regular nurse visit. Nevertheless, this discussion shows that it is important to be aware that even our weakened, more realistic assumption of conditional independence is often not fully satisfied in practical situations. Hence, one must be careful when verifying the assumptions and potentially adapt the experimental setting such that they are satisfied.

### 4.2 Setting with Dependence

To allow for dependence, we only consider the probability measure $\mathbb{P}$ and define $X, n, K, t_i, \tau, M, \mathcal{A}_t$ similar as before, but all on the same (filtered) probability space $(\Omega, \mathcal{F}, \mathbb{F}, \mathbb{P})$ associated with $\mathbb{P}$. For a random variable $Z$ and a family of sets $\mathcal{B}$ we use the natural notation for their smallest jointly generated sigma algebra $\boldsymbol{\sigma}(Z, \mathcal{B}) := \boldsymbol{\sigma}(Z) \vee \boldsymbol{\sigma}(\mathcal{B})$.

Then we need the following assumptions, most of which are borrowed from Assumption 2.1. Only (vii) and (viii) are new, while (i) is a strict generalisation of Assumption 2.1 (i).

**Assumption 4.1.** *We assume that Assumption 2.1 items (ii) to (vi) hold, with all instances of $\tilde{\mathbb{P}}$ and $\mathbb{P} \times \tilde{\mathbb{P}}$ replaced by $\mathbb{P}$. Additionally, we assume that:*

(i) *$M_{0,j} = 1$ for all $1 \le j \le d_X$ ($X$ is completely observed at 0) and $|M_k|_1 > 0$ for every $1 \le k \le K$ $\mathbb{P}$-almost surely (at every observation time at least one coordinate is observed).*

(vii) *For every $1 \le k \le n$, $X_{t_k-}$ is conditionally independent of $\boldsymbol{\sigma}(n, M_{t_k})$ given $\mathcal{A}_{t_k-}$.*

*(viii) For all $1 \leq k \leq K$, $1 \leq j \leq d_X$ there is some $\eta_{k,j} > 0$ such that $\mathbb{P}(M_{k,j} = 1 \mid \boldsymbol{\sigma}(n, \mathcal{A}_{t_k-})) > \eta_{k,j}$ (i.e., given the currently known information and $n$, for each coordinate the probability of observing it at the next observation time is positive).*

*(ix) For every $1 \leq k \leq K$ the process $X$ is conditionally independent of $t_k$ given $\mathcal{A}_{t_{k-1}}$.*

**Remark 4.2.** *The relaxations on the assumption of observing $X_0$ completely discussed in Krach et al. (2022, Remark 2.2) can equivalently be applied here.*

We can use the original objective function (4) and its Monte Carlo approximation (5).

### 4.3 Convergence Theorem with Dependence

In the setting defined in Section 4.2, Theorem 2.7 holds equivalently.

**Theorem 4.3.** *If Assumption 4.1 is satisfied and using the definitions of Section 4.2, the claims of Theorem 2.7 hold equivalently. In particular, we obtain convergence of our estimator $Y^{\theta_{m,N_m}^{\min}}$ to the true conditional expectation $\hat{X}$ in $d_k$.*

The main technical challenge when generalising Theorem 2.7 to Assumption 4.1 is to replace the arguments using independence and e.g. Fubini's theorem, by arguments using conditional independence. This difficulty is well visible when comparing the proof of the orthogonal projection Lemma 4.6 below, with the original proof of Krach et al. (2022, Lemma 4.2). Before, the claim followed relatively immediately from Fubini's theorem and the standard orthogonal projection result for conditional expectations of random variables. In contrast to this, now we have to condition first on the known information and the additional random variables of the observation framework, then we have to argue by conditional independence that this is the same as conditioning only on the known information and only after that we can apply the conditional version Proposition 4.4 of the orthogonal projection result. The relevance of Lemma 4.6 as the backbone of the proof of Theorem 4.3 was already discussed in Section 3.2. Finally, in Lemma 4.8 we show how Assumption 4.1(viii) can replace independence between $M$ and $X$, when deriving that the distance between $\hat{X}_{t_k-}$ and $Y_{t_k-}$ can be bounded through their distance when multiplied with the observation mask.

First, we prove the extension of the standard $L^2$-orthogonality result that was stated in Herrera et al. (2021, Proposition B.2).

**Proposition 4.4.** *Let $(\Omega, \mathcal{F}, \mathbb{P})$ be a fixed probability space, and $\mathcal{A}, \mathcal{B}$ be sub-$\sigma$-algebras such that $\mathcal{B} \subseteq \mathcal{A} \subseteq \mathcal{F}$. For some random variable $X \in L^2(\Omega, \mathcal{F}, \mathbb{P})$ we define $\hat{X} := \mathbb{E}[X \mid \mathcal{A}]$. Then for every random variable $Z \in L^2(\Omega, \mathcal{A}, \mathbb{P})$ with $\mathbb{P}(Z \neq \hat{X}) > 0$ we have*

$$\mathbb{E}\left[ |X - Z|_2^2 \,\Big|\, \mathcal{B} \right] = \mathbb{E}\left[ \left| X - \hat{X} \right|_2^2 \,\Big|\, \mathcal{B} \right] + \mathbb{E}\left[ \left| Z - \hat{X} \right|_2^2 \,\Big|\, \mathcal{B} \right] \geq \mathbb{E}\left[ \left| X - \hat{X} \right|_2^2 \,\Big|\, \mathcal{B} \right],$$

*with strict inequality with positive probability.*

The proof is based on Durrett (2010, Theorem 5.1.8). We focus on the one-dimensional case, though this can easily be generalised to multiple dimensions via the 2-norm, as in Herrera et al. (2021).

*Proof.* We begin by expanding the left hand side.

$$\mathbb{E}\left[ |X - Z|^2 \,\Big|\, \mathcal{B} \right] = \mathbb{E}\left[ \left| (X - \hat{X}) - (Z - \hat{X}) \right|^2 \,\Big|\, \mathcal{B} \right]$$

$$= \mathbb{E}\left[ \left| X - \hat{X} \right|^2 \,\Big|\, \mathcal{B} \right] + \mathbb{E}\left[ \left| Z - \hat{X} \right|^2 \,\Big|\, \mathcal{B} \right] - 2\mathbb{E}\left[ (X - \hat{X})(Z - \hat{X}) \,\Big|\, \mathcal{B} \right]$$

We now analyse the cross term, which expands to $\mathbb{E}[Z(X - \hat{X}) \mid \mathcal{B}] - \mathbb{E}[\hat{X}(X - \hat{X}) \mid \mathcal{B}]$. Focusing on the first term, we note that since $Z \in L^2(\Omega, \mathcal{A}, \mathbb{P})$, it holds that $Z\mathbb{E}[X \mid \mathcal{A}] = \mathbb{E}[ZX \mid \mathcal{A}]$. By taking expectation (conditioned on $\mathcal{B}$) of both sides, we get

$$\mathbb{E}\left[ Z\hat{X} \,\Big|\, \mathcal{B} \right] = \mathbb{E}\left[ Z\mathbb{E}\left[ X \mid \mathcal{A} \right] \,\Big|\, \mathcal{B} \right] = \mathbb{E}\left[ \mathbb{E}\left[ ZX \mid \mathcal{A} \right] \,\Big|\, \mathcal{B} \right] = \mathbb{E}\left[ ZX \mid \mathcal{B} \right]$$

via the tower property, as $\mathcal{B} \subseteq \mathcal{A}$. Hence, $\mathbb{E}[Z(X - \hat{X}) \,|\, \mathcal{B}] = 0$. Note that showing this only required that $Z$ is $\mathcal{A}$-measurable. Since this is also satisfied by $\hat{X}$ we directly have $\mathbb{E}[\hat{X}(X - \hat{X}) \,|\, \mathcal{B}] = 0$ and therefore the cross term vanishes and the equality follows. The inequality holds since $\mathbb{E}[|Z - \hat{X}|_2^2 \,|\, \mathcal{B}]$ is non-negative. We therefore just need to show that the inequality is strict with positive probability. To this end, assume for the sake of contradiction that $\mathbb{E}[|Z - \hat{X}|_2^2 \,|\, \mathcal{B}] = 0$ $\mathbb{P}$-a.s, which implies (by the tower property) that $\mathbb{E}[|Z - \hat{X}|_2^2] = 0$. This is only possible if $Z = \hat{X}$ $\mathbb{P}$-a.s., which contradicts the assumption that $\mathbb{P}(Z \neq \hat{X}) > 0$. $\qquad\square$

As a next step we show that under Assumption 4.1(ix) we can recover that $\hat{X}$ is the optimal predictor.

**Proposition 4.5.** *Under Assumption 4.1(ix) we have*

$$\hat{X}_{t_k-} = \mathbb{E}\left[X_t \,\middle|\, \mathcal{A}_{t_{k-1}}\right]\Big|_{t=t_k-} = \mathbb{E}\left[X_{t_k-} \,\middle|\, \mathcal{A}_{t_k-}\right].$$

*Proof.* The result is a direct consequence of Proposition A.9. $\qquad\square$

Next we re-prove Krach et al. (2022, Lemma 4.2) under the relaxed assumptions with the help of Proposition 4.5.

**Lemma 4.6.** *For any $\mathbb{A}$-adapted process $Z \in L^2(\Omega, \mathbb{A}, \mathbb{P})$ it holds that*

$$\mathbb{E}\left[\frac{1}{n}\sum_{i=1}^n |M_{t_i} \odot (X_{t_i} - Z_{t_i-})|_2^2\right] = \mathbb{E}\left[\frac{1}{n}\sum_{i=1}^n \left|M_{t_i} \odot (X_{t_i} - \hat{X}_{t_i-})\right|_2^2\right] + \mathbb{E}\left[\frac{1}{n}\sum_{i=1}^n \left|M_{t_i} \odot (\hat{X}_{t_i-} - Z_{t_i-})\right|_2^2\right].$$

*Proof.*

$$
\begin{aligned}
\mathbb{E}\left[\frac{1}{n}\sum_{i=1}^n |M_{t_i} \odot (X_{t_i-} - Z_{t_i-})|_2^2\right] &= \sum_{i=1}^K \mathbb{E}\left[\frac{1}{n}\mathbb{1}_{\{i \leq n\}} |M_{t_i} \odot (X_{t_i-} - Z_{t_i-})|_2^2\right] \\
&= \sum_{i=1}^K \mathbb{E}\left[\mathbb{E}\left[\frac{1}{n}\mathbb{1}_{\{i \leq n\}} |M_{t_i} \odot (X_{t_i-} - Z_{t_i-})|_2^2 \,\middle|\, \boldsymbol{\sigma}(n, M_{t_i}, \mathcal{A}_{t_i-})\right]\right] \\
&= \sum_{i=1}^K \mathbb{E}\left[\frac{1}{n}\mathbb{1}_{\{i \leq n\}} \sum_{j=1}^{d_X} M_{t_i,j} \mathbb{E}\left[|X_{t_i-,j} - Z_{t_i-,j}|^2 \,\middle|\, \boldsymbol{\sigma}(n, M_{t_i}, \mathcal{A}_{t_i-})\right]\right] \\
&= \sum_{i=1}^K \mathbb{E}\left[\frac{1}{n}\mathbb{1}_{\{i \leq n\}} \sum_{j=1}^{d_X} M_{t_i,j} \mathbb{E}\left[|X_{t_i-,j} - Z_{t_i-,j}|^2 \,\middle|\, \mathcal{A}_{t_i-}\right]\right].
\end{aligned}
$$

The first step follows by monotone convergence, the last by Lemma 4.7 below. Now we can conclude by first applying the equality from Proposition 4.4 with $\mathcal{A}, \mathcal{B} = \mathcal{A}_{t_i-}$, which yields that $\mathbb{E}[X_{t_i-} \,|\, \mathcal{A}_{t_i-}]$ minimises the expression, and then using Proposition 4.5 to conclude that $\hat{X}_{t_i-}$ equals this minimizer. Finally, we can reverse the above steps to arrive at the desired form. $\qquad\square$

**Lemma 4.7.** *Assume the context of Lemma 4.6. Then for all $i, j$ it holds that*

$$\mathbb{E}\left[|X_{t_i-,j} - Z_{t_i-,j}|^2 \,\middle|\, \boldsymbol{\sigma}(n, M_{t_i}, \mathcal{A}_{t_i-})\right] = \mathbb{E}\left[|X_{t_i-,j} - Z_{t_i-,j}|^2 \,\middle|\, \mathcal{A}_{t_i-}\right].$$

*Proof.* We prove this by showing that if we expand the $|X_{t_i-,j} - Z_{t_i-,j}|^2$ term, all three resulting terms can just be conditioned on $\mathcal{A}_{t_i-}$. This is a valid argument as $X$ and $Z$ are both assumed to be square-integrable. Note that squaring a random variable plays no role in the information given by the $\sigma$-algebra it is being conditioned on, and so we only need to analyse the terms $Z_{t_i-,j}$, $X_{t_i-,j}$, and $X_{t_i-,j}Z_{t_i-,j}$. See Appendix A for an overview of conditional independence and how it's used here.

CASE $Z_{t_i-,j}$: $Z$ is $\mathbb{A}$-adapted, and so $Z_{t_i-,j}$ is $\mathcal{A}_{t_i-}$-measurable. Thus we have

$$\mathbb{E}\left[Z_{t_i-,j} \,|\, \boldsymbol{\sigma}(n, M_{t_i}, \mathcal{A}_{t_i-})\right] = Z_{t_i-,j} = \mathbb{E}\left[Z_{t_i-,j} \,|\, \mathcal{A}_{t_i-}\right]$$

as desired.

CASE $X_{t_i-,j}$: Assumption 4.1 point (vii) implies that $X_{t_i-,j}$ is conditionally independent of $\boldsymbol{\sigma}(n, M_{t_i})$ given $\mathcal{A}_{t_i-}$. We therefore have by Proposition A.4

$$\mathbb{E}\left[X_{t_i-,j} \mid \boldsymbol{\sigma}(n, M_{t_i}, \mathcal{A}_{t_i-})\right] = \mathbb{E}\left[X_{t_i-,j} \mid \mathcal{A}_{t_i-}\right].$$

CASE $X_{t_i-,j}Z_{t_i-,j}$: We combine the previous two ideas, namely that $Z_{t_i-,j}$ is $\mathcal{A}_{t_i-}$-measurable and that $X_{t_i-,j}$ is conditionally independent of $\boldsymbol{\sigma}(n, M_{t_i})$ given $\mathcal{A}_{t_i-}$. Thus

$$\mathbb{E}\left[X_{t_i-,j}Z_{t_i-,j} \mid \boldsymbol{\sigma}(n, M_{t_i}, \mathcal{A}_{t_i-})\right] = Z_{t_i-,j}\mathbb{E}\left[X_{t_i-,j} \mid \boldsymbol{\sigma}(n, M_{t_i}, \mathcal{A}_{t_i-})\right]$$
$$= Z_{t_i-,j}\mathbb{E}\left[X_{t_i-,j} \mid \mathcal{A}_{t_i-}\right] = \mathbb{E}\left[Z_{t_i-,j}X_{t_i-,j} \mid \mathcal{A}_{t_i-}\right].$$

Combining these 3 cases proves the claim. $\qquad\square$

The following lemma shows how Assumption 4.1(viii) can replace the independence between $M$ and $X$.

**Lemma 4.8.** *There exists some $c_2(k) > 0$ such that for any $\mathbb{A}$-adapted process $Z \in L^2(\Omega, \mathbb{A}, \mathbb{P})$ we have*

$$\mathbb{E}\left[\mathbb{1}_{\{n \geq k\}}\left|\hat{X}_{t_k-} - Z_{t_k-}\right|_1\right] \leq \frac{1}{c_2(k)}\mathbb{E}\left[\mathbb{1}_{\{n \geq k\}}\left|M_{t_k} \odot (\hat{X}_{t_k-} - Z_{t_k-})\right|_1\right].$$

*Proof.* Assumption 4.1 point (viii) states that $0 < \eta_{k,j} < \mathbb{P}(M_{k,j} = 1 \mid \sigma(n, A_{t_k-})) = \mathbb{E}[M_{t_k,j} \mid \boldsymbol{\sigma}(n, \mathcal{A}_{t_k-})]$ for all $k, j$. Let $c_2 := c_2(k) := \min_{1 \leq j \leq d_X} \eta_{k,j}$, then $c_2 > 0$ and

$$\mathbb{E}\left[\mathbb{1}_{\{n \geq k\}}\left|M_{t_k} \odot (\hat{X}_{t_k-} - Z_{t_k-})\right|_1\right] = \sum_{j=1}^{d_X}\mathbb{E}\left[\mathbb{1}_{\{n \geq k\}}M_{t_k,j}\left|\hat{X}_{t_k-,j} - Z_{t_k-,j}\right|\right]$$

$$= \sum_{j=1}^{d_X}\mathbb{E}\left[\mathbb{1}_{\{n \geq k\}}\left|\hat{X}_{t_k-,j} - Z_{t_k-,j}\right|\mathbb{E}\left[M_{t_k,j} \mid \boldsymbol{\sigma}(n, \mathcal{A}_{t_k-})\right]\right]$$

$$\geq c_2 \mathbb{E}\left[\mathbb{1}_{\{n \geq k\}}\left|\hat{X}_{t_k-} - Z_{t_k-}\right|_1\right],$$

where we used that $\mathbb{1}_{\{n \geq k\}}|\hat{X}_{t_k-,j} - Z_{t_k-,j}|$ is $\boldsymbol{\sigma}(n, \mathcal{A}_{t_k-})$-measurable in the second line and the definition of $c_2$ in the last line. $\qquad\square$

With the help of these lemmas we are now ready to prove Theorem 4.3. In the following we again give a sketch of the proof, by only outlining those parts of it that need to be changed in comparison with the original proof of Theorem 2.7 in Krach et al. (2022). We refer the interested reader to the full proof in Appendix C for all details.

*Sketch of Proof of Theorem 4.3.* As before it follows that $\hat{X}$ is a minimizer of $\Psi$. To show its uniqueness, we first note that

$$\mathbb{E}\left[\mathbb{1}_{\{n \geq k\}}\left|\hat{X}_{t_k-} - Z_{t_k-}\right|_2\right] \leq \frac{c_3}{c_2(k)}\mathbb{E}\left[\mathbb{1}_{\{n \geq k\}}\left|M_{t_k} \odot (\hat{X}_{t_k-} - Z_{t_k-})\right|_2\right] \tag{12}$$

is implied as before when using Lemma 4.8 instead of the independence. Then

$$\mathbb{E}\left[\frac{1}{n}\sum_{i=1}^{n}\left|M_{t_i} \odot (\hat{X}_{t_i-} - Z_{t_i-})\right|_2^2\right] \geq \left(\frac{c_2(k)}{c_0 c_1 c_3}\right)^2 d_k(\hat{X}, Z)^2 > 0,$$

follows as before, implying the uniqueness of $\hat{X}$ as minimizer of $\Psi$.

The approximation of the functions $f_j$, $F_j$ also works as before. With this we have

$$\min_{Z \in \mathbb{D}}\Psi(Z) \leq \Phi(\theta_m^{\min}) \leq \Phi(\theta_m^*)$$

$$
\begin{aligned}
&= \mathbb{E}\left[\frac{1}{n}\sum_{i=1}^{n}\left(\left|M_{t_i}\odot(X_{t_i}-Y_{t_i}^{\theta_m^*})\right|_2+\left|M_{t_i}\odot(Y_{t_i}^{\theta_m^*}-Y_{t_i-}^{\theta_m^*})\right|_2\right)^2\right]\\
&\le \mathbb{E}\left[\frac{1}{n}\sum_{i=1}^{n}\left(\left|M_{t_i}\odot(\hat{X}_{t_i}-Y_{t_i}^{\theta_m^*})\right|_2+\left|M_{t_i}\odot(Y_{t_i}^{\theta_m^*}-\hat{X}_{t_i})\right|_2\right.\right.\\
&\qquad\qquad\left.\left.+\left|M_{t_i}\odot(\hat{X}_{t_i}-\hat{X}_{t_i-})\right|_2+\left|M_{t_i}\odot(\hat{X}_{t_i-}-Y_{t_i-}^{\theta_m^*})\right|_2\right)^2\right]\\
&\le \mathbb{E}\left[\frac{1}{n}\sum_{i=1}^{n}\left(\left|\hat{X}_{t_i}-Y_{t_i}^{\theta_m^*}\right|_2+\left|Y_{t_i}^{\theta_m^*}-\hat{X}_{t_i}\right|_2\right.\right.\\
&\qquad\qquad\left.\left.+\left|M_{t_i}\odot(\hat{X}_{t_i}-\hat{X}_{t_i-})\right|_2+\left|\hat{X}_{t_i-}-Y_{t_i-}^{\theta_m^*}\right|_2\right)^2\right]\\
&\le \mathbb{E}\left[\frac{1}{n}\sum_{i=1}^{n}\left(\left|M_{t_i}\odot(X_{t_i}-\hat{X}_{t_i-})\right|_2+3c_m\right)^2\right]\\
&= \Psi(\hat{X})+\mathbb{E}\left[\frac{1}{n}\sum_{i=1}^{n}\left(6c_m\left|M_{t_i}\odot(X_{t_i}-\hat{X}_{t_i-})\right|_2+9c_m^2\right)\right]\\
&= \Psi(\hat{X})+\mathbb{E}\left[\frac{1}{n}\sum_{i=1}^{n}6c_m\left|M_{t_i}\odot(X_{t_i}-\hat{X}_{t_i-})\right|_2\right]+9\mathbb{E}\left[c_m^2\right]\\
&\le \Psi(\hat{X})+6\mathbb{E}\left[\frac{1}{n}\sum_{i=1}^{n}c_m\left|X_{t_i}-\hat{X}_{t_i-}\right|_2\right]+9\mathbb{E}\left[c_m^2\right]\\
&\le \Psi(\hat{X})+6\mathbb{E}\left[\frac{1}{n}\sum_{i=1}^{n}c_m\cdot 2TB(X_T^\star+1)^p\right]+9\mathbb{E}\left[c_m^2\right]\\
&= \Psi(\hat{X})+12TB\mathbb{E}\left[c_m(X_T^\star+1)^p\right]+9\mathbb{E}\left[c_m^2\right],
\end{aligned}
$$

where the triangle inequality was used in the third line, and we use Assumption 4.1 point (iv) (which implies that $|\hat{X}_t|\le TB(X^\star+1)^p$ for all $t$) to construct a crude bound in the second to last line. As in Krach et al. (2022), dominated convergence can be used to show

$$
\min_{Z\in\mathbb{D}}\Psi(Z)=\Psi(\hat{X})\le\Phi(\theta_m^{\min})\le\Phi(\theta_m^*)\xrightarrow{m\to\infty}\min_{Z\in\mathbb{D}}\Psi(Z),
$$

since $2p$ is again the largest power of $X_T^\star$ in both terms (remember $c_m$ has an $(X_T^\star)^p$ term). The convergence of $d_k(\hat{X},Y^{\theta_m^\star})\to 0$ follows as before, finishing the first part of the theorem. Finally, it is easy to see that the proof of the convergence of the Monte Carlo approximation is not affected by our more general dependence assumptions such that also the second part of the theorem follows. □

The assumption that $X_{t_i-}$ and $n$ are conditionally independent given $\mathcal{A}_{t_i-}$ seems a bit odd, because $n$ is a quantity that is not known at $t_i$, i.e., not $\mathcal{A}_{t_i}$-measurable. This assumption is needed, because we weight the observations in the loss by $1/n$, which means that we weight the terms known at time $t_i$ with a quantity that is in general only known at final time $T$. In line with this, we can drop the assumption if instead we weight the terms with the time step $\Delta t_i:=t_i-t_{i-1}$, which is $\mathcal{A}_{t_i-}$-measurable. However, this needs an additional integrability assumption as explained in the following proposition.

**Proposition 4.9.** *If we change the weighting in the loss function to*

$$
\Psi(Z):=\mathbb{E}\left[\sum_{i=1}^{n}\Delta t_i\left(|M_i\odot(X_{t_i}-Z_{t_i})|_2+|M_i\odot(X_{t_i}-Z_{t_i-})|_2\right)^2\right],
$$

*then Assumption 4.1 (vii) can be replaced by the weaker assumption (vii'-a) together with the additional integrability assumption (vii'-b) below.*

*(vii'-a) For every $1 \leq i \leq n$, $X_{t_i-}$ is conditionally independent of $\sigma(M_{t_i})$ given $\mathcal{A}_{t_i-}$.*

*(vii'-b) For every $1 \leq i \leq K$, $\mathbb{E}\left[\frac{1}{\Delta t_i}\right] < \infty$.*

*Proof.* First, we note that $\{i \leq n\} = \{t_i < \infty\}$ and $\Delta t_i$ are $\mathcal{A}_{t_i-}$-measurable, hence the proofs of Lemma 4.6 and Lemma 4.8 work equivalently under the weaker assumption (vii'-a). Next we note that (23) has to be replaced in the analysis by

$$\mathbb{E}\left[|Z|_2\right] = \mathbb{E}\left[\frac{\sqrt{\Delta t_i}}{\sqrt{\Delta t_i}}\,|Z|_2\right] \leq c_1' \,\mathbb{E}\left[\Delta t_i\,|Z|_2^2\right]^{1/2} \tag{13}$$

where $c_1' := c_1'(i) := \mathbb{E}[1/\Delta t_i] < \infty$ holds by the additional assumption (vii'-b). Finally, we note that $\sum_{i=1}^n \Delta t_i \leq T$, hence integrability of the loss is still satisfied. These are all needed changes in the proof. □

**Remark 4.10.** *The different weightings in the loss function by either $1/n$ or $\Delta t_i$ put different importance on the observations. In the former case, every observation of one sample path is given equal importance, while in the latter case, observations have more importance if the previous observation is farther away. Although both loss functions yield the same unique optimizer, the choice of weighting can influence the training of the model for finite $m$ or $N$. In particular, the latter choice corresponds to an inductive bias, which makes it more important for the model to predict well after a long time without observations, while it is less important to predict well whenever the frequency of observations is high. Of course other weightings, e.g. by $1/\mathbb{E}[n]$, are possible, too, and reasonable in some situations.*

### 4.4 Examples of Processes Satisfying the Assumptions

Clearly, all the examples from Krach et al. (2022) are trivially valid for our generalised settings of Section 4.2 since points (vii) & (viii) of Assumption 4.1 are implied by independence. Furthermore, we provide a relatively general class of examples in Section 4.4.1 and extend two of the examples from Krach et al. (2022) to a setting where independence does not hold in Example 4.11 and Example 4.13.

#### 4.4.1 Class of Examples Incorporating Dependence

A main problem when constructing examples where the observation times $t_i$ have a dependence on the process $X$, is that in general this will also lead to $n$ having a dependence on $X$ (since $n$, as the amount of observations, increases when observations become more likely). This, in turn, might lead to a contradiction of Assumption 4.1 point (vii).

One way to circumvent this is to define $n$ conditionally independent of $X_{t_i-}$ given $\mathcal{A}_{t_i-}$ for all $i$ and to allow for pseudo observation times, i.e., observation times at which no coordinate is observed (cf. Remark 2.2). Then we can control whether the process $X$ is observed at an observation time $t_i$ via the observation mask $M_i$. In this way, the times at which $X$ is actually observed can depend on $X$ through the observation mask $M_k$, even though the $t_k$ do not depend on $X$. Therefore, the original problem is replaced by having an observation mask depending on $X$, which will be discussed in detail below.

When using a time grid on which the process $X$ is sampled, one concrete example of defining $n$ conditionally independent of $X_{t_i-}$ given $\mathcal{A}_{t_i-}$ is to use the grid points as observation times $t_i$ leading to $n$ being the number of grid points. In this case, both $n$ and the $t_i$ are deterministic, hence, they satisfy the conditional independence assumptions.

We need to ensure that Assumption 4.1 points (vii) and (viii) are satisfied (assuming that a dummy variable is observed at every observation time), when defining the observation mask. One way to define $M_i$ such that point (vii) is satisfied is to define it as a function of random variables that are $\mathcal{A}_{t_i-}$-measurable and random variables that are independent of $\sigma(\mathcal{A}_{t_i-}, X_{t_i-}, n)$.

In particular, let $M_i := g_i\left((U_j^i)_{j \in J_1^i}, (V_j^i)_{j \in J_2^i}\right)$, where $g_i$ is a measurable function mapping to $\{0,1\}^{d_X}$; $J_1^i, J_2^i \subseteq \mathbb{N}$ are index sets; $U_j^i$ is a $\mathcal{A}_{t_i-}$-measurable random variable; and $V_j^i$ a random variable independent of $\sigma(\mathcal{A}_{t_i-}, X_{t_i-}, n)$ for all $j$ in $J_1^i$ and $J_2^i$ respectively.

By Proposition A.5 we need to show that for any measurable function $\phi$ we have $\mathbb{E}[\phi(X_{t_i-}) \,|\, \boldsymbol{\sigma}(n, M_i, \mathcal{A}_{t_i-})] = \mathbb{E}[\phi(X_{t_i-}) \,|\, \mathcal{A}_{t_i-}]$. Indeed, $M_i$ is $\boldsymbol{\sigma}(\mathcal{A}_{t_i-}, (V_j^i)_{j \in J_2^i})$-measurable by construction, therefore, we have for such a $\phi$ that

$$\mathbb{E}\left[\phi(X_{t_i-}) \mid \boldsymbol{\sigma}(n, M_i, \mathcal{A}_{t_i-})\right] = \mathbb{E}\left[\mathbb{E}\left[\phi(X_{t_i-}) \,\Big|\, \boldsymbol{\sigma}(n, (V_j^i)_{j \in J_2^i}, \mathcal{A}_{t_i-})\right] \,\Big|\, \boldsymbol{\sigma}(n, M_i, \mathcal{A}_{t_i-})\right],$$

by the tower property. On the other hand,

$$\mathbb{E}\left[\phi(X_{t_i-}) \,\Big|\, \boldsymbol{\sigma}(n, (V_j^i)_{j \in J_2^i}, \mathcal{A}_{t_i-})\right] = \mathbb{E}\left[\phi(X_{t_i-}) \mid \boldsymbol{\sigma}(n, \mathcal{A}_{t_i-})\right] = \mathbb{E}\left[\phi(X_{t_i-}) \mid \mathcal{A}_{t_i-}\right],$$

using the independence of $V_j^i$ together with Corollary A.2 in the first equality and $n$ being conditionally independent together with Proposition A.5 in the second equality. Together, this implies the claim and therefore Assumption 4.1 point (vii).

For Assumption 4.1 point (viii), we note that by Durrett (2010, Lemma 6.2.1) we have

$$\begin{aligned}
\mathbb{P}(M_k = 1 \mid \boldsymbol{\sigma}(n, A_{t_k-})) &= \mathbb{E}\left[M_{t_k} \mid \boldsymbol{\sigma}(n, \mathcal{A}_{t_k-})\right] \\
&= \mathbb{E}\left[g_k\left((U_j^k)_{j \in J_1^k}, (V_j^k)_{j \in J_2^k}\right) \,\Big|\, \boldsymbol{\sigma}(n, \mathcal{A}_{t_k-})\right] \\
&= \tilde{g}_k((U_j^k)_{j \in J_1^k}),
\end{aligned}$$

for $\tilde{g}_k((u_j^k)_{j \in J_1^k}) := \mathbb{E}\left[g_k\left((u_j^k)_{j \in J_1^k}, (V_j^k)_{j \in J_2^k}\right)\right]$. Hence, we need to define the $g_k$ and $V_j^k$ such that $\tilde{g}_k > \eta_k$ (coordinate-wise) for some $\eta_k > 0$.

**Example 4.11** (Homogeneous Poisson Point Process with Dependent Observations). *We use a 1-dimensional homogeneous Poisson point process $X = N^\lambda$ (Krach et al., 2022, Section 7.3) and defined observations depending on its value following the instructions above. To begin with, we define the observation times to be the grid points of the sampling grid of the process and $n$ accordingly to be the number of grid points. To permit observation times of the process depending on the process, we define the observation mask as*

$$M_i = \begin{cases} \mathbb{1}_{\{x_i \geq \lambda t_{i-1}\}}, \ x_i \sim \mathcal{N}(N_{t_{i-1}}^\lambda, \sigma^2) & \text{if } M_{i-1}^X = 1, \\ u_i \sim \text{Bernoulli}(p) & \text{if } M_{i-1}^X = 0. \end{cases}$$

*for some $\sigma > 0$ and $p \in (0,1)$. Thus the process is more likely to be observed if the previous value was observed and was large (note $\mathbb{E}[N_{t_k}^\lambda] = \lambda t_k$ for all $k$). Otherwise the mask value is sampled from a Bernoulli distribution.*

*To satisfy Assumption 4.1 point (i), we define $M_0 := 1$. To see that Assumption 4.1 point (vii) holds, let $V_1^i \sim \text{Bernoulli}(p)$ and $V_2^i \sim \mathcal{N}(0, \sigma^2)$ be independent random variables independent of $\boldsymbol{\sigma}(\mathcal{A}_{t_i-}, N_{t_i-}^\lambda)$. Then*

$$M_i = \mathbb{1}_{\{M_{i-1}=1\}} \mathbb{1}_{\{N_{t_{i-1}}^\lambda + V_2^i \geq \lambda t_{i-1}\}} + V_1^i \mathbb{1}_{\{M_{i-1}=0\}} =: g_i(M_{i-1}, N_{t_{i-1}}^\lambda, t_{i-1}, V_1^i, V_2^i),$$

*and the claim follows as explained above. Moreover, Assumption 4.1 point (viii) is satisfied because*

$$\begin{aligned}
\tilde{g}_i(M_{i-1}, N_{t_{i-1}}^\lambda, t_{i-1}) &= \mathbb{1}_{\{M_{i-1}=1\}} \mathbb{E}\left[\mathbb{1}_{\{a+V_2^i \geq \lambda b\}}\right]\Big|_{(a,b)=\left(N_{t_{i-1}}^\lambda, t_{i-1}\right)} + \mathbb{1}_{\{M_{i-1}=0\}} \mathbb{E}\left[V_1^i\right] \\
&\geq \min\left(\mathbb{P}\left[V_2^i \geq \lambda b - a\right]\Big|_{(a,b)=\left(N_{t_{i-1}}^\lambda, t_{i-1}\right)}, p\right) \\
&\geq \min\left(\mathbb{P}\left[V_2^i \geq \lambda T\right], p\right) =: \eta_i > 0,
\end{aligned}$$

*using that $\lambda t_{i-1} - N_{t_{i-1}}^\lambda \leq \lambda T$, since $N^\lambda \geq 0$, $t_{i-1} \leq T$ and $\lambda > 0$, in the last line.*

**Remark 4.12.** *We note that the choice $X = N^\lambda$ is only explicitly used for the fact that $N^\lambda$ is lower bounded (by 0). Hence, the Poisson point process could be replaced by any other process that is lower bounded and satisfies Assumption 2.1 points (iv) & (v).*

**Example 4.13** (Black–Scholes with Dependent Observations). *We use a 1-dimensional Black–Scholes process (geometric Brownian motion) [(Krach et al., 2022, Example 7.3)](#) with constant drift and volatility $\mu, \sigma$ starting at $x_0$ and again define observations depending on its value. The observation times and $n$ are defined as in the previous example and we define the actual times when $X$ is observed via the mask $M$. We set $M_0 = 1$. Moreover, we redefine $\tau$ to be the last time before $t$ at which $X$ was observed, i.e., $\tau(t) = \max\{t_i \,|\, t_i < t, M_i = 1\}$. Let $V_1^i \sim \text{Bernoulli}\left(\frac{t_i - t_{i-1}}{t_i - \tau(t_i)}\right)$, $V_2^i \sim \mathcal{N}(0, \eta^2)$ and $V_3^i \sim \text{Bernoulli}(p)$ be independent random variables for some $\eta > 0$ and $p \in (0, 1)$. Then we define*

$$M_i := V_1^i \, \mathbb{1}_{\{X_{\tau(t_i)} + V_2^i \geq x_0 e^{\mu t_i}\}} + (1 - V_1^i) \, V_3^i,$$

*for all $i \geq 1$. In particular, if $X$ was observed at the previous observation time $V_1^i = 1$ and the probability of observing $X$ increases with the size of the last observation of $X$ (compared to the current expected value of $X$ at the current time). The further the last observation of $X$ is in the past, the more likely $V_1^i = 0$ in which case $X$ is observed with probability $p$. Upon noting that the $t_i$ are deterministic, it follows as in the previous example that Assumption 4.1 points (vii) and (viii) are satisfied.*

## 5 Practical Implications of the Convergence Result

In this section we discuss which practically relevant conclusions we can draw from convergence in the metrics $d_k$ for $1 \leq k \leq K$. We mainly focus on the setting in Section 4.2, however, the same is also true in the combined setting of Section C.1. In Section 5.1, we study the practical meaning of Assumption 4.1(ix). We give an intuitive counterexample, where this assumption is not satisfied and our model does not converge to $\hat{X}$. Secondly, in Section 5.2, we discuss the implications of convergence in the (pseudo) metrics $d_k$ (which ensures a good approximation at left-limits of observation times) for general times $t$. In particular, we study two practically relevant examples for the conditional distribution of the observation times and show that in these cases, with high probability, our model approximates the conditional expectation well on the entire support of the observation times.

### 5.1 Practical meaning of Assumption 4.1(ix)

If Assumption 4.1(ix) is not satisfied, this can lead to situations where $\hat{X}$ is not the minimizer of $\Psi$ and therefore our model does not converge to it, as we explain in the following example. Assume that patients in a hospital get asked by a nurse every morning at 8 am, whether they feel feverish. If the answer is yes, then their temperature is measured right away and logged at 8 am; if the answer is no, their temperature is measured and logged at 4 pm. The (non-)feverish feeling is not logged at all. This data satisfies Assumption 4.1 (since $n$ is deterministic; $d_X = 1$ and $M_i = 1$) except for item (ix). It is clear that the model will predict higher temperatures in the morning than in the afternoon for any test sample, even if the patient always has the same temperature, since the model only saw this behaviour in the training data. If we assume that on average patients have the same (or even lower) temperature in the morning as in the afternoon, this prediction is not optimal in practice.

The mathematical reason for this discrepancy is that our model learns to approximate $\mathbb{E}[X_{t_k-} \,|\, \mathcal{A}_{t_k-}]$ (see proof of Lemma 4.6), which does not coincide with $\hat{X}$, i.e.,

$$\mathbb{E}\left[X_{t_k-} \,\big|\, \mathcal{A}_{t_k-}\right] \neq \mathbb{E}\left[X_t \,\big|\, \mathcal{A}_t\right]\Big|_{t=t_k-} = \mathbb{E}\left[X_t \,\big|\, \mathcal{A}_{t_{k-1}}\right]\Big|_{t=t_k-} = \hat{X}_{t_k-}$$

(surprisingly) holds for this example. Indeed, Proposition 4.5 (which would turn this inequality into an equality) heavily relies on Assumption 4.1(ix). $\mathbb{E}[X_{t_k-} \,|\, \mathcal{A}_{t_k-}]$ minimizes the test error (if the joint distribution of $(X, t_k)$ is the same as during training), since it can exploit the knowledge[7] that it is queried at an observation time $t_k$ (i.e., high temperature if $t_k$ is in the morning and low temperature if $t_k$ is in the afternoon). However, in this example we do *not* obtain the optimal forecast $\hat{X}_t = \mathbb{E}[X_t \,|\, \mathcal{A}_{t_{k-1}}]$ for any time $t$, which is the best prediction of $X$ at any time, given the information available at the previous observation time.

---

[7]Mathematically, $\mathcal{A}_{t_k-} = \boldsymbol{\sigma}(\mathcal{A}_{t_{k-1}}, t_k)$ includes the information that a measurement is taken at time $t_k$.

Assumption 4.1(ix) ensures (via Proposition 4.5) that our model converges to the practically meaningful version of the conditional expectation $\hat{X}_{t_k} = \mathbb{E}[X_t \,|\, \mathcal{A}_{t_{k-1}}]\big|_{t=t_k-}$ at observation times. In the following, we discuss what this implies for the approximation at any $t \in [0, T]$.

## 5.2 From Approximations at Observation Times to Approximations at any Time

Under Assumption 4.1 we have established that our model output $Y_t^{\theta_m^{\min}}(\tilde{X}^{\leq t_k-})$ converges to $\mathbb{E}[X_t \,|\, \mathcal{A}_{t_{k-1}}]$ at every observation time $t_k$. Both, $Y_t^{\theta_m^{\min}}(\tilde{X}^{\leq t_k-})$ and $\mathbb{E}[X_t \,|\, \mathcal{A}_{t_{k-1}}]$ are functions of the random variables summarized in $\mathcal{A}_{t_{k-1}}$ (by definition for our model and by the Doob-Dynkin lemma for the conditional expectation) and therefore conditionally independent of $t_k$ given $\mathcal{A}_{t_{k-1}}$. We note in particular that $\tilde{X}^{\leq t_k-}$ carries exactly the information of $\mathcal{A}_{t_{k-1}}$, i.e., it has no information about $t_k$ (despite its notation). Hence, intuitively speaking, the approximation has to be good for every $t$ in the support of $t_k$ for the expectation to converge. In the following we formalize this.

Let $\mathbb{P}_k$ be the probability measure conditioned on the event that $n \geq k$, i.e.

$$\mathbb{P}_k(\cdot) = \mathbb{P}(\cdot \,|\, \{n \geq k\}),$$

and denote by $\mathbb{E}_k$ the expectation operator with respect to this probability measure. Then the following lemma is a consequence of the definitions of $\mathbb{P}_k$ and $c_0(k)$.

**Lemma 5.1.** *We have $d_k(\hat{X}, Y^{\theta_m^{\min}}) = \mathbb{E}_k\left[\left|\hat{X}_{t_k-} - Y_{t_k-}^{\theta_m^{\min}}\right|_2\right]$ and the equivalent result for $Y^{\theta_{m,N_m}^{\min}}$.*

*Proof.* It is enough to note that $\mathbb{E}[\cdot \mathbb{1}_{\{n \geq k\}}] = \mathbb{P}(n \geq k) \mathbb{E}_k[\cdot]$. $\qquad\square$

Next, we define for each $1 \leq k \leq K$ the (regular) conditional distribution of $t_k$ given $\tilde{X}^{\leq t_k-}$ as the almost surely unique probability kernel (or random measure) $\mu_{t_k}(\cdot\,; \tilde{X}^{\leq t_k-})$ satisfying

$$\mu_{t_k}(B\,; \tilde{X}^{\leq t_k-}) = \mathbb{P}_k\left(t_k \in B \,\big|\, \mathcal{A}_{t_{k-1}}\right) \quad a.s., \quad \forall B \in \mathcal{B}([0, T]).$$

Kallenberg (2021, Theorem 8.5) ensures its existence (since $[0, T]$ is Borel) and implies for any measurable function $\phi$ that almost surely

$$\mathbb{E}_k\left[\phi\left(\tilde{X}^{\leq t_k-}, t_k\right) \,\big|\, \mathcal{A}_{t_{k-1}}\right] = \int_0^T \phi\left(\tilde{X}^{\leq t_k-}, t\right) \mu_{t_k}(dt\,; \tilde{X}^{\leq t_k-}). \tag{14}$$

In particular, this implies the following result.

**Proposition 5.2.** *Under Assumption 4.1 we have, with the same notation as in Theorem 2.7, that for every $1 \leq k \leq K$*

$$d_k(\hat{X}, Y^{\theta_m^{\min}}) = \mathbb{E}_k\left[\int_0^T \left|\mathbb{E}\left[X_{t-} \,\big|\, \tilde{X}^{\leq t_k-}\right] - Y_{t-}^{\theta_m^{\min}}\left(\tilde{X}^{\leq t_k-}\right)\right|_2 \mu_{t_k}(dt\,; \tilde{X}^{\leq t_k-})\right] \xrightarrow{m \to \infty} 0$$

*and the equivalent result for $Y^{\theta_{m,N_m}^{\min}}$.*

*Proof.* We first apply Lemma 5.1 and then use the tower property to get a nested conditional expectation (conditioning on $\mathcal{A}_{t_{k-1}}$) inside the outer expectation, which can be rewritten by (14) as an integral. Finally, we use Lemma 4.5 and then the Doob-Dynkin lemma (Taraldsen, 2018, Lemma 2) to rewrite $\mathbb{E}[X_t \,|\, \mathcal{A}_{t_{k-1}}]$ as a function of $\tilde{X}^{\leq t_k-}$, which we denote by $\mathbb{E}[X_t \,|\, \tilde{X}^{\leq t_k-}]$ (and we replace $t$ by $t-$ for the left-continuous version of it). $\qquad\square$

Proposition 5.2 implies convergence of a conditional $L^1$-norm (or equivalently, an $L^1$-norm integrated against a random measure) nested inside an $L^1$-norm. Since $L^p$-convergence implies convergence in probability, we have in particular that for every $\epsilon > 0$

$$\mathbb{P}_k\left(\int_0^T \left|\mathbb{E}\left[X_{t-} \,\big|\, \tilde{X}^{\leq t_k-}\right] - Y_{t-}^{\theta_m^{\min}}\left(\tilde{X}^{\leq t_k-}\right)\right|_2 \mu_{t_k}(dt\,; \tilde{X}^{\leq t_k-}) > \epsilon\right) \xrightarrow{m \to \infty} 0. \tag{15}$$

In the following, we study two examples of the conditional distribution $\mu_{t_k}$ that are relevant in practice. We study them for $Y^{\theta_m^{\min}}$ but note that the hold equivalently for $Y^{\theta_{m,N_m}^{\min}}$. We first remark that in general the support of $t_k$ is a $\mathcal{A}_{t_{k-1}}$-measurable random set (in particular, it often depends at least on $t_{k-1}$). We denote this set-valued random variable with

$$S_{t_k} := \operatorname{supp}\left(t_k \,|\, \mathcal{A}_{t_{k-1}}\right),$$

which is defined to be the smallest closed random subset $S_{t_k} \subset [0,T]$ such that $\mu_{t_k}(S_{t_k}\,;\tilde{X}^{\leq t_k -}) = 1$ $\mathbb{P}_k$-almost surely. By Doob-Dynkin's lemma (Taraldsen, 2018, Lemma 2) we can write $S_{t_k}$ as a function $\tilde{X}^{\leq t_k -}$, i.e., $S_{t_k} = S_{t_k}(\tilde{X}^{\leq t_k -})$.

**Example 5.3.** *Assume that $S_k := \cup_\omega S_{t_k}(\omega)$ is finite and that there exists some $\alpha > 0$ such that for all $t \in S_k$ we have $\mu_{t_k}(\{t\}\,;\tilde{X}^{\leq t_k -}) \geq \alpha \mathbb{1}_{\{t > t_{k-1}\}}$ $\mathbb{P}_k$-almost surely. This is the case if all observation times are sampled from a (finite) grid with positively lower bounded probability to take any value of the grid, which is larger than the previous observation time (as in the synthetic examples in this paper and in Krach et al. (2022)). Then Proposition 5.2 implies that for every $1 \leq k \leq K$*

$$\sum_{t \in S_k} \mathbb{E}_k \left[ \mathbb{1}_{\{t > t_{k-1}\}} \left| \mathbb{E}\left[ X_{t-} \,|\, \tilde{X}^{\leq t_k -} \right] - Y_{t-}^{\theta_m^{\min}}\left(\tilde{X}^{\leq t_k -}\right) \right|_2 \right] \xrightarrow{m \to \infty} 0$$

*and (15) implies that for every $\epsilon, \delta > 0$ there exists $m_0 \in \mathbb{N}$ such that for $m \geq m_0$ there is a subset $\Omega_m \subset \Omega$ with $\mathbb{P}_k(\Omega_m) > 1 - \delta$ and for every $\omega \in \Omega_m$ with $n(\omega) \geq k$ and every $t \in S_k$ with $t > t_{k-1}(\omega)$*

$$\left| \mathbb{E}\left[ X_{t-} \,|\, \tilde{X}^{\leq t_k -}(\omega) \right] - Y_{t-}^{\theta_m^{\min}}\left(\tilde{X}^{\leq t_k -}(\omega)\right) \right|_2 < \epsilon.$$

*In particular, with high probability, our model is close to the optimal prediction at the left limit of every possible grid point if $m$ is large enough. At the same time it is apparent that we cannot infer anything for $t \notin S_k$. In our synthetic examples this is not a problem since we only plot our model and the optimal prediction (and measure the distance between them) on the grid points.*

**Remark 5.4** (Convergence in the Evaluation Metric). *In the same setting as in Example 5.3, if there exists one finite grid from which all observation times are sampled, i.e., $S \subset [0,T]$ finite, such that $S_k \subset S$ for all $k$ and $T \in S$, then $n \leq |S|$. Revisiting the proof of Theorem 2.7 in Appendix C, we note that (23) can therefore be replaced by*

$$\mathbb{E}\left[|Z|_2^2\right] \leq |S|\,\mathbb{E}\left[\frac{1}{n}|Z|_2^2\right],$$

*and therefore we can show the convergence stated in Theorem 2.7 in the stronger $L^2$-type (pseudo) metric*

$$d_k^2(Z,\xi) := c_0(k)\,\mathbb{E}\left[\mathbb{1}_{\{n \geq k\}}|Z_{t_k -} - \xi_{t_k -}|_2^2\right], \tag{16}$$

*for which Proposition 5.2 holds equivalently.*

*We assume that for $\mathbb{P}$-a.e. $\omega$ and for every $t \in S$ we have that $t_{k-1}(\omega) < t \leq t_k(\omega)$ implies that $t \in S_{t_k}(\omega)$. Then the evaluation metric (cf. Section 6) defined on $S$ for $N_{test}$ i.i.d. test samples is*

$$\operatorname{eval}_{S,N_{test}}(\hat{X}, Y^{\theta_{m,N_m}^{\min}}) := \frac{1}{|S|} \sum_{t \in S} \frac{1}{N_{test}} \sum_{j=1}^{N_{test}} \left| \mathbb{E}\left[ X_{t-} \,\Big|\, \tilde{X}^{\leq t-,(j)} \right] - Y_{t-}^{\theta_{m,N_m}^{\min}}\left(\tilde{X}^{\leq t-,(j)}\right) \right|_2^2$$

*and by a similar argument as in the second part of the proof of Theorem 2.7 (cf. (31), (32)), we have*

$$\operatorname{eval}_{S,N_{test}}(\hat{X}, Y^{\theta_{m,N_m}^{\min}}) \xrightarrow[N_{test} \to \infty]{\mathbb{P}-a.s.} \mathbb{E}\left[ \frac{1}{|S|} \sum_{t \in S} \left| \mathbb{E}\left[ X_{t-} \,\big|\, \tilde{X}^{\leq t-} \right] - Y_{t-}^{\theta_{m,N_m}^{\min}}\left(\tilde{X}^{\leq t-}\right) \right|_2^2 \right] =: (I).$$

*If we add pseudo observation times at $T$ (by choosing the respective mask to be $0$ in case there is no actual observation at $T$, cf. Remark 2.2), we have $t_n = T$.[8] Then, using that $S$ is finite and with our assumption*

---

[8]We need to do this because otherwise we have terms where $t > t_n$ which cannot be upper bounded by the distances in $d_k^2$. However, we note that the objective function is the same except for dividing by $n+1$ instead of $n$ for the samples which do not have an observation at $T$. Hence, also the trained model is nearly the same.

*for $S_{t_k}$, we have*

$$
\begin{aligned}
(I) &= \sum_{k=1}^{|S|} \mathbb{E}\left[\mathbb{1}_{\{n=k\}} \frac{1}{|S|} \sum_{t \in S} \left| \mathbb{E}\left[X_{t-}\big| \tilde{X}^{\leq t-}\right] - Y_{t-}^{\theta_{m,N_m}^{\min}}\left(\tilde{X}^{\leq t-}\right)\right|_2^2\right] \\
&= \frac{1}{|S|} \sum_{k=1}^{|S|} \mathbb{E}\left[\mathbb{1}_{\{n=k\}} \sum_{\kappa=1}^{k} \sum_{t \in S_\kappa} \mathbb{1}_{\{t_{\kappa-1}<t\leq t_\kappa\}} \left| \mathbb{E}\left[X_{t-}\big| \tilde{X}^{\leq t_\kappa-}\right] - Y_{t-}^{\theta_{m,N_m}^{\min}}\left(\tilde{X}^{\leq t_\kappa-}\right)\right|_2^2\right] \\
&\leq \frac{1}{|S|} \sum_{k=1}^{|S|} \mathbb{E}\left[\mathbb{1}_{\{n=k\}} \sum_{\kappa=1}^{k} \mathbb{1}_{\{n\geq\kappa\}} \sum_{t \in S_\kappa} \mathbb{1}_{\{t_{\kappa-1}<t\}} \left| \mathbb{E}\left[X_{t-}\big| \tilde{X}^{\leq t_\kappa-}\right] - Y_{t-}^{\theta_{m,N_m}^{\min}}\left(\tilde{X}^{\leq t_\kappa-}\right)\right|_2^2\right] \\
&\leq \frac{1}{|S|} \sum_{k=1}^{|S|} \mathbb{E}\left[\sum_{\kappa=1}^{|S|} \mathbb{1}_{\{n\geq\kappa\}} \sum_{t \in S_\kappa} \mathbb{1}_{\{t_{\kappa-1}<t\}} \left| \mathbb{E}\left[X_{t-}\big| \tilde{X}^{\leq t_\kappa-}\right] - Y_{t-}^{\theta_{m,N_m}^{\min}}\left(\tilde{X}^{\leq t_\kappa-}\right)\right|_2^2\right] \\
&\leq \sum_{\kappa=1}^{|S|} \sum_{t \in S_\kappa} \mathbb{E}\left[\mathbb{1}_{\{n\geq\kappa\}} \mathbb{1}_{\{t_{\kappa-1}<t\}} \left| \mathbb{E}\left[X_{t-}\big| \tilde{X}^{\leq t_\kappa-}\right] - Y_{t-}^{\theta_{m,N_m}^{\min}}\left(\tilde{X}^{\leq t_\kappa-}\right)\right|_2^2\right] \\
&\leq \sum_{\kappa=1}^{|S|} \sum_{t \in S_\kappa} \mathbb{E}_\kappa\left[\mathbb{1}_{\{t_{\kappa-1}<t\}} \left| \mathbb{E}\left[X_{t-}\big| \tilde{X}^{\leq t_\kappa-}\right] - Y_{t-}^{\theta_{m,N_m}^{\min}}\left(\tilde{X}^{\leq t_\kappa-}\right)\right|_2^2\right] \xrightarrow{m\to\infty} 0,
\end{aligned}
$$

*which converges to 0 according to Example 5.3. In particular, we have $\mathbb{P}$-almost surely*

$$
\lim_{m\to\infty} \lim_{N_{test}\to\infty} \mathrm{eval}_{S,N_{test}}(\hat{X}, Y^{\theta_{m,N_m}^{\min}}) = 0,
$$

*i.e., the evaluation metric converges to 0 when the number of evaluation samples, the number of training samples and the network sizes increase. Moreover, if the number of evaluation samples $N_{test} < \infty$ is fixed, the evaluation metric $\mathrm{eval}_{S,N_{test}}(\hat{X}, Y^{\theta_{m,N_m}^{\min}})$ converges to zero in probability as $m$ and $N_m$ tend to infinity.*

**Example 5.5.** *Assume that for $1 \leq k \leq K$, $\mathbb{P}_k$-almost surely we have that $S_{t_k} = (t_{k-1}, T]$, that $\mu_{t_k}(\cdot; \tilde{X}^{\leq t_k-})$ is absolutely continuous with respect to the Lebesgue measure on $[0,T]$, and that its Radon-Nikodym derivative w.r.t. the Lebesgue measure satisfies*

$$
\frac{d\mu_{t_k}(\cdot; \tilde{X}^{\leq t_k-})}{dt} \geq \alpha\, \mathbb{1}_{(t_{k-1},T]}(\cdot)
$$

*for some $\alpha > 0$. This is for example the case if the observation time $t_k$ is uniformly or exponentially (with an upper bound) distributed on $(t_{k-1}, T]$. Then Proposition 5.2 implies that*

$$
\mathbb{E}_k\left[\int_{t_{k-1}}^{T} \left| \mathbb{E}\left[X_{t-}\big| \tilde{X}^{\leq t_k-}\right] - Y_{t-}^{\theta_m^{\min}}\left(\tilde{X}^{\leq t_k-}\right)\right|_2 dt\right] \xrightarrow{m\to\infty} 0
$$

*and (15) implies that for every $\epsilon, \delta > 0$ there exists $m_0 \in \mathbb{N}$ such that for $m \geq m_0$ there is a subset $\Omega_m \subset \Omega$ with $\mathbb{P}_k(\Omega_m) > 1 - \delta$ and for every $\omega \in \Omega_m$ with $n(\omega) \geq k$*

$$
\int_{t_{k-1}}^{T} \left| \mathbb{E}\left[X_{t-}\big| \tilde{X}^{\leq t_k-}(\omega)\right] - Y_{t-}^{\theta_m^{\min}}\left(\tilde{X}^{\leq t_k-}(\omega)\right)\right|_2 dt < \epsilon.
$$

*In particular, with high probability, the $L^1$-distance on $(t_{k-1}, T]$ between our model and the optimal prediction is small if $m$ is large enough. Since our model and the optimal prediction are both continuous in $t$, we know that in the limit they have to be point-wise the same, i.e., for every $t \in (t_{k-1}, T]$.*

To summarize, under Assumption 4.1, at every time $s$, for every future time $t \in [s, T]$ that was not deterministically excluded from being the next observation time given the previous observations,[9] our model $Y_t^{\theta_{m,N_m}^{\min}}(\tilde{X}^{\leq s})$ converges in probability to the correct conditional expectation $\mathbb{E}[X_t \,|\, \mathcal{A}_s]$ as $m$ tends to infinity.

---

[9]Mathematically this just means, "for every $t \in S_{t_k}$" given that $\tau(s) = t_{k-1}$ for some $k$.

### 5.3 Implications for a Trained PD-NJ-ODE model

Finally, we discuss what we can say about the distance between the true conditional expectation $\hat{X}$ and the output of the PD-NJ-ODE model $Y^{\theta^\star}$ for parameters $\theta^\star$ which are $\varepsilon$-optimal. This is an important practical question which arises, when training the model with some optimization scheme (e.g. a version of stochastic gradient descent) yields such parameters $\theta^\star$. We note that the existence of such parameters follows from Theorem 2.7 or its versions for noisy observations (Theorem 3.3), dependence (Theorem 4.3) or both extensions (Theorem C.3).

**Proposition 5.6.** *Assume that $\theta^\star$ are $\varepsilon$-optimal parameters for the PD-NJ-ODE, i.e.,*

$$\Phi(\theta^\star) \leq \inf_{\theta \in \Theta} \Phi(\theta) + \varepsilon,$$

*for some $\varepsilon > 0$.[10] Then for every $1 \leq k \leq K$ there exists a constant $c = c(k) > 0$ independent of $\theta^\star$ and $\varepsilon$ such that $d_k(\hat{X}, Y^{\theta^\star}) \leq c\sqrt{\varepsilon}$.*

*Proof.* The claim follows from applying (1), (24), (23) and finally Lemma C.4, which was summarized in (29) in the full proof of the main theorem with both extensions. $\square$

We note that the results from Section 5.2 equivalently hold here. Therefore, we are able to make statements about the distance between the paths of $\hat{X}$ and $Y^{\theta^\star}$.

**Remark 5.7.** *Under the same assumption as in Proposition 5.6 as well as Assumption 4.1, we have that for every $1 \leq k \leq K$*

$$d_k(\hat{X}, Y^{\theta^\star}) = \mathbb{E}_k \left[ \int_0^T \left| \mathbb{E} \left[ X_{t-} \big| \tilde{X}^{\leq t_k -} \right] - Y_{t-}^{\theta^\star} \left( \tilde{X}^{\leq t_k -} \right) \right|_2 \mu_{t_k}(dt\,; \tilde{X}^{\leq t_k -}) \right] \leq c(k)\sqrt{\varepsilon}.$$

*Moreover, if additionally the assumptions of Example 5.3 are satisfied, then we have*

$$\sum_{t \in S_k} \mathbb{E}_k \left[ \mathbb{1}_{\{t > t_{k-1}\}} \left| \mathbb{E} \left[ X_{t-} \big| \tilde{X}^{\leq t_k -} \right] - Y_{t-}^{\theta^\star} \left( \tilde{X}^{\leq t_k -} \right) \right|_2 \right] \leq \frac{c(k)}{\alpha} \sqrt{\varepsilon}.$$

*Equivalently, if additionally the assumptions of Example 5.5 are satisfied, then we have*

$$\mathbb{E}_k \left[ \int_{t_{k-1}}^T \left| \mathbb{E} \left[ X_{t-} \big| \tilde{X}^{\leq t_k -} \right] - Y_{t-}^{\theta^\star} \left( \tilde{X}^{\leq t_k -} \right) \right|_2 dt \right] \leq \frac{c(k)}{\alpha} \sqrt{\varepsilon}.$$

## 6 Experiments

The code with all new experiments and those from Krach et al. (2022) is available at https://github.com/FlorianKrach/PD-NJODE. Further details about the experiments can be found in Appendix D. In particular, in Appendix D.1 we give details on the slight deviation of the practical implementation from the theoretical description.

As in Krach et al. (2022) we use the following evaluation metric to quantify and compare the training success.

$$\text{eval}(\hat{X}, Y^\theta) := \frac{1}{N_{\text{test}}} \sum_{j=1}^{N_{\text{test}}} \frac{1}{\kappa + 1} \sum_{i=0}^{\kappa} \left| \hat{X}_{\frac{iT}{\kappa}-}^{(j)} - Y_{\frac{iT}{\kappa}-}^{\theta, j} \right|_2^2,$$

where the outer sum runs over the test set of size $N_{\text{test}}$ and the inner sum runs over the equidistant grid points on the time interval $[0, T]$.

In Sections 6.1 and 6.2 we provide two illustrative experiments on easy synthetic datasets for the extensions discussed in this paper. We note that the extension to a dependence between the observation framework and

---

[10]Note that $\inf_{\theta \in \Theta} \Phi(\theta) = \min_{Z \in \mathbb{D}} \Psi(Z) = \Psi(\hat{X})$ according to our main theorem.

the process $X$ is purely theoretical, not changing anything in the implementation of the model, hence, the experiments on real world datasets of Krach et al. (2022) are representative for this extension as well. In particular, it is very likely that there actually is such a dependence in the Physionet dataset (Goldberger et al., 2000), as discussed in the introduction and in Section 4.1. Hence, our paper provides the theoretical foundation for those empirical results of Krach et al. (2022) and, vice versa, those results show that the method discussed in our paper is applicable to complex high-dimensional settings.

In Section 6.3 we provide further experiments on the Physionet dataset to compare the noise-adapted training framework of Section 3 with the original one.

## 6.1 Noisy Observations – Brownian Motion with Gaussian Observation Noise

We test the PD-NJ-ODE trained with the loss function adapted to noisy observations (6) in the context of Section 3.4.1. In particular, $X$ is a Brownian motion and we assume to have observation noise of a centered normal distribution with standard deviation $\sigma = 0.5$. Moreover, we compare these results to using the original loss function (3) with the noisy observation. PD-NJ-ODE adapted to noisy observations achieves a minimal evaluation metric of $1.1 \cdot 10^{-3}$ while using the original loss function leads to a nearly 20 times larger evaluation metric of $1.9 \cdot 10^{-2}$. Moreover, in Figure 1 we see that the noise-adapted method learns to correctly jump when new observations become available, while the original method jumps to the noisy observations and afterwards tries to get close to the true conditional expectation quickly. We note that this is the expected behaviour.

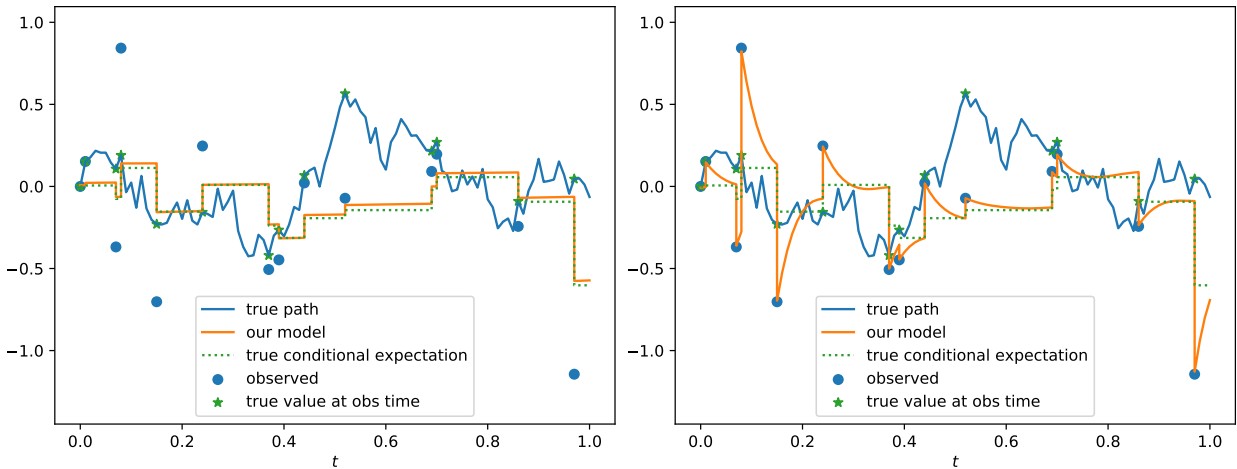

Figure 1: A test sample of a Brownian motion $X$ with noisy observations $O_{t_i} = X_{t_i} + \epsilon_i$ together with the true and predicted conditional expectation. The PD-NJ-ODE is trained with the noise-adapted loss (left) and the original loss (right).

## 6.2 Dependent Observation Framework – a Black–Scholes Example

Based on Example 4.13 we train a PD-NJ-ODE on a 1-dimensional geometric Brownian motion with drift $\mu = 2$ and volatility $\sigma = 0.3$ and with observation probability depending on the last observation of $X$, the time since the last observation and on independent random variables $V_{\{2,3\}}^i$ for which we use the parameters $\eta = 3$ and $p = 0.1$. As our theoretical result suggests, our model learns to predict the conditional expectation well with a minimal evaluation metric of $1.1 \cdot 10^{-3}$ which is also visible in Figure 2.

## 6.3 Physionet with Observation Noise

To test the new training framework for noisy observations in a 41-dimensional, complex real world setting, we use the Physionet dataset (Goldberger et al., 2000). Even though the Physionet dataset, as it is, is

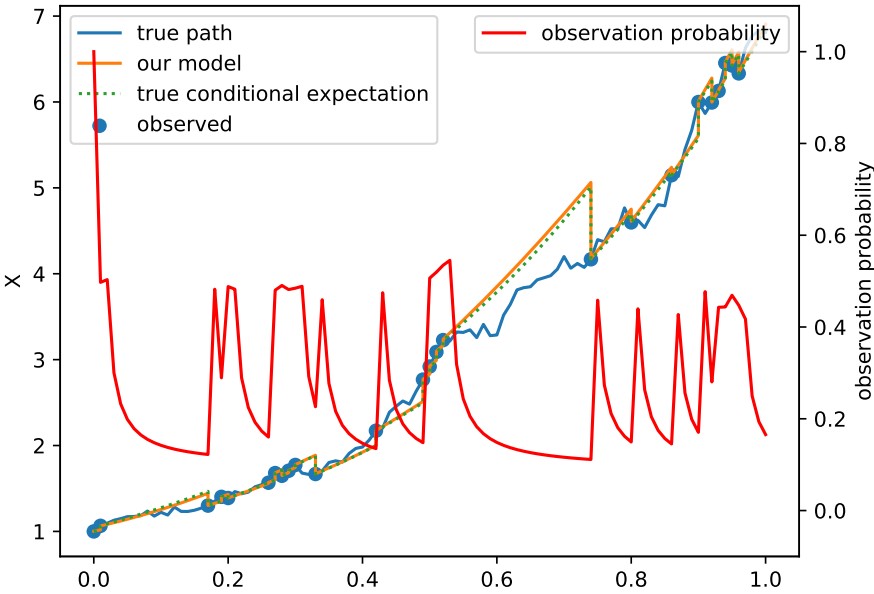

Figure 2: A test sample of a geometric Brownian motion $X$ with observation probability depending on previously observed values of $X$. Plotted together with the true and predicted conditional expectation (scale on the left) and the observation probability over time (scale on the right).

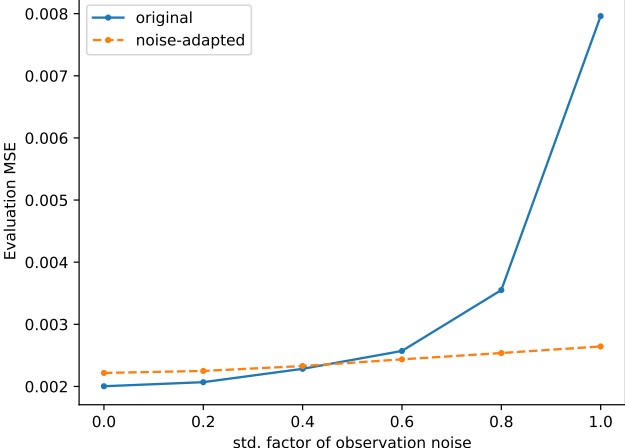

Figure 3: Evaluation MSE of the PD-NJ-ODE trained on the Physionet dataset with the original and the noise-adapted loss with different size for the standard deviation factor of the synthetically added observation noise.

likely to have some observation noise, it turns out that this noise is too small to have a big impact on the training (cf. Figure 3 and Section 3.5). Therefore, we add synthetic noise to all observations that are used as input for the PD-NJ-ODE (but not to the observation on which the evaluation MSE is computed) and study its impact on the evaluation MSE depend on the size of its standard deviation. More precisely, we first compute each coordinate's standard deviation $\sigma_{data,j}$ on the training set and then add i.i.d. noise samples $\epsilon_{i,j} \sim \mathcal{N}(0, \zeta^2 \, \sigma_{data,j}^2)$, for $\zeta \geq 0$, to each observed coordinate $X_{t_i,j}$. In Figure 3 we show the minimal evaluation MSE results when training the PD-NJ-ODE with the original loss (5) and the noise-adapted loss (see Equations (6) and (7)) on these datasets for increasing standard deviation factor $\zeta$. Without additional noise ($\zeta = 0$), the original loss function leads to slightly better results than the noise-adapted loss function (which is approximately 10% larger), suggesting that in this case the potential intrinsic observation noise has a smaller impact than the inductive bias of the original loss function (cf. Section 3.5). The larger the

synthetically added noise gets, the larger is its impact. For the noise-adapted loss function the evaluation MSE only grows linearly (with a small slope) when $\zeta$ increases. In contrast to this, it grows more than linearly for the original loss function. For some $\zeta \in (0.4, 0.5)$ there is the turning point, where the noise-adapted framework becomes more efficient. Overall, the evaluation MSE grows less than 20% for the noise-adapted loss (from $\zeta = 0$ to $\zeta = 1$), while it grows by nearly 300% for the original loss function. Moreover, the evaluation MSE with $\zeta = 1$ is more than 3 times larger for the original PD-NJ-ODE than for the noise-adapted one. This example demonstrates well the importance of the noise-adapted training framework when the observation noise is large.

## 7  Conclusion

In this work we broadened the applicability of the PD-NJ-ODE of Krach et al. (2022) by extending the theoretical foundation to allow for the observation framework (i.e., observation times and masks) to depend on previous information and additionally proposed a new loss function that provably leads to optimal predictions even if observations are noisy. In particular, we showed that any centered i.i.d. observation noise satisfying some integrability conditions can be dealt with by switching to the noise-adapted objective function (6). Moreover, we showed that the proof of the main result can be retained when lifting the independence assumption between the process $X$ and the observation framework, by extensively working with conditional independence. Finally, we provided experiments showing empirically that the PD-NJ-ODE works well in those extended settings.

## Acknowledgement

All authors would like to thank the anonymous reviewers for their detailed feedback and great suggestions that led to significant improvements of the paper.

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

## Appendix

## A    Conditional independence

Let $(\Omega, \mathcal{F}, \mathbb{P})$ be a probability space, $\mathcal{G}, \mathcal{H} \subseteq \mathcal{F}$ be two sigma-algebras and let $X$ be a random variable.

The assumption that $X$ is independent of $\mathcal{G}$ leads to the natural but incorrect conclusion that $\mathbb{E}[X \mid \boldsymbol{\sigma}(\mathcal{G}, \mathcal{H})] = \mathbb{E}[X \mid \mathcal{H}]$. For this to hold, we actually need a stronger assumption as in the following proposition that is due to Hansen (2015); Yoo (2014).

**Proposition A.1.** *If $\boldsymbol{\sigma}(X, \mathcal{H})$ is independent of $\mathcal{G}$ then $\mathbb{E}[X \mid \boldsymbol{\sigma}(\mathcal{G}, \mathcal{H})] = \mathbb{E}[X \mid \mathcal{H}]$.*

*Proof.* We prove the desired equality in the context of basic measure theory. We first assume that $X$ is integrable on $\boldsymbol{\sigma}(\mathcal{G}, \mathcal{H})$, since otherwise neither of the expectations are valid. Then we recall that conditional expectation is simply a random variable $Z$ that satisfies the following three properties

1. $Z$ is $\boldsymbol{\sigma}(\mathcal{G}, \mathcal{H})$-measurable,

2. $Z$ is integrable,

3. $\int_A X d\mathbb{P} = \int_A Z d\mathbb{P}$ for all $A \in \boldsymbol{\sigma}(\mathcal{G}, \mathcal{H})$.

By the definition of the conditional expectation we know that $\mathbb{E}[X \mid \boldsymbol{\sigma}(\mathcal{G}, \mathcal{H})]$ satisfies these properties. To show the claim, it is therefore enough to prove that also $Z = \mathbb{E}[X \mid \mathcal{H}]$ satisfies them. The first two follow trivially since $\mathcal{H} \subseteq \boldsymbol{\sigma}(\mathcal{G}, \mathcal{H})$. For the third, we note that by Dynkin's $\pi$-$\lambda$ theorem it is enough to consider the $\cap$-stable generator $P = \{A \cap B \mid A \in \mathcal{G}, B \in \mathcal{H}\}$ of $\boldsymbol{\sigma}(\mathcal{G}, \mathcal{H})$ and show that

$$\int_{A \cap B} X d\mathbb{P} = \int_{A \cap B} Z d\mathbb{P} \quad \forall A \in \mathcal{G}, B \in \mathcal{H},$$

or equivalently

$$\mathbb{E}\left[X \cdot \mathbb{1}_{A \cap B}\right] = \mathbb{E}\left[\mathbb{E}\left[X \mid \mathcal{H}\right] \cdot \mathbb{1}_{A \cap B}\right] \quad \forall A \in \mathcal{G}, B \in \mathcal{H}. \tag{17}$$

Indeed, $P$ is a $\pi$-system (with $\boldsymbol{\sigma}(P) = \boldsymbol{\sigma}(\mathcal{G}, \mathcal{H})$) that is trivially included in the smallest Dynkin system $D$ including $P$ (this is the system that includes all sets in $P$, is closed under complements and under countable unions of disjoint sets). Moreover, if (17) holds for every set in $P$, it is easy to see (by linearity of the integral and dominated convergence) that the corresponding property also holds for every set in $D$, since for a set $A \in D$ we have $\mathbb{1}_{A^{\complement}} = 1 - \mathbb{1}_A$ and for disjoint sets $(A_k)_{k \geq 1} \in D$ we have $\mathbb{1}_{\cup_{k \geq 1} A_k} = \sum_{k \geq 1} \mathbb{1}_{A_k}$. Dynkin's $\pi$-$\lambda$ theorem implies that $\boldsymbol{\sigma}(P) \subset D$, therefore the third point follows.

To show (17), note that $\boldsymbol{\sigma}(X, \mathcal{H})$ being independent of $\mathcal{G}$ implies that $\mathcal{H}$ is independent of $\mathcal{G}$. Therefore,

$$
\begin{aligned}
\mathbb{E}\left[X \cdot \mathbb{1}_{A \cap B}\right] &= \mathbb{E}\left[X \cdot \mathbb{1}_B \mathbb{1}_A\right] \\
&= \mathbb{E}\left[X \cdot \mathbb{1}_B\right] \mathbb{E}\left[\mathbb{1}_A\right] && (\boldsymbol{\sigma}(X, \mathcal{H}) \text{ indep. of } \mathcal{G}) \\
&= \mathbb{E}\left[\mathbb{E}\left[X \cdot \mathbb{1}_B \mid \mathcal{H}\right]\right] \cdot \mathbb{E}\left[\mathbb{1}_A\right] && (\text{tower property}) \\
&= \mathbb{E}\left[\mathbb{E}\left[X \cdot \mathbb{1}_B \mid \mathcal{H}\right] \cdot \mathbb{1}_A\right] && (\mathcal{H} \text{ indep. of } \mathcal{G}) \\
&= \mathbb{E}\left[\mathbb{E}\left[X \mid \mathcal{H}\right] \cdot \mathbb{1}_A \mathbb{1}_B\right] && (\mathbb{1}_B \text{ is } \mathcal{H}\text{-measurable}) \\
&= \mathbb{E}\left[\mathbb{E}\left[X \mid \mathcal{H}\right] \cdot \mathbb{1}_{A \cap B}\right],
\end{aligned}
$$

completing the proof. □

**Corollary A.2.** *If $\boldsymbol{\sigma}(X, \mathcal{H})$ is independent of $\mathcal{G}$ then $\mathbb{E}[\phi(X) \mid \boldsymbol{\sigma}(\mathcal{G}, \mathcal{H})] = \mathbb{E}[\phi(X) \mid \mathcal{H}]$ for all measurable and integrable functions $\phi$.*

*Proof.* This follows directly from Proposition A.1 upon replacing $X$ by $\phi(X)$ and noting that $\boldsymbol{\sigma}(\phi(X), \mathcal{H}) \subseteq \boldsymbol{\sigma}(X, \mathcal{H})$, implying the needed condition. □

It should now be apparent that the assumption that $X$ and $\mathcal{G}$ are independent is insufficient if we want to show $\mathbb{E}[X \,|\, \boldsymbol{\sigma}(\mathcal{G}, \mathcal{H})] = \mathbb{E}[X \,|\, \mathcal{H}]$ (a counterexample is provided in Hansen (2015)). However, we can actually make a weaker assumption (though still stronger than the assumption that $X$ and $\mathcal{G}$ are independent) to attain the same result, as shown in the next proposition.

**Definition A.3.** *$X$ and $\mathcal{G}$ are conditionally independent given $\mathcal{H}$ if for all $x \in \boldsymbol{\sigma}(X)$, $A \in \mathcal{G}$,*

$$\mathbb{E}\left[\mathbb{1}_x \mathbb{1}_A \,|\, \mathcal{H}\right] = \mathbb{E}\left[\mathbb{1}_x \,|\, \mathcal{H}\right] \mathbb{E}\left[\mathbb{1}_A \,|\, \mathcal{H}\right].$$

**Proposition A.4.** *If $X$ is conditionally independent of $\mathcal{G}$ given $\mathcal{H}$ then $\mathbb{E}[X \,|\, \boldsymbol{\sigma}(\mathcal{G}, \mathcal{H})] = \mathbb{E}[X \,|\, \mathcal{H}]$.*

*Proof.* Following the previous proof it is clear that we only need to show

$$\mathbb{E}\left[X \cdot \mathbb{1}_{A \cap B}\right] = \mathbb{E}\left[\mathbb{E}\left[X \,|\, \mathcal{H}\right] \cdot \mathbb{1}_{A \cap B}\right]$$

for all $A \in \mathcal{G}$ and $B \in \mathcal{H}$. We do this by measure-theoretic induction (as in Durrett (2010, Proof of Theorem 1.6.9)), in particular, we proceed in a four-part case distinction of $X$. In the first case, assume $X$ is an indicator function, i.e., $X = \mathbb{1}_x$ for some $x \in \mathcal{F}$. Then

$$
\begin{aligned}
\mathbb{E}\left[X \cdot \mathbb{1}_{A \cap B}\right] &= \mathbb{E}\left[\mathbb{1}_x \mathbb{1}_{A \cap B}\right] \\
&= \mathbb{E}\left[\mathbb{E}\left[\mathbb{1}_x \mathbb{1}_{A \cap B} \,|\, \mathcal{H}\right]\right] && \text{(tower property)} \\
&= \mathbb{E}\left[\mathbb{E}\left[\mathbb{1}_x \mathbb{1}_A \,|\, \mathcal{H}\right] \mathbb{1}_B\right] && (B \in \mathcal{H}) \\
&= \mathbb{E}\left[\mathbb{E}\left[\mathbb{1}_x \,|\, \mathcal{H}\right] \mathbb{E}\left[\mathbb{1}_A \,|\, \mathcal{H}\right] \mathbb{1}_B\right] && \text{(cond. indep.)} \\
&= \mathbb{E}\left[\mathbb{E}\left[\mathbb{E}\left[\mathbb{1}_x \,|\, \mathcal{H}\right] \cdot \mathbb{1}_A \,|\, \mathcal{H}\right] \mathbb{1}_B\right] && (\mathbb{E}\left[\mathbb{1}_x \,|\, \mathcal{H}\right] \, \mathcal{H}\text{-mbl.}) \\
&= \mathbb{E}\left[\mathbb{E}\left[\mathbb{E}\left[\mathbb{1}_x \,|\, \mathcal{H}\right] \cdot \mathbb{1}_{A \cap B} \,|\, \mathcal{H}\right]\right] && (B \in \mathcal{H}) \\
&= \mathbb{E}\left[\mathbb{E}\left[\mathbb{1}_x \,|\, \mathcal{H}\right] \cdot \mathbb{1}_{A \cap B}\right] && \text{(tower property)} \\
&= \mathbb{E}\left[\mathbb{E}\left[X \,|\, \mathcal{H}\right] \cdot \mathbb{1}_{A \cap B}\right].
\end{aligned}
$$

Thus in the most basic case we have the required property. In the second case we let $X = \sum_{i=1}^{k} c_i \mathbb{1}_{x_i}$ be a finite weighted sum of indicator random variables, where $k \in \mathbb{N}$, $c_i \in \mathbb{R}$ and $x_i \in \mathcal{F}$. The result of the first case combined with linearity of expectation immediately shows that the property holds for $X$ in this form too.

In the third case, we assume $X$ is some non-negative function. We define a random variable $X_k$ that is simple and such that $X_k \uparrow X$ as $k \to \infty$. For example, we can take $X_k = \sum_{i=0}^{k2^k - 1} \frac{i}{2^k} \mathbb{1}\{\frac{i}{2^k} \le X < \frac{i+1}{2^k}\} + k \mathbb{1}_{k \le X}$, as in Durrett (2010, Proof of Theorem 1.6.9). Then, by monotone convergence and the previous case, the property also holds when $X$ is an arbitrary non-negative function.

Finally, in the fourth case, we let $X$ be an arbitrary integrable function. Then we can write $X = X^+ - X^-$ where $f^+(x) := \max\{0, f(x)\}$ and $f^-(x) := \min\{0, f(x)\}$. Integrability of $X$ means that $X^+$ and $X^-$ are themselves integrable. We can use linearity of expectation and the previous cases to conclude that, in this general setting, the property still holds. This concludes the proof by measure-theoretic induction. $\quad\square$

**Proposition A.5.** *$X$ is conditionally independent of $\mathcal{G}$ given $\mathcal{H}$ if and only if $\mathbb{E}[\phi(X) \,|\, \boldsymbol{\sigma}(\mathcal{G}, \mathcal{H})] = \mathbb{E}[\phi(X) \,|\, \mathcal{H}]$ for all measurable and integrable functions $\phi$.*

*Proof.* The "$\Rightarrow$" direction is a simple corollary of Proposition A.4 that follows with the same proof upon replacing $X$ by $\phi(X)$ and noticing that $\phi(X)$ is a $\boldsymbol{\sigma}(X)$-measurable random variable.

The "$\Leftarrow$" direction is also easy to see. Let $x \in \boldsymbol{\sigma}(X)$, $A \in \mathcal{G}$ and define $\phi$ such that $\phi(X) = \mathbb{1}_x$. Then we have

$$
\begin{aligned}
\mathbb{E}\left[\mathbb{1}_x \mathbb{1}_A \,|\, \mathcal{H}\right] &= \mathbb{E}\left[\mathbb{E}\left[\mathbb{1}_x \mathbb{1}_A \,|\, \boldsymbol{\sigma}(\mathcal{G}, \mathcal{H})\right] \,|\, \mathcal{H}\right] && \text{(tower property)} \\
&= \mathbb{E}\left[\mathbb{E}\left[\mathbb{1}_x \,|\, \boldsymbol{\sigma}(\mathcal{G}, \mathcal{H})\right] \mathbb{1}_A \,|\, \mathcal{H}\right] && (A \in \mathcal{G} \subseteq \boldsymbol{\sigma}(\mathcal{G}, \mathcal{H})) \\
&= \mathbb{E}\left[\mathbb{E}\left[\mathbb{1}_x \,|\, \mathcal{H}\right] \mathbb{1}_A \,|\, \mathcal{H}\right] && \text{(RHS of claim with } \phi(X) = \mathbb{1}_x) \\
&= \mathbb{E}\left[\mathbb{1}_x \,|\, \mathcal{H}\right] \mathbb{E}\left[\mathbb{1}_A \,|\, \mathcal{H}\right] && (\mathbb{E}\left[\mathbb{1}_x \,|\, \mathcal{H}\right] \, \mathcal{H}\text{-mbl.})
\end{aligned}
$$

This completes the proof. $\quad\square$

**Proposition A.6.** *If $X$ is conditionally independent of $\mathcal{G}$ given $\mathcal{H}$, and $Y$ is independent of $\boldsymbol{\sigma}(X, \mathcal{G}, \mathcal{H})$ then for any measurable and integrable function $f$ also $f(X, Y)$ is conditionally independent of $\mathcal{G}$ given $\mathcal{H}$.*

*Proof.* By Proposition A.5 we have to show that for any measurable $\phi$ we have $\mathbb{E}[\phi(f(X, Y)) \mid \boldsymbol{\sigma}(\mathcal{G}, \mathcal{H})] = \mathbb{E}[\phi(f(X, Y)) \mid \mathcal{H}]$. First note that it is enough to show this for $\phi$ being the identity, since both $\phi$ and $f$ are arbitrary measurable and integrable functions. Then we have for $g(x) := \mathbb{E}_Y[f(X, Y)]$, which is a measurable and integrable function again, that

$$
\begin{aligned}
\mathbb{E}\left[f(X, Y) \mid \boldsymbol{\sigma}(\mathcal{G}, \mathcal{H})\right] &= \mathbb{E}\left[\mathbb{E}\left[f(X, Y) \mid \boldsymbol{\sigma}(X, \mathcal{G}, \mathcal{H})\right] \mid \boldsymbol{\sigma}(\mathcal{G}, \mathcal{H})\right] && \text{(tower property)} \\
&= \mathbb{E}\left[g(X) \mid \boldsymbol{\sigma}(\mathcal{G}, \mathcal{H})\right] && \text{(Durrett (2010, Lemma 6.2.1))} \\
&= \mathbb{E}\left[g(X) \mid \mathcal{H}\right] && \text{(Proposition A.5)} \\
&= \mathbb{E}\left[f(X, Y) \mid \mathcal{H}\right], && \text{(reversing step 1 \& 2)}
\end{aligned}
$$

as wanted, proving the claim. $\qquad\square$

**Proposition A.7.** *If $X$ is conditionally independent of $Y$ given $\mathcal{H}$ then*

$$
\mathbb{E}[XY \mid \mathcal{H}] = \mathbb{E}[X \mid \mathcal{H}]\,\mathbb{E}[Y \mid \mathcal{H}].
$$

*Proof.* From Proposition A.4 we know that $\mathbb{E}[X \mid \boldsymbol{\sigma}(Y, \mathcal{H})] = \mathbb{E}[X \mid \mathcal{H}]$. Therefore,

$$
\begin{aligned}
\mathbb{E}\left[XY \mid \mathcal{H}\right] &= \mathbb{E}\left[\mathbb{E}\left[XY \mid \boldsymbol{\sigma}(Y, \mathcal{H})\right] \mid \mathcal{H}\right] && \text{(tower property)} \\
&= \mathbb{E}\left[\mathbb{E}\left[X \mid \boldsymbol{\sigma}(Y, \mathcal{H})\right] Y \mid \mathcal{H}\right] && (Y \in \boldsymbol{\sigma}(Y, \mathcal{H})) \\
&= \mathbb{E}\left[\mathbb{E}\left[X \mid \mathcal{H}\right] Y \mid \mathcal{H}\right] && \text{(Proposition A.4)} \\
&= \mathbb{E}\left[X \mid \mathcal{H}\right]\mathbb{E}\left[Y \mid \mathcal{H}\right], && (\mathbb{E}\left[X \mid \mathcal{H}\right]\, \mathcal{H}\text{-mbl.})
\end{aligned}
$$

as wanted. $\qquad\square$

**Proposition A.8.** *If $X$ is independent of $\boldsymbol{\sigma}(\mathcal{G}, \mathcal{H})$ then $X$ is conditionally independent of $\mathcal{G}$ given $\mathcal{H}$.*

*Proof.* Follows directly from Proposition A.5. $\qquad\square$

**Proposition A.9.** *Let $X, Y, Z$ be random variables such that $X$ and $Y$ are conditionally independent given $\boldsymbol{\sigma}(Z)$. Then, for any measurable function $\phi$ we have*

$$
\mathbb{E}\left[\phi(X, Y) \mid \boldsymbol{\sigma}(Z, Y)\right] = \mathbb{E}\left[\phi(X, y) \mid \boldsymbol{\sigma}(Z)\right]\big|_{y=Y}.
$$

We follow the proof of the somewhat less general result Durrett (2010, Lemma 6.2.1).

*Proof.* By its definition, it is clear that the r.h.s. is $\boldsymbol{\sigma}(Y, Z)$-measurable. Hence, by the definition of the conditional expectation, we only need to show that

$$
\mathbb{E}\left[\phi(X, Y)\mathbb{1}_C\right] = \mathbb{E}\left[\mathbb{E}\left[\phi(X, y) \mid \boldsymbol{\sigma}(Z)\right]\big|_{y=Y}\mathbb{1}_C\right], \tag{18}
$$

for all $C \in \boldsymbol{\sigma}(Y, Z)$.

Assume first that $\phi(x, y) = \mathbb{1}_A(x)\mathbb{1}_B(y)$ and let $C \in \boldsymbol{\sigma}(Y, Z)$ be of the form $C = \{Y \in C_1, Z \in C_2\}$ for Borel sets $A, B, C_1, C_2$ in the images of $X, Y, Z$, respectively. Then,

$$
\begin{aligned}
\mathbb{E}\left[\phi(X, Y)\mathbb{1}_C\right] &= \mathbb{E}\left[\mathbb{E}\left[\mathbb{1}_A(X)\mathbb{1}_B(Y)\mathbb{1}_{C_1}(Y)\mathbb{1}_{C_2}(Z) \mid \boldsymbol{\sigma}(Z)\right]\right] \\
&= \mathbb{E}\left[\mathbb{1}_{C_2}(Z)\mathbb{E}\left[\mathbb{1}_A(X) \mid \boldsymbol{\sigma}(Z)\right]\mathbb{E}\left[\mathbb{1}_B(Y)\mathbb{1}_{C_1}(Y) \mid \boldsymbol{\sigma}(Z)\right]\right],
\end{aligned} \tag{19}
$$

using the tower property in the first and conditional independence in the second equality. Moreover, note that

$$
\mathbb{E}\left[\phi(X, y) \mid \boldsymbol{\sigma}(Z)\right] = \mathbb{E}\left[\mathbb{1}_A(X)\mathbb{1}_B(y) \mid \boldsymbol{\sigma}(Z)\right] = \mathbb{1}_B(y)\mathbb{E}\left[\mathbb{1}_A(X) \mid \boldsymbol{\sigma}(Z)\right],
$$

hence,

$$
\begin{aligned}
\mathbb{E}\left[\mathbb{E}\left[\phi(X,y) \mid \boldsymbol{\sigma}(Z)\right]\big|_{y=Y} \mathbb{1}_C\right] &= \mathbb{E}\left[\mathbb{1}_B(Y)\,\mathbb{E}\left[\mathbb{1}_A(X) \mid \boldsymbol{\sigma}(Z)\right]\,\mathbb{1}_C\right] \\
&= \mathbb{E}\left[\mathbb{E}\left[\mathbb{1}_B(Y)\,\mathbb{E}\left[\mathbb{1}_A(X) \mid \boldsymbol{\sigma}(Z)\right]\,\mathbb{1}_{C_1}(Y)\,\mathbb{1}_{C_2}(Z) \mid \boldsymbol{\sigma}(Z)\right]\right] \qquad (20) \\
&= \mathbb{E}\left[\mathbb{E}\left[\mathbb{1}_A(X) \mid \boldsymbol{\sigma}(Z)\right]\,\mathbb{1}_{C_2}(Z)\,\mathbb{E}\left[\mathbb{1}_B(Y)\,\mathbb{1}_{C_1}(Y) \mid \boldsymbol{\sigma}(Z)\right]\right],
\end{aligned}
$$

using the tower property in the second and measurability in the third equality. (19) and (20) together show that (18) holds for the simple functions and the special $C$. Next, we note that by Dynkin's $\pi$-$\lambda$ theorem, the same holds for general $C \in \boldsymbol{\sigma}(Y,Z)$ (cf. proof of Proposition A.1). Then the claim follows for all bounded functions by the monotone class theorem (Durrett, 2010, Theorem 6.1.3), as outlined in the proof of Durrett (2010, Lemma 6.2.1). Moreover, it follows for general measurable functions by dominated convergence. $\qquad \square$

## B    Inductive Bias

In this section we first discuss the inductive bias induced by the old loss function. Afterwards we discuss the general inductive bias introduced by the PD-NJ-ODE (2). In this paper, we have shown that under certain assumptions our estimator based on the new loss (10) is consistent, and thus asymptotically unbiased as the model complexity $m$ and the number of training paths $N$ tend to infinity. However in such an infinite hypothesis space it is impossible to obtain an unbiased estimator for any finite number of training paths $N \in \mathbb{N}$. For this reason, it is helpful to have an appropriate inductive bias that guides the estimator in the right direction when there is a limited amount of training data (Mitchell, 1980).[11]

### B.1    Inductive Bias of the Loss

While the new loss (10) is asymptotically unbiased, the old loss (3) is in general always biased. Even in the limit $m, N \to \infty$ this bias induced by the old loss (3) does not vanish. On average, models obtained from the old loss will jump too closely to new observation $O_{t_k}$ at observation times $t_k$ as can be observed in the right subplot of Figure 1. However, in the case of no noise (i.e., $X_{t_k} = O_{t_k}$), the correct model should always exactly jump to new observations. Hence, teaching the model explicitly to jump to new observations via the term $|M_k \odot (X_{t_k} - Z_{t_k})|_2$ in the old loss (3) incorporates helpful prior knowledge which is particularly helpful for small training datasets. Even if the observations $O_{t_k}$ are noisy (with a small noise-scale $\sqrt{\mathbb{E}[\epsilon_k^2]}$), the term $|M_k \odot (O_{t_k} - Z_{t_k})|_2$ in the old loss (3) can be a beneficial inductive bias, if one only has access to a very small training dataset. If one has very little information from other observations about the dynamics of $X$, then it might be a reasonable best guess to jump close to new observations $O_{t_k}$ at observation times $t_k$. For example, if the model $Y_{t_k}^{\theta_{m,N}^{\min}}(\tilde{X}^{\leq t_{k-1}})$ obtained from the new loss (10) is further away from the true conditional expectation $\mathbb{E}[X_t \mid \mathcal{A}_{t_{k-1}}]\big|_{t=t_k}$ than the noisy observation $O_{t_k}$, the term $|M_k \odot (O_{t_k} - Z_{t_k})|_2$ in the old loss (3) would be helpful to push the model closer to the truth. This explains why we can see in Figure 3 that the old loss (3) outperforms the new loss (10) in the case of very small noise relative to the overall variability of the data.[12]

Furthermore, different weighting schemes in the loss also influence the inductive bias (see Remark 4.10). The term $|M_k \odot (O_{t_k} - Z_{t_k})|_2$ in the old loss (3) can be particularly helpful if the loss is weighted via $\Delta t_k$.

---

[11]Occam's razor suggests, that one should always pick the simplest (or most parsimonious) model that explains the data, i.e., the inductive bias should be directed towards simplicity/parsimony. Different machine learning (ML) models incorporate simplicity/parsimony in various different ways, so one should pick the right model according to the alignment of the inductive bias of the model with one's prior belief.

[12]In this experiment, we trained also only for 100 epochs. Additionally to the statistical learning argument given before, there is probably also an algorithmic effect, that the term $|M_k \odot (O_{t_k} - Z_{t_k})|_2$ in the old loss (3) helps the model to move directly in the right direction starting from the first epochs. At least during the first epochs the model $Y_{t_k}^{\theta}(\tilde{X}^{\leq t_{k-1}})$ is usually very far off (at observation times $t_k$) compared to the noisy $O_{t_k}$ observations thus the term $|M_k \odot (O_{t_k} - Z_{t_k})|_2$ in the old loss (3) pushes the model directly in the right direction at observation times, while the new loss (10) only has an indirect affect on the model at the observation times $t_k$.

## B.2 Inductive Bias of the PD-NJ-ODE architecture

As shown in our main theorem the PD-NJ-ODE architecture (2) is universal and when it is trained with the right loss, such as (10), it is asymptotically unbiased. However, for any finite number of training paths $N$, there are infinitely many models (i.e., infinitely $\theta \in \Theta$) that all perfectly fit the data, i.e., achieve $\hat{\Phi}_N(\theta) = 0$, but correspond to very different predictions $Y_t^\theta(\tilde{X}^{\leq s})$ on unseen test data. Therefore, for $N < \infty$, our architecture (including the training algorithm) necessarily needs to have an appropriate inductive bias to make a reasonable choice among all the models with reasonable low training loss $\hat{\Phi}_N(\theta)$.[13] (Additionally we want this bias to vanish in the limit $N \to \infty$, which is already given by our theory.) First, we give some intuition on our understanding of the important inductive bias of PD-NJ-ODE (2) and afterwards we will give some theoretical arguments that motivate these hypotheses. Most machine learning models have and inductive bias towards some form of simplicity for the function mapping inputs to predictions. E.g., some NN-architectures have an inductive bias towards low second derivative of this function w.r.t. the inputs (Savarese et al., 2019; Ongie et al., 2019; Heiss et al., 2019; 2023; 2022; Parhi & Nowak, 2022) (i.e., the function can still be universal but among all functions with equal loss the one which is "most linear" is chosen). The crucial difference of PD-NJ-ODE (2) is that its inductive bias favours simplicity of the functions $f_{\theta_1}, \rho_{\theta_2}, \tilde{g}_{\tilde{\theta}_3}$ instead of simply favouring simplicity of the map $(\tilde{X}^{\leq s}, t) \mapsto Y_t^\theta(\tilde{X}^{\leq s})$. For many applications of PD-NJ-ODEs, a linear ODE is intuitively one of the most simple/parsimonious/conservative/plausible models possible. The following example illustrates the benefit of the inductive bias of PD-NJ-ODEs.

**Example B.1.** *Consider the situation where the true underlying process of $X$ follows a linear SDE (driven by a Brownian motion, i.e., a linear Itô-diffusion), then the conditional expectation $\hat{X}$ follows a linear ODE (except the jumps at observation times $t_k$, cf. Krach et al. (2022, Example B.2)). For simplicity consider the case of complete noiseless observations $X_{t_k}$ (i.e., $M = 1$ and $\epsilon = 0$). Then $\hat{X}$ can be perfectly expressed by our PD-NJ-ODE (2) with 3 linear functions $f_{\theta_1}, \rho_{\theta_2}, \tilde{g}_{\tilde{\theta}_3}$ instead of simply favouring simplicity of the map $(\tilde{X}^{\leq s}, t) \mapsto Y_t^\theta(\tilde{X}^{\leq s})$. E.g., $f_{\theta_1}$ is the right-hand side of the ODE resulting in $\hat{X}$, $\rho_{\theta_2}$ maps new observations $X_{t_k}$ simply to $X_{t_k}$,[14] and $\tilde{g}_{\tilde{\theta}_3}$ is also simply the identity. As these 3 functions are linear, their second derivative is zero. Hence, for example, the inductive bias of L2-regularized ReLU-NNs $f_{\theta_1}, \rho_{\theta_2}, \tilde{g}_{\tilde{\theta}_3}$ strongly favours such simple linear functions. Therefore, the model can learn these linear dynamics already from a relatively small amount of training paths and extrapolate well far into the future. On the other hand, in this setting the map $(\tilde{X}^{\leq s}, t) \mapsto \hat{X}_t(\tilde{X}^{\leq s})$ grows exponentially in the input $t$ and thus has huge (second) derivative with respect to $t$, which is considered the opposite of simple by most of the commonly used ML-models. Hence, if one trains, on the same dataset, a feed-forward neural network $\check{F}_\theta : (\tilde{X}^{\leq s}, t) \mapsto \check{F}_\theta(\tilde{X}^{\leq s}, t)$ that directly approximates $F$, the model would find many other "simpler"/flatter functions $\check{F}_\theta$ that fit the finite training dataset equally well. Even for larger $N$, such a model, favouring a low Sobolev-norm of the map $\check{F}_\theta$, would always have troubles to make predictions far ahead into the future $t \gg \tau(t)$: For values of $t - \tau(t)$ larger than the largest $t_k - t_{k-1}$ ever observed in training, such a model has no reason to keep growing exponentially but would rather extrapolate as flat as possible. In contrast, $Y_t^\theta(\tilde{X}^{\leq \tau(t)})$, obtained from our PD-NJ-ODE with linear functions $f_{\theta_1}, \rho_{\theta_2}, \tilde{g}_{\tilde{\theta}_3}$, would keep the exponential growth in $t$ for arbitrarily large values of $t - \tau(t)$ even far out of sample and even far out of distribution.[15] In line with this, it is expected that the PD-NJ-ODE leads to better results than the model $\check{F}_\theta$, when evaluated on previously unseen test data. This was confirmed by the experiment in Krach et al. (2022, Appendix E).*

To summarize: While the PD-NJ-ODE (2) is able to learn arbitrarily complicated dynamics (fulfilling some mild regularity assumptions), it favours models $\theta$ with "simple" functions $f_{\theta_1}, \rho_{\theta_2}, \tilde{g}_{\tilde{\theta}_3}$ among different models with equal training loss $\hat{\Phi}_N(\theta)$, where the definition of "simplicity" depends on the architectures used for $f_{\theta_1}, \rho_{\theta_2}, \tilde{g}_{\tilde{\theta}_3}$. E.g., for L2-regularized ReLU-NNs the notion of simplicity relates to the second derivative of

---

[13]If the underlying process $X$ is not deterministic or if the observations are noisy, even the exact conditional expectation does not fit the observations $O_{t_k}$ exactly (i.e., $\Psi(\hat{X}) > 0$) due to stochasticity. Hence, the model has to learn to average out the noise. Therefore, we do not only need to choose among models that achieve $\hat{\Phi}_N(\theta) = 0$, but even among a wider range of models achieving $\hat{\Phi}_N(\theta) < c$.

[14]Note that the the first $d_X$ signatures of level 1 of $\tilde{X}^{\leq t_k} - X_0$ are exactly $X_{t_k} - X_0$. Thus, the function $\rho_{\theta_2}$ defined by $\rho_{\theta_2}\left(H_{t-}, t, \pi_m(\tilde{X}^{\leq \tau(t)} - X_0), X_0\right) = X_{t_k}$ is linear.

[15]For our theoretical result, we assumed $f_{\theta_1}$ to be bounded, which would not enable this exponential growth arbitrarily far into the future, but at least for a very long time if $\gamma_1$ is large enough.

$f_{\theta_1}, \rho_{\theta_2}, \tilde{g}_{\tilde{\theta}_3}$ (Savarese et al., 2019; Ongie et al., 2019; Heiss et al., 2019; 2023; 2022; Parhi & Nowak, 2022). This can lead to better generalization performance.

### B.2.1 Inductive Bias towards Multi-task Learning

If one trains multiple tasks together (i.e., if one wants to forecast a multi-dimensional process $X$), many deep learning applications have shown that an inductive bias towards functions that use similar features among multiple tasks (rather than using very different features for each of these tasks) can be highly beneficial. This phenomena is known as *multi-task learning* (Caruana et al., 1997). One can argue that multi-task learning (highly related to feature learning, representation learning, transfer learning, metric learning) is one of the biggest strengths of deep learning models compared to shallow learning models in terms of generalization (Heiss et al., 2022). The inductive bias of our PD-NJ-ODE (2) towards feature sharing (i.e., multi-task learning) is twofold:

1. Per definition the hidden state $H_t$ in (2) is shared among all tasks/outputs. I.e., each coordinate of the output $Y_t$ is a function of the same hidden state $H_t$. Therefore, the hidden state $H_t$ becomes more parsimonious, if it mainly contains features that are helpful for all coordinates of $Y_t$.

2. If the neural network architectures of $f_{\theta_1}, \rho_{\theta_2}, \tilde{g}_{\tilde{\theta}_3}$ contain trainable parameters in multiple hidden layers, then these functions favour feature sharing within their hidden layers, as shown by Heiss et al. (2022) in the case of L2-regularization[16]. Caruana et al. (1997); Heiss et al. (2022) provide more intuition on multi-task learning.

### B.2.2 Implicit Regularization

Even without any explicit L2-regularization and even for very large theoretical model complexity $m$ (i.e., large $\Theta_m$ in our setting), it is widely assumed that (stochastic) gradient descent (initialized close to zero) induces an implicit inductive bias towards simplicity (Neyshabur, 2017). In some settings this implicit regularization is extremely similar (or even exactly identical) to explicit L2-regularization (Heiss et al., 2019) and in other settings it can be (slightly) different. In our setting of training PD-NJ-ODEs, implicit regularization of the gradient descent does most probably not exactly correspond to explicit L2-regularization, but on a high level the qualitative behaviour might be similar.[17]

In theory, $m$ could also be too large in relation to $N$, since too large $m$ allows for over-fitting. In contrast to this, $N$ can never be too large. In particular, if one picks an $\theta_{m,N}^{\min} \in \Theta_{m,N}^{\min}$ with unnecessarily large L2-norm over-fitting can be a serious problem. However, in practice, too many neurons are usually not a problem due to implicit regularization of (stochastic) gradient descent as suggested empirically by Herrera et al. (2021, Figure 6) and theoretically by Neyshabur (2017); Savarese et al. (2019); Ongie et al. (2019); Heiss et al. (2019; 2023; 2022); Parhi & Nowak (2022).

### B.2.3 Mathematical Theory on the Inductive Bias of PD-NJ-ODEs

For implicit regularization it seems difficult to get precise global results for PD-NJ-ODEs. However, for explicit L2-regularization of $\theta$, there is much more hope to get nice theoretical results on the inductive bias of the PD-NJ-ODEs (2). (Note that the bound $m$ on the L2-norm of $\theta$ in the definition of $\Theta_m$ corresponds to L2-regularization in a Lagrangian sense.[18]) The results by Savarese et al. (2019); Ongie

---

[16]In the case of L2-regularization Heiss et al. (2022) shows (for certain architectures) that this multi-task learning effect does not vanish when the width of the network increases, while it would vanish with increasing width for some specific other training methods. For most deep learning methods used in practice, there is some multi-task learning effect assumed to be present even if it is theoretically not always as well understood as for L2-regularized wide ReLU-NNs.

[17]Note that a classical gradient flow is always identical to a gradient flow with respect to the L2 norm and thus *locally* moves as little w.r.t. the L2-distance as possible. Therefore, it seems plausible that gradient descent does not increase the L2-norm of $\theta$ *much* more than necessary to achieve a certain training loss.

[18]The L2-regularization regularization parameter $\lambda_{\text{reg}}$ can be seen as a KKT-multiplier of the constraint $|\theta|_2 < m$. However, if $m$ is too large the constraint is not active and $\theta_{m,N}^{\min}$ is highly undetermined. For L2-regularization with $\lambda_{\text{reg}} > 0$, the *parameters* $\theta_{m,N}^{\min}$ are usually also not uniquely defined, but the solution $Y^{\theta_{m,N}^{\min}}$ can be unique.

et al. (2019); Heiss et al. (2019; 2023; 2022); Parhi & Nowak (2022) strongly support the hypothesis that $\text{argmin}_\theta \left( \hat{\Phi}_N(\theta) + \lambda_{\text{reg}} |\theta|_2^2 \right)$ exactly converges to $\text{argmin}_{f,\rho,\tilde{g}} \left( \hat{\Psi}_N(Y^{f,\rho,\tilde{g}}) + \lambda_{\text{reg}} \left( P_1(f) + P_2(\rho) + P_3(\tilde{g}) \right) \right)$, where $P_i$ describe some regularization functionals (such as weighted $L^p$ norms of the second derivative) depending on the exact architectures used for $f, \rho, \tilde{g}$ given in Savarese et al. (2019); Ongie et al. (2019); Heiss et al. (2019; 2023; 2022); Parhi & Nowak (2022). However, (for rather technical reasons) their proofs do not directly apply to PD-NJ-ODEs. The formulation of the the main theorem in Heiss et al. (2023) could be applied more directly to PD-NJ-ODE by considering the map $(f, \rho, \tilde{g}) \mapsto \hat{\Psi}_N(Y^{f,\rho,\tilde{g}})$ as the loss functional.[19] Proofing such results rigorously would be interesting future work.

## C  Combining the Two Extensions with Full Proof

In this section we prove convergence of the PD-NJ-ODE to the optimal prediction in the most general setting that allows for non-Markovian processes with irregular incomplete noisy observations, where dependence between the observation framework and the process is possible.

### C.1  Setting

As in Section 4.2 we consider only the probability space $(\Omega, \mathcal{F}, \mathbb{F}, \mathbb{P})$ on which we define $X, n, K, t_i, \tau, M$. As in Section 3.1, we additionally define the observation noise $(\epsilon_k)_{0 \leq k \leq K}$, the noisy observations $O_{t_k} := X_{t_k} + \epsilon_k$ for $0 \leq k \leq n$, the filtration of the currently available information via

$$\mathcal{A}_t := \boldsymbol{\sigma} \left( O_{t_i, j}, t_i, M_{t_i} | t_i \leq t, j \in \{1 \leq l \leq d_X | M_{t_i, l} = 1\} \right),$$

$\tilde{O}^{\leq t}$ and the functions $F_j$ such that $\hat{X}_{t,j} = F_j(t, \tau(t), \tilde{O}^{\leq \tau(t)})$. Then we make the following assumptions.

**Assumption C.1.** *We assume that:*

   (i) $M_{0,j} = 1$ for all $1 \leq j \leq d_X$ *(X is completely observed at 0) and* $|M_k|_1 > 0$ *for every* $1 \leq k \leq K$ $\mathbb{P}$-almost surely *(at every observation time at least one coordinate is observed).*

   (ii) *The probability that any two observation times are closer than* $\epsilon > 0$ *converges to 0 when* $\epsilon$ *does, i.e., if* $\delta(\omega) := \min_{0 \leq i \leq n(\omega)} |t_{i+1}(\omega) - t_i(\omega)|$ *then* $\lim_{\epsilon \to 0} \mathbb{P}(\delta < \epsilon) = 0$.

   (iii) *Almost surely* $X$ *is not observed at a jump, i.e.,* $\mathbb{P}(t_i \in \mathcal{J} \mid i \leq n) = \mathbb{P}(\Delta X_{t_i} \neq 0 \mid i \leq n) = 0$ *for all* $1 \leq i \leq K$.

   (iv) $F_j$ *are continuous and differentiable in their first coordinate* $t$ *such that their partial derivatives with respect to* $t$, *denoted by* $f_j$, *are again continuous and there exists a* $B > 0$ *and* $p \in \mathbb{N}$ *such that for every* $t \in [0, T]$ *the functions* $f_j, F_j$ *are polynomially bounded in* $X^\star$, *i.e.,*

$$|F_j(\tau(t), \tau(t), \tilde{O}^{\leq \tau(t)})| + |f_j(t, \tau(t), \tilde{O}^{\leq \tau(t)})| \leq B(X_t^\star + 1)^p + B \sum_{i=0}^n |\epsilon_i|.$$

   (v) $X^\star$ *is* $L^{2p}$-integrable, i.e., $\mathbb{E}[(X_T^\star)^{2p}] < \infty$.

   (vi) $n$ *is square-integrable, i.e.,* $\mathbb{E}[n^2] < \infty$.

   (vii) *The i.i.d.* $\epsilon_k$ *are independent of* $X, n, M, (t_i)_{1 \leq i \leq K}$, *are centered and square-integrable, i.e.,* $\mathbb{E}[\epsilon_k] = 0$ *and* $\mathbb{E}[|\epsilon_k|^2] < \infty$.

   (viii) *For every* $1 \leq i \leq n$, $X_{t_i-}$ *is conditionally independent of* $\boldsymbol{\sigma}(n, M_{t_i})$ *given* $\mathcal{A}_{t_i-}$.[20]

---

[19] One would need to check if this loss functional meets the assumptions of Heiss et al. (2023) for a rigorous proof.

[20] More precisely, one should formulate this assumption as "For every $1 \leq i \leq K$, $X_{t_i-} \mathbb{1}_{\{i \leq n\}}$ is conditionally independent of $\boldsymbol{\sigma}(n, M_{t_i})$ given $\mathcal{A}_{t_i-}$."

(ix) *For all $1 \leq k \leq K$, $1 \leq j \leq d_X$ there is some $\eta_{k,j} > 0$ such that $\mathbb{P}(M_{k,j} = 1 \mid \boldsymbol{\sigma}(n, \mathcal{A}_{t_k-})) > \eta_{k,j}$ (i.e., given the currently known information and $n$, for each coordinate the probability of observing it at the next observation time is positive).*

(x) *We assume that for every $1 \leq k \leq K$ the process $X$ is conditionally independent of $t_k$ given $\mathcal{A}_{t_{k-1}}$.*

**Remark C.2.** *All the relaxations and extensions discussed in Sections 2 to 5 hold also in this combined setting.*

As in Section 3.1, the PD-NJ-ODE uses the noisy observations $O_{t_i}$ and $\tilde{O}^{\leq \tau(t)}$ as inputs instead of $X_{t_i}$ and $\tilde{X}^{\leq \tau(t)}$ and is trained with the objective functions (6) respectively (7) and their Monte Carlo approximations.

## C.2 Convergence Theorem

**Theorem C.3.** *If Assumption C.1 is satisfied and using the definitions of Appendix C.1, the claims of Theorem 2.7 hold equivalently, upon replacing the original loss functions and their Monte Carlo approximations by their noise-adapted versions[21]. In particular, we obtain convergence of our estimator $Y^{\theta_{m,N_m}^{\min}}$ to the true conditional expectation $\hat{X}$ in $d_k$.[22]*

We start with the following result which is a combination of Lemma 3.4 and Lemma 4.6.

**Lemma C.4.** *For any $\mathbb{A}$-adapted process $Z$ it holds that*

$$\mathbb{E}\left[\frac{1}{n}\sum_{i=1}^{n}|M_{t_i} \odot (O_{t_i} - Z_{t_i-})|_2^2\right] = \mathbb{E}\left[\frac{1}{n}\sum_{i=1}^{n}\left|M_{t_i} \odot (O_{t_i} - \hat{X}_{t_i-})\right|_2^2\right] + \mathbb{E}\left[\frac{1}{n}\sum_{i=1}^{n}\left|M_{t_i} \odot (\hat{X}_{t_i-} - Z_{t_i-})\right|_2^2\right].$$

*Proof.* First note that by Assumption C.1 point (iii) we have that $X_{t_i} = X_{t_i-}$ almost surely and when defining $O_{t_i-} := X_{t_i-} + \epsilon_i$ we therefore also have that $O_{t_i} = O_{t_i-}$ almost surely. Next notice that Assumption C.1 point (vii) & (viii) together with Proposition A.6 imply that $O_{t_i-}$ is conditionally independent of $\boldsymbol{\sigma}(n, M_i)$ given $\mathcal{A}_{t_i-}$. Hence, for $\hat{O}_{t_i-} := \mathbb{E}[O_{t_i-} \mid \mathcal{A}_{t_i-}]$ it follows as in Lemma 4.6 that

$$\mathbb{E}\left[\frac{1}{n}\sum_{i=1}^{n}|M_{t_i} \odot (O_{t_i-} - Z_{t_i-})|_2^2\right] = \mathbb{E}\left[\frac{1}{n}\sum_{i=1}^{n}\left|M_{t_i} \odot (O_{t_i-} - \hat{O}_{t_i-})\right|_2^2\right] + \mathbb{E}\left[\frac{1}{n}\sum_{i=1}^{n}\left|M_{t_i} \odot (\hat{O}_{t_i-} - Z_{t_i-})\right|_2^2\right].$$

Then we can conclude the proof as in Lemma 3.4, by noting that with Proposition 4.5 we obtain,

$$\hat{O}_{t_i-} = \hat{X}_{t_i-} + \mathbb{E}[\epsilon_i \mid \mathcal{A}_{t_i-}] = \hat{X}_{t_i-} + \mathbb{E}[\epsilon_i] = \hat{X}_{t_i-},$$

using that $\epsilon_i$ has expectation 0 and is independent of $\mathcal{A}_{t_i-}$. $\qquad\square$

In the following, we use the notation $\Theta_m^i$ and $\tilde{\Theta}_m^i$ if we speak of the projections of the sets on the weights $\theta_i$ and $\tilde{\theta}_i$ respectively.

*Proof of Theorem C.3 – Part 1.* We start by showing that $\hat{X} \in \mathbb{D}$ is the unique minimizer of $\Psi$ up to indistinguishability (as defined in Definition 2.4). Note that for every $t_i$ we have $M_{t_i} \odot \hat{X}_{t_i} = M_{t_i} \odot X_{t_i}$ and that $X_{t_i} = X_{t_i-}$ if $t_i \notin \mathcal{J}$, hence with probability 1. It follows directly from Lemma C.4 that $\hat{X}$ is a minimizer of $\Psi$, since

$$\Psi(Z) = \Psi(\hat{X}) + \mathbb{E}\left[\frac{1}{n}\sum_{i=1}^{n}\left|M_{t_i} \odot (\hat{X}_{t_i-} - Z_{t_i-})\right|_2^2\right]. \tag{22}$$

---

[21]In particular, we replace the original empirical loss $\hat{\Phi}_N(\theta) = (5)$ by the noise-adapted empirical loss function,

$$\hat{\Phi}_N(\theta) := \frac{1}{N}\sum_{j=1}^{N}\frac{1}{n^{(j)}}\sum_{i=1}^{n^{(j)}}\left|M_i^{(j)} \odot \left(O_{t_i^{(j)}}^{(j)} - Y_{t_i^{(j)}-}^{\theta,j}\right)\right|_2^2. \tag{21}$$

[22]For a convergence result at times $t$ that are not observation times, see Section 5.2.

Before we can show uniqueness of $\hat{X}$, we need some additional results. For those, let $Z \in \mathbb{D}$. Let $c_1 :=$ $\mathbb{E}\left[n\right]^{1/2} \in (0, \infty)$, then the Hölder inequality, together with the fact that $n \geq 1$, yields

$$\mathbb{E}\left[|Z|_2\right] = \mathbb{E}\left[\frac{\sqrt{n}}{\sqrt{n}}|Z|_2\right] \leq c_1 \mathbb{E}\left[\frac{1}{n}|Z|_2^2\right]^{1/2}. \tag{23}$$

By Lemma 4.8 and by the equivalence of 1- and 2-norm we have for some constant $c_3 > 0$ and for any $1 \leq k \leq K$

$$\mathbb{E}\left[\mathbb{1}_{\{n \geq k\}}\left|\hat{X}_{t_k-} - Z_{t_k-}\right|_2\right] \leq \frac{c_3}{c_2(k)}\mathbb{E}\left[\mathbb{1}_{\{n \geq k\}}\left|M_{t_k} \odot (\hat{X}_{t_k-} - Z_{t_k-})\right|_2\right]. \tag{24}$$

To see that $\hat{X}$ is the unique minimiser up to indistinguishability, let $Z \in \mathbb{D}$ be a process which is not indistinguishable from $\hat{X}$. Hence, there exists some $1 \leq k \leq K$ such that $d_k(\hat{X}, Z) > 0$. We have

$$\begin{aligned}
\mathbb{E}\left[\frac{1}{n}\sum_{i=1}^{n}\left|M_{t_i} \odot (\hat{X}_{t_i-} - Z_{t_i-})\right|_2^2\right] &= \mathbb{E}\left[\frac{1}{n}\sum_{i=1}^{K}\mathbb{1}_{\{n \geq i\}}\left|M_{t_i} \odot (\hat{X}_{t_i-} - Z_{t_i-})\right|_2^2\right] \\
&\geq \mathbb{E}\left[\frac{1}{n}\mathbb{1}_{\{n \geq k\}}\left|M_{t_k} \odot (\hat{X}_{t_k-} - Z_{t_k-})\right|_2^2\right] \\
&\geq c_1^{-2}\mathbb{E}\left[\mathbb{1}_{\{n \geq k\}}\left|M_{t_k} \odot (\hat{X}_{t_k-} - Z_{t_k-})\right|_2\right]^2 \\
&\geq \left(\frac{c_2}{c_1 c_3}\right)^2\mathbb{E}\left[\mathbb{1}_{\{n \geq k\}}\left|\hat{X}_{t_k-} - Z_{t_k-}\right|_2\right]^2 \\
&= \left(\frac{c_2}{c_0 c_1 c_3}\right)^2 d_k(\hat{X}, Z)^2 > 0,
\end{aligned} \tag{25}$$

where we used (23) for the 3rd, (24) for the 4th and (1) for the last line. Together with (22) this implies $\Psi(Z) > \Psi(\hat{X})$.

Next we show that (2) can approximate $\hat{X}$ arbitrarily well. Since the dimension $d_H$ can be chosen freely, let us fix it to $d_H := d_X$. Furthermore, let us fix $\tilde{\theta}_3^\star$ such that $\tilde{g}_{\tilde{\theta}_3^\star} = \mathrm{id}$, which is possible since we assumed that $\mathrm{id} \in \tilde{\mathcal{N}}$. Let $\varepsilon > 0$, $N_\varepsilon := \lceil 2(T+1)\varepsilon^{-4}\rceil$ (implying that $\lim_{\varepsilon \to 0} N_\varepsilon = \infty$) and $\mathcal{P}_\varepsilon \subseteq \mathrm{BV}^c([0, T])$ be the closure of the set $A_{N_\varepsilon}$ of Krach et al. (2022, Remark 3.11), which is compact. For any $1 \leq j \leq d_X$, the function $f_j$ is continuous by Assumption C.1 and can (by abuse of notation) equivalently be written as a (continuous) function $f_j(t, \tau(t), \tilde{O}^{\leq \tau(t)} - O_0, O_0)$. Therefore, Krach et al. (2022, Proposition 3.8) implies that there exists an $m_0 = m_0(\varepsilon) \in \mathbb{N}$ and a continuous function $\hat{f}_j$ such that

$$\sup_{(t, \tau, X) \in [0, T]^2 \times \mathcal{P}_\varepsilon}\left|f_j(t, \tau, X) - \hat{f}_j(t, \tau, \pi_{m_0}(X - X_0), X_0)\right| \leq \varepsilon/2.$$

Since the variation of functions in $\mathcal{P}_\varepsilon$ is uniformly bounded by a finite constant, the set of their truncated signatures $\pi_{m_0}(\mathcal{P}_\varepsilon)$ is a bounded subset in $\mathbb{R}^d$ for some $d \in \mathbb{N}$ (depending on $d_X$ and $m_0$), hence its closure, denoted by $\Pi_\varepsilon$, is compact. Therefore, the universal approximation theorem for neural networks (Hornik et al., 1989, Theorem 2.4) implies that there exists an $m_1 = m_1(\varepsilon) \in \mathbb{N}$ and neural network weights $\tilde{\theta}_1^{\star, m_1} \in \tilde{\Theta}_{m_1}^1$ such that for every $1 \leq j \leq d_X$ the function $\hat{f}_j$ is approximated up to $\varepsilon/2$ by the $j$-th coordinate of the neural network $\tilde{f}_{\tilde{\theta}_1^{\star, m_1}} \in \tilde{\mathcal{N}}$ (denoted by $\tilde{f}_{\tilde{\theta}_1^{\star, m_1}, j}$) on the compact set $[0, T]^2 \times \Pi_\varepsilon$. Hence, combining the two approximations we get (by triangle inequality)

$$\sup_{(t, \tau, X) \in [0, T]^2 \times \mathcal{P}_\varepsilon}\left|f_j(t, \tau, X) - \tilde{f}_{\tilde{\theta}_1^{\star, m_1}, j}(t, \tau, \pi_{m_0}(X - X_0), X_0)\right| \leq \varepsilon.$$

Obviously, extending the input of the neural network does not make the approximation worse, by simply setting the corresponding weights to 0, hence, also $H_{t-}$ can be used as additional input. Similarly we get that there exists an $m_2 = m_2(\varepsilon) \in \mathbb{N}$ and neural network weights $\tilde{\theta}_2^{\star, m_2} \in \tilde{\Theta}_{m_2}^2$ such that for every $1 \leq j \leq d_X$

$$\sup_{(t, X) \in [0, T] \times \mathcal{P}_\varepsilon}\left|F_j(t, t, X) - \tilde{\rho}_{\tilde{\theta}_2^{\star, m_2}, j}(t, \pi_{m_1}(X - X_0), X_0)\right| \leq \varepsilon.$$

As before, $H_{t-}$ can be used as additional input without worsening the approximation.

Next we define the bounded output neural networks based on these neural networks. For this let us define

$$\gamma_1 := \max_{(t,\tau,X)\in[0,T]^2\times\mathcal{P}_\varepsilon} \left| \tilde{f}_{\tilde{\theta}_1^{\star,m_1}}(t,\tau,\pi_{m_0}(X-X_0),X_0) \right|$$

and $\gamma_2$ equivalently for $\tilde{\rho}_{\tilde{\theta}_2^{\star,m_2}}$. Since the neural networks are continuous functions they take a finite maximum on the compact sets, hence $\gamma_1,\gamma_2$ are finite. Then we define the bounded output neural networks $f_{\theta_1^{\star,m_1}}, \rho_{\theta_2^{\star,m_2}} \in \mathcal{N}$ with $\theta_i^{\star,m_i} := (\tilde{\theta}_i^{\star,m_i},\gamma_i)$. Clearly, these bounded output neural networks coincide with the neural networks on the compact sets. Therefore, they satisfy the same $\varepsilon$-approximation and since $F_j, f_j$ are bounded by $U := B\left((X_T^\star+1)^p + \sum_{i=0}^n |\epsilon_i|\right)$ (Assumption C.1(iv)), it follows that $f_{\theta_1^{\star,m_1}}, \rho_{\theta_2^{\star,m_2}}$ are bounded by $U+\varepsilon$. In particular, we have for $\epsilon < B$ the global bounds $|f_j - f_{\theta_1^{\star,m_1},j}|_\infty \le 3U$ and $|F_j - \rho_{\theta_2^{\star,m_2},j}|_\infty \le 3U$. Setting $m := \max(m_0,m_1,m_2,\gamma_1,\gamma_2,|\tilde{\theta}_i^{\star,m_2}|_2,|\tilde{\theta}_2^{\star,m_2}|_2)$, it follows that $\theta_m^\star := (\theta_1^{\star,m_1},\theta_2^{\star,m_2},\tilde{\theta}_3^\star) \in \Theta_m$.

Now we can bound the distance between $Y_t^{\theta_m^\star}(\tilde{O}^{\le t})$ and $\hat{X}_t$. Whenever $X_T^\star < 1/\varepsilon$, the number of observations satisfies $n < 1/\varepsilon$, the minimal difference between any two consecutive observation times $\delta > \varepsilon$ and all noise terms satisfy $|\epsilon_i|_2 < 1/\varepsilon$ we know that the corresponding path $\tilde{O}^{\le \tau(t)} - O_0$ is an element of $A_{N_\varepsilon}$ and therefore the neural network approximations up to $\varepsilon$ hold. Otherwise, one of those conditions is not satisfied and the global upper bound can be used. Hence, if $t \in \{t_1,\ldots,t_n\}$, we have for $F = (F_j)_{1\le j\le d_X}$ and $f = (f_j)_{1\le j\le d_X}$

$$\left| Y_t^{\theta_m^*} - \hat{X}_t \right|_1 = \left| \rho_{\theta_2^{\star,m_2}}\left(H_{t-},t,\pi_m(\tilde{O}^{\le t}-O_0),O_0\right) - F\left(t,t,\tilde{O}^{\le t}\right) \right|_1$$

$$\le \varepsilon d_X \mathbb{1}_{\{X_T^\star<1/\varepsilon\}}\mathbb{1}_{\{n<1/\varepsilon\}}\mathbb{1}_{\{\delta>\epsilon\}}\mathbb{1}_{\{\forall 0\le i\le n: |\epsilon_i|_2<1/\varepsilon\}}$$

$$+ d_X 3U\left(\mathbb{1}_{\{X_T^\star\ge 1/\varepsilon\}} + \mathbb{1}_{\{n\ge 1/\varepsilon\}} + \mathbb{1}_{\{\delta\le\epsilon\}} + \mathbb{1}_{\{\exists 0\le i\le n: |\epsilon_i|_2\ge 1/\varepsilon\}}\right),$$

and if $t \notin \{t_1,\ldots,t_n\}$,

$$\left| Y_t^{\theta_m^*} - \hat{X}_t \right|_1 \le \left| Y_{\tau(t)}^{\theta_m^*} - \hat{X}_{\tau(t)} \right|_1 + \int_{\tau(t)}^t \left| f_{\theta_1^{\star,m_1}}\left(H_{s-},s,\tau(t),\pi_m(\tilde{O}^{\le\tau(t)}-O_0),O_0\right) - f(s,\tau(t),\tilde{O}^{\le\tau(t)}) \right|_1 ds$$

$$\le \varepsilon(T+1)d_X \mathbb{1}_{\{X_T^\star<1/\varepsilon\}}\mathbb{1}_{\{n<1/\varepsilon\}}\mathbb{1}_{\{\delta>\varepsilon\}}\mathbb{1}_{\{\forall 0\le i\le n: |\epsilon_i|_2<1/\varepsilon\}}$$

$$+ (T+1)d_X 3U\left(\mathbb{1}_{\{X_T^\star\ge 1/\varepsilon\}} + \mathbb{1}_{\{n\ge 1/\varepsilon\}} + \mathbb{1}_{\{\delta\le\varepsilon\}} + \mathbb{1}_{\{\exists 0\le i\le n: |\epsilon_i|_2\ge 1/\varepsilon\}}\right).$$

Moreover, by equivalence of the 1- and 2-norm, there exists a constant $c > 0$ such that for all $t \in [0,T]$

$$\left| Y_t^{\theta_m^*} - \hat{X}_t \right|_2 \le c\,\varepsilon(T+1)d_X + c(T+1)d_X 3U\left(\mathbb{1}_{\{X_T^\star\ge 1/\varepsilon\}} + \mathbb{1}_{\{n\ge 1/\varepsilon\}} + \mathbb{1}_{\{\delta\le\varepsilon\}} + \mathbb{1}_{\{\exists 0\le i\le n: |\epsilon_i|_2\ge 1/\varepsilon\}}\right) =: c_m.$$

So far, we have fixed an $\varepsilon > 0$ and argued that there exists some $m \in \mathbb{N}$ such that the neural network approximation bounds hold with $\varepsilon$-error. However, what we actually need to show is that this error converges to 0 when increasing the truncation level and network size $m$. Therefore, we define $\varepsilon_m \ge 0$ to be the smallest number such that the above bounds hold with error $\varepsilon_m$ when using an architecture with signature truncation level $m \in \mathbb{N}$ and weights in $\Theta_m$. Since increasing $m$ can only make the approximations better (by setting the new weights to 0, the same approximation error as before is achieved, but potentially there exists a better choice), we have $\varepsilon_{m_1} \ge \varepsilon_{m_2}$ for any $m_1 \le m_2$. In particular $(\varepsilon_m)_{m\ge 0}$ is a a decreasing sequence, hence, our derivations before proof that $\lim_{m\to\infty}\varepsilon_m = 0$. In the following we denote by $\theta_m^\star \in \Theta_m$ the best choice for the weights within the set $\Theta_m$ to approximate the functions $F_j, f_j$.

Since $\theta_m^{\min} \in \operatorname{argmin}_{\theta\in\Theta_m}\{\Phi(\theta)\}$ (note that at least one minimum exists in the compact set $\Theta_m$ since $\Phi$ is continuous), we get with Lemma C.4 that

$$\min_{Z\in\mathbb{D}}\Psi(Z) \le \Phi(\theta_m^{\min}) \le \Phi(\theta_m^*) = \mathbb{E}\left[\frac{1}{n}\sum_{i=1}^n \left| M_{t_i}\odot\left(O_{t_i}-Y_{t_i}^{\theta_m^*}\right) \right|_2^2\right]$$

$$= \Psi(\hat{X}) + \mathbb{E}\left[\frac{1}{n}\sum_{i=1}^n \left| M_{t_i}\odot\left(\hat{X}_{t_i-}-Y_{t_i-}^{\theta_m^*}\right) \right|_2^2\right] \le \Psi(\hat{X}) + \mathbb{E}\left[c_m^2\right]. \tag{26}$$

Integrability of $|X^\star_T|_2$, $|n|$ and $|\epsilon_i|$ together with Assumption C.1(ii) on $\delta$ imply that

$$\mathbb{1}_{\{X^\star_T \geq 1/\varepsilon_m\}} + \mathbb{1}_{\{n \geq 1/\varepsilon_m\}} + \mathbb{1}_{\{\delta \leq \varepsilon_m\}} + \mathbb{1}_{\{\exists 0 \leq i \leq n : |\epsilon_i|_2 \geq 1/\varepsilon_m\}} \xrightarrow[m\to\infty]{\mathbb{P}-a.s.} 0.$$

Therefore, we have for a suitable constant $c > 0$ (not depending on $\varepsilon_m$ and $m$),

$$\mathbb{E}\left[c_m^2\right] \leq c\varepsilon_m^2 + c\mathbb{E}\left[U^2\left(\mathbb{1}_{\{X^\star_T \geq 1/\varepsilon_m\}} + \mathbb{1}_{\{n \geq 1/\varepsilon_m\}} + \mathbb{1}_{\{\delta \leq \varepsilon_m\}} + \mathbb{1}_{\{\exists 0 \leq i \leq n : |\epsilon_i|_2 \geq 1/\varepsilon_m\}}\right)\right] \xrightarrow{m\to\infty} 0,$$

by dominated convergence, since $U$ is $L^2$-integrable. Indeed,

$$\mathbb{E}[U^2] \leq 8B^2\mathbb{E}\left[(X^\star_t + 1)^{2p} + n\sum_{i=0}^n |\epsilon_i|^2\right] = 8B^2\left(\mathbb{E}\left[(X^\star_t + 1)^{2p}\right] + \mathbb{E}[n^2]\mathbb{E}\left[|\epsilon_0|^2\right]\right) < \infty, \tag{27}$$

using Cauchy-Schwarz inequality for the first step, that the $\epsilon_i$ are i.i.d. and independent of $n$ for the equality and the integrability of $X^\star$, $\epsilon_0$ and $n^2$ for the upper bound. Using this and $\Psi(\hat{X}) = \min_{Z\in\mathbb{D}} \Psi(Z)$, we get from (26)

$$\min_{Z\in\mathbb{D}} \Psi(Z) \leq \Phi(\theta_m^{\min}) \leq \Phi(\theta_m^*) \xrightarrow{m\to\infty} \min_{Z\in\mathbb{D}} \Psi(Z). \tag{28}$$

Finally, we show that $\lim_{m\to\infty} d_k\left(\hat{X}, Y^{\theta_m^{\min}}\right) = 0$ for all $1 \leq k \leq K$. Applying (23), (24) and (1) in reverse order than it was done in (25) and finally Lemma C.4, yields

$$d_k\left(\hat{X}, Y^{\theta_m^*}\right) \leq \frac{c_0\,c_1\,c_3}{c_2}\,\mathbb{E}\left[\frac{1}{n}\sum_{i=1}^n \left|M_{t_i} \odot (\hat{X}_{t_i^-} - Y^{\theta_m^*}_{t_i^-})\right|_2^2\right]^{1/2} = \frac{c_0\,c_1\,c_3}{c_2}\left(\Phi(\theta_m^*) - \Psi(\hat{X})\right)^{1/2} \xrightarrow{m\to\infty} 0, \tag{29}$$

which completes the first part of the proof. $\qquad\square$

Next we assume the size $m$ of the neural network and of the signature truncation level is fixed and we study the convergence of the Monte Carlo approximation when the number of training paths $N$ increases. The convergence analysis is based on Lapeyre & Lelong (2021, Chapter 4.3) and follows Herrera et al. (2021, Theorem E.13). We define the separable Banach space $\mathcal{S} := \{x = (x_i)_{i\in\mathbb{N}} \in \ell^1(\mathbb{R}^d) \mid \|x\|_{\ell^1} < \infty\}$ for a suitable $d$ (see below) with the norm $\|x\|_{\ell^1} := \sum_{i\in\mathbb{N}} |x_i|_2$, the function

$$G(x, y, m) := |m \odot (x - y)|_2$$

and $\xi_j := (\xi_{j,0}, \ldots, \xi_{j,n^{(j)}}, 0, \ldots)$, where $\xi_{j,k} := (t_k^{(j)}, O_{t_k^{(j)}}^{(j)}, M_{t_k^{(j)}}^{(j)}, \pi_m(\tilde{O}^{\leq t_k^{(j)}, (j)})) \in \mathbb{R}^d$ and $t_k^{(j)}$, $M_{t_k^{(j)}}^{(j)}$ and $O_{t_i^{(j)}}^{(j)}$ (with 0 entries for coordinates which are not observed) are random variables describing the $j$-th realization of the training data (cf. Section 2). Let $n^j(\xi_j) := \max_{k\in\mathbb{N}}\{\xi_{j,k} \neq 0\}$, $t_k(\xi_j) := t_k^{(j)}$, $O_k(\xi_j) := O_{t_k^{(j)}}^{(j)}$ and $M_k(\xi_j) := M_{t_k^{(j)}}^{(j)}$. By this definition we have $n^{(j)} = n^j(\xi_j)$ $\mathbb{P}$-almost-surely. Moreover, we have that $\xi_j$ are i.i.d. random variables taking values in $\mathcal{S}$. Let us write $Y_t^\theta(\xi)$ to make the dependence of $Y$ on the input and the weight $\theta$ explicit. Then we define

$$h(\theta, \xi_j) := \frac{1}{n^j(\xi_j)} \sum_{i=1}^{n^j(\xi_j)} G\left(O_i(\xi_j), Y^\theta_{t_i(\xi_j)-}(\xi_j), M_i(\xi_j)\right)^2.$$

The following lemma is known from Krach et al. (2022).

**Lemma C.5.** *Almost-surely the random function $\theta \in \Theta_m \mapsto Y_t^\theta$ is uniformly continuous for every $t \in [0, T]$.*

Now we are ready to prove the second part of our main theorem.

*Proof of Theorem C.3 – Part 2.* First we note that, $Y_t^\theta$ is the (integration over the) output of (bounded output) neural networks and therefore bounded in terms of the input, the weights (which are bounded by $m$), $T$ and some constant depending on the architecture and the activation functions of the neural network. In particular we have that $|Y_t^\theta(\xi_j)| \leq \tilde{B} \leq \tilde{B}\left((X_T^{\star,(j)}+1)^p + \sum_{i=0}^{n^j(\xi_j)} |\epsilon_i^{(j)}|\right)$ for all $t \in [0,T]$ and $\theta \in \Theta_m$ for some constant $\tilde{B}$ (possibly depending on $m$), where $X^{\star,(j)}, \epsilon_i^{(j)}$ corresponds to the input $\xi_j$. Hence,

$$G\left(O_i(\xi_j), Y_{t_i(\xi_j)-}^\theta(\xi_j), M_i(\xi_j)\right)^2 = \left|M_i(\xi_j) \odot (O_i(\xi_j) - Y_{t_i(\xi_j)-}^\theta(\xi_j))\right|_2^2$$

$$\leq \left((B+\tilde{B})\left((X_T^\star+1)^p + \sum_{i=0}^{n^j(\xi_j)} |\epsilon_i|\right)\right)^2 = \left(\frac{B+\tilde{B}}{B}U^{(j)}\right)^2,$$

where $U^{(j)}$ is as defined before corresponding to the input $\xi_j$. Hence,

$$\mathbb{E}\left[\sup_{\theta \in \Theta_m} h(\theta, \xi_j)\right] \leq \mathbb{E}\left[\frac{1}{n^j(\xi_j)} \sum_{i=1}^{n^j(\xi_j)} \left(\frac{B+\tilde{B}}{B}U^{(j)}\right)^2\right] < \infty, \tag{30}$$

by Assumption 2.1 and (27). By Lemma C.5, the function $\theta \mapsto h(\theta, \xi_1)$ is continuous, hence, we can apply Krach et al. (2022, Lemma 4.6), yielding that almost-surely for $N \to \infty$ the function

$$\theta \mapsto \frac{1}{N} \sum_{j=1}^N h(\theta, \xi_j) = \hat{\Phi}_N(\theta) \tag{31}$$

converges uniformly on $\Theta_m$ to

$$\theta \mapsto \mathbb{E}[h(\theta, \xi_1)] = \Phi(\theta). \tag{32}$$

Moreover, we deduce from Krach et al. (2022, Lemma 4.6) that $d(\theta_{m,N}^{\min}, \Theta_m^{\min}) \to 0$ a.s. when $N \to \infty$. Then there exists a sequence $(\hat{\theta}_{m,N}^{\min})_{N \in \mathbb{N}}$ in $\Theta_m^{\min}$ such that $|\theta_{m,N}^{\min} - \hat{\theta}_{m,N}^{\min}|_2 \to 0$ a.s. for $N \to \infty$. The uniform continuity of the random functions $\theta \mapsto Y_t^\theta$ on $\Theta_m$ implies that for any fixed deterministic and bounded $\xi_0$ (taking values in the same space as the $\xi_j$),

$$|Y_t^{\theta_{m,N}^{\min}}(\xi_0) - Y_t^{\hat{\theta}_{m,N}^{\min}}(\xi_0)|_2 \to 0 \text{ a.s. for all } t \in [0,T] \text{ as } N \to \infty.$$

By continuity of $G$ this yields $|h(\theta_{m,N}^{\min}, \xi_0) - h(\hat{\theta}_{m,N}^{\min}, \xi_0)| \to 0$ a.s. as $N \to \infty$. Let $\xi_0$ now be a random variable which is independent of and identically distributed as the $\xi_j$ defined on a copy $(\Omega_0, \mathbb{F}_0, \mathcal{F}_0, \mathbb{P}_0)$ of the filtered probability space $(\Omega, \mathbb{F}, \mathcal{F}, \mathbb{P})$. Then the above statements hold for $\xi_0(\omega_0)$ for $\mathbb{P}_0$-a.e. fixed $\omega_0$. Hence, we have for $\mathbb{P}$-a.e. fixed $\omega$, that $|h(\theta_{m,N}^{\min}(\omega), \xi_0) - h(\hat{\theta}_{m,N}^{\min}(\omega), \xi_0)| \to 0$ $\mathbb{P}_0$-a.s. as $N \to \infty$. With (30) we can apply dominated convergence which yields

$$\lim_{N \to \infty} \mathbb{E}_{\xi_0}\left[|h(\theta_{m,N}^{\min}(\omega), \xi_0) - h(\hat{\theta}_{m,N}^{\min}(\omega), \xi_0)|\right] = 0 \text{ for } \mathbb{P}\text{-a.e. } \omega.$$

Since for every integrable random variable $Z$ we have $0 \leq |\mathbb{E}[Z]| \leq \mathbb{E}[|Z|]$ and since $\hat{\theta}_{m,N}^{\min} \in \Theta_m^{\min}$ we can deduce that for $\mathbb{P}$-a.e. fixed $\omega$,

$$\lim_{N \to \infty} \Phi(\theta_{m,N}^{\min}(\omega)) = \lim_{N \to \infty} \mathbb{E}_{\xi_0}\left[h(\theta_{m,N}^{\min}(\omega), \xi_0)\right] = \lim_{N \to \infty} \mathbb{E}_{\xi_0}\left[h(\hat{\theta}_{m,N}^{\min}(\omega), \xi_0)\right] = \Phi(\theta_m^{\min}). \tag{33}$$

Now by triangle inequality, for $\mathbb{P}$-a.e. fixed $\omega$, we have for $\hat{\Phi}_{\tilde{N}}$ and $\Phi$ defined through test samples $\tilde{\xi}_j$ on $\Omega_0$, i.e., independent of and identically distributed as the training samples $\xi_j$ yielding $\theta_{m,N}^{\min}(\omega)$,

$$|\hat{\Phi}_{\tilde{N}}(\theta_{m,N}^{\min}(\omega)) - \Phi(\theta_m^{\min})| \leq |\hat{\Phi}_{\tilde{N}}(\theta_{m,N}^{\min}(\omega)) - \Phi(\theta_{m,N}^{\min}(\omega))| + |\Phi(\theta_{m,N}^{\min}(\omega)) - \Phi(\theta_m^{\min})|. \tag{34}$$

(31) and (32) imply that the first term on the right hand side converges to 0 when $\tilde{N} \to \infty$ and (33) implies that the second term on the right hand side converges to 0 a.s. when $N \to \infty$. Moreover, the uniform

convergence in (31) and (32) yields the same result when setting $\tilde{N} = N$. Furthermore, Krach et al. (2022, Lemma 4.6) yields the same result for $\hat{\tilde{\Phi}}_N(\theta_{m,N}^{\min})(\omega)$, i.e., when $\hat{\tilde{\Phi}}_N$ and $\Phi$ are defined through the $\xi_j$ on the probability space corresponding to the training data. This finishes the proof of the convergence with respect to $N$.

Finally, we want to show the joint convergence. We define $N_0 := 0$ and for every $m \in \mathbb{N}$

$$N_m(\omega) := \min \left\{ N \in \mathbb{N} \mid N > N_{m-1}(\omega), |\Phi(\theta_{m,N}^{\min}(\omega)) - \Phi(\theta_m^{\min})| \leq \tfrac{1}{m} \right\},$$

which is possible due to (33) for $\mathbb{P}$-a.e. $\omega$. Then (28) implies that for $\mathbb{P}$-a.e. $\omega$

$$|\Phi(\theta_{m,N_m(\omega)}^{\min}(\omega)) - \Psi(\hat{X})| \leq \tfrac{1}{m} + |\Phi(\theta_m^{\min}) - \Psi(\hat{X})| \xrightarrow{m \to \infty} 0.$$

Therefore, we can apply the same arguments as in the first part of the proof (cf. (29)) to show that

$$d_k \left( \hat{X}, Y^{\theta_{m,N_m(\omega)}^{\min}(\omega)} \right) \leq \frac{c_0 \, c_1 \, c_3}{c_2} \left( \Phi(\theta_{m,N_m(\omega)}^{\min}(\omega)) - \Psi(\hat{X}) \right)^{1/2} \xrightarrow{m \to \infty} 0,$$

for every $1 \leq k \leq K$ and for $\mathbb{P}$-a.e. $\omega$. $\qquad\qquad\square$

The following corollary follows as in Krach et al. (2022).

**Corollary C.6.** *In the setting of Theorem C.3, we also have that $\mathbb{P}$-a.s.*

$$\Phi(\theta_{m,N_m}^{\min}) \xrightarrow{m \to \infty} \Psi(\hat{X}) \quad \text{and} \quad \hat{\tilde{\Phi}}_{\tilde{N}_m}(\theta_{m,\tilde{N}_m}^{\min}) \xrightarrow{m \to \infty} \Psi(\hat{X}),$$

*for a suitable increasing random sequence $(\tilde{N}_m)_{m \in \mathbb{N}}$ in $\mathbb{N}$.*

# D  Experimental Details

Our experiments are based on the implementation used by Krach et al. (2022), which is available at https://github.com/FlorianKrach/PD-NJODE. Therefore, we refer the reader to its Appendix for any details that are not provided here.

## D.1  Differences between the Implementation and the Theoretical Description of the PD-NJ-ODE

Since we use the same implementation of the PD-NJ-ODE, all differences between the implementation and the theoretical description listed in Krach et al. (2022, Appendix D.1.1) also apply here. In particular, we use standard neural networks for $f_{\theta_1}$ and $\rho_{\theta_2}$ with additional inputs. In contrast to the original objective function, we do not need to add a regularizing constant for the noise-adapted objective functions (6) and (7), since no square-root needs to be computed. We do not use the $\tilde{u}$-coordinates of $\hat{X}^{\leq t}$ (i.e., we only use the first $d_X$ coordinates and omit the remaining coordinates from $d_X + 1$ to $2d_X$) in the implementation, since we anyways use the observation times and masks as additional inputs for $\rho_{\theta_2}$.

## D.2  Details for Noisy Observations

**Dataset.** We sample paths from a standard 1-dimensional Brownian motion on the interval $[0, 1]$, i.e., with $T = 1$, and discretisation time grid with step size $0.01$. At each time point we observe the process with probability $p = 0.1$. Whenever the process is observed, an independent observation noise is sampled from a centered normal distribution with standard deviation $\sigma = 0.5$ and added to the observation. The model never sees the observation of the original process, but only the noisy observation. We sample $20'000$ paths of which $80\%$ are used as training set and the remaining $20\%$ as test set.

**Architecture.** We use the PD-NJ-ODE with the following architecture. The latent dimension is $d_H = 100$, the readout network is a linear map and the other 2 neural networks have the same structure of 1 hidden layer with ReLU activation function and 100 nodes. The signature is used up to truncation level 3, the encoder is recurrent and the decoder uses a residual connection.

**Training.** We use the Adam optimizer with the standard choices $\beta = (0.9, 0.999)$, weight decay of 0.0005 and learning rate 0.001. Moreover, a dropout rate of 0.1 is used for every layer and training is performed with a mini-batch size of 200 for 200 epochs. The model is trained once with the noise-adapted objective function (6) and once with the original one (3).

### D.3 Details for Dependent Observation Framework

**Dataset.** We use the Euler scheme to sample paths from a 1-dimensional Black–Scholes model (geometric Brownian motion) with drift $\mu = 2$, volatility $\sigma = 0.3$, and starting value $X_0 = 1$. At each time point we observe the process with the probability $\mathbb{P}(M_i = 1 \,|\, \mathcal{A}_{t_i-}) = \mathbb{E}[M_i \,|\, \mathcal{A}_{t_i-}]$, where $M_i$ is described in Example 4.13, using $\eta = 3$ and $p = 0.1$. We use the same discretisation grid and dataset sizes as in Section D.2.

**Architecture.** We use the PD-NJ-ODE with the following architecture. The latent dimension is $d_H = 50$ and all 3 neural networks have the same structure of 2 hidden layers with tanh activation function and 50 nodes. The signature is used up to truncation level 3, the encoder is recurrent and the decoder does not use a residual connection.

**Training.** Same as in Section D.2 but only trained with original loss function (3).

### D.4 Details for Physionet with Observation Noise

**Dataset.** Details on the standard Physionet dataset are given in Herrera et al. (2021, Appendix F.5.3). In particular, we use a train-test split of 80% to 20% (i.e., $N = 6\,400$ training paths, and $N_{\text{test}} = 1\,600$ test paths). On the train set, the entire path amounting to 48 hours is used as input to the model, while on the test set only the first 24 hours are used as input and the second 24 hours are used to compute the evaluate MSE. In particular, the model predicts starting from 24 hours until 48 hours and at every time point where there is an observation, the squared error is computed between the observation and the prediction. Importantly, the observation is not used as an input to the model afterwards. I.e., we evaluate $Y_{t_i}^{\theta_{m,N}^{\min}}(\tilde{X}^{\leq 24\,\text{hours}})$, while in theory we could get even better results by evaluating $Y_{t_i}^{\theta_{m,N}^{\min}}(\tilde{X}^{\leq t_i-})$.[23] For the datasets with synthetically added noise, the variance of the noise is chosen relative to the variance of the data. Since the $d_X = 41$ different coordinates of the Physionet dataset describe very different health parameters on different scales, we first compute each coordinate's standard deviation $\sigma_{data,j}$ on the training set. Then we add i.i.d. noise samples $\epsilon_{i,j} \sim \mathcal{N}(0, \zeta^2 \sigma_{data,j}^2)$, for $\zeta \geq 0$, to each observed coordinate $X_{t_i,j}$ of input samples for the model, i.e., to each observation of the training set as well as to each observation on the first 24 hours of the test set (but not to any of the observations on the second 24 hours of the test set, which are exclusively used for evaluating the model). This is done for $\zeta \in \{0, 0.2, 0.4, 0.6, 0.8, 1\}$, where the same seed is used for each of them, to make the results more comparable.

**Architecture.** We use the PD-NJ-ODE with the following architecture. The latent dimension is $d_H = 50$ and all 3 neural networks have the same structure of 1 hidden layer with tanh activation function and 50 nodes. The signature is used up to truncation level 2, the encoder is recurrent and the decoder uses a residual connection.

**Training.** For each dataset the model is trained with mini-batch size 50 for 100 epochs, once with the noise-adapted objective function (6) and once with the original one (3). Otherwise the same as in Section D.2.

---

[23] In practice both predictions $Y_{t_i}^{\theta_{m,N}^{\min}}(\tilde{X}^{\leq 24\,\text{hours}})$ and $Y_{t_i}^{\theta_{m,N}^{\min}}(\tilde{X}^{\leq t_i-})$ can be useful depending on the situation. One has to use $Y_t^{\theta_{m,N}^{\min}}(\tilde{X}^{\leq 24\,\text{hours}})$ if one is at the situation that one has observed the patient for exactly 24 hours and wants to forecast the next 24 hours. On the other hand, at any time $s$ one should use all the currently available information to make the optimal prediction $Y_t^{\theta_{m,N}^{\min}}(\tilde{X}^{\leq s}) = Y_t^{\theta_{m,N}^{\min}}(\tilde{X}^{\leq \tau(s)})$. While our model is flexible enough to make predictions $Y_t^{\theta_{m,N}^{\min}}(\tilde{X}^{\leq s})$ for every combination $t > s$, we only evaluate $Y_{t_i}^{\theta_{m,N}^{\min}}(\tilde{X}^{\leq 24\,\text{hours}})$, since some competing methods compared to in Herrera et al. (2021) are limited to this restricted setting of one fixed value of $s = 24\,\text{hours}$. However, especially if one is interested in now-casting, the performance of $Y_{t_i}^{\theta_{m,N}^{\min}}(\tilde{X}^{\leq t_i-})$ would be the more relevant metric (in which we expect our model to outperform its competitors even more clearly).

We always report the minimal evaluation MSE among all trained epochs, as was done in prior works. Due to the shorter training compared to the 175 epochs in Krach et al. (2022), the evaluation MSE of the original PD-NJ-ODE on the dataset without synthetic noise is slightly larger ($2 \cdot 10^{-3}$ compared to $1.93 \cdot 10^{-3}$). We could imagine that for our new loss, one should actually use even more epochs rather than less epochs, since the additional term that was only present in the old loss pushed the jump-network $\rho_{\theta_2}$ faster into the right direction directly from the start of the training, while the new loss trains the jump-network only indirectly and thus probably slower.

## E  Pseudocode

In addition to our code repository (`https://github.com/FlorianKrach/PD-NJODE`), we provide pseudocode for the key parts of our method. We remark that multiple python packages are available for computing the signature of a process numerically, as for example `ESig`, `iisignature` and `signatory`. Therefore, we do not provide pseudocode for its computation. For our implementation, we use the great `iisignature` package implemented by Reizenstein & Graham (2018). We note that our definition of $\tilde{X}^{\leq t}$, with its time-consistency, allows for efficient online updates of the signature (i.e., we do not have to recompute the signature for the whole path at every new observation time $t_k$).

In Algorithm 1 we present the forward pass of the PD-NJ-ODE, which can be used for training as well as for evaluating the model. Algorithms 2 and 3 present the standard loss function and the loss function for noisy observations, respectively. The training for the model follows the standard neural network training approach via stochastic gradient descent (SGD) and is presented in Algorithm 4. We assume that a library is used for the neural network implementation which allows for automatic differentiation, as for example `PyTorch` (Paszke et al., 2019), which we used in our implementation. For simplicity, we present the simplest type of SGD, with a fixed step size and no momentum or weight decay. In our implementation we actually use the more sophisticated method Adam (Kingma & Ba, 2014).

## Notation

$F_j$  The Doob-Dynkin Lemma (Taraldsen, 2018, Lemma 2) implies the existence of measurable functions $F_j : [0, T] \times [0, T] \times BV^c([0, T]) \to \mathbb{R}$ such that $\hat{X}_{t,j} = F_j\left(t, \tau(t), \tilde{X}^{\leq \tau(t)}\right)$, since $\hat{X}_{t,j} := \mathbb{E}[X_{t,j}|\mathcal{A}_t] = \mathbb{E}[X_{t,j}|\tilde{X}^{\leq \tau(t)}]$ (see Section 2). The goal of our model $Y^\theta$ is to learn this function $F = (F_j)_{1 \leq j \leq d_X}$, i.e., $\lim_{m \to \infty} Y_t^{\theta_{m,N_m}^{\min}}\left(\tilde{X}^{\leq \tau(t)}\right) = F\left(t, \tau(t), \tilde{X}^{\leq \tau(t)}\right) = \hat{X}_t$, while we are usually not able to write down $F$ explicitly. In settings with noise (e.g., in Section 3 and in the appendix) we replace $\tilde{X}^{\leq \tau(t)}$ by $\tilde{O}^{\leq \tau(t)}$. 4, 8, 9, 16, 34, 36, 38, 39

$K$  The "maximal" value of $n$, i.e., the essential supremum $K := \sup\left\{k \in \mathbb{N} \,|\, \tilde{\mathbb{P}}(n \geq k) > 0\right\} \in \mathbb{N} \cup \{\infty\}$ of $n$ (see Section 2). In Sections 2 and 3, $K$ is defined with respect to $\tilde{\mathbb{P}}$, but in Section 4 (and all sections thereafter including the appendix), $K$ is defined with respect to $\mathbb{P}$. 3–8, 13–15, 18, 20–24, 36–38, 40, 42, 44, 46, 49

$M$  The observation mask $M = (M_k)_{0 \leq k \leq K}$, which is a sequence of random variables on $(\tilde{\Omega}, \tilde{\mathcal{F}}, \tilde{\mathbb{P}})$ taking values in $\{0, 1\}^{d_X}$ such that $M_k$ is $\tilde{\mathcal{F}}_{t_k}$-measurable. The $j$-th coordinate of the $k$-th element of the sequence $M$, i.e., $M_{k,j}$, signals whether $X_{t_k,j}$, denoting the $j$-th coordinate of the stochastic process at observation time $t_k$, is observed. By abuse of notation we also write $M_{t_k} := M_k$ (see Section 2). In Sections 2 and 3, $M$ is defined on $(\tilde{\Omega}, \tilde{\mathcal{F}}, \tilde{\mathbb{P}})$, but in Section 4 (and all sections thereafter including the appendix), $M$ is defined on $(\Omega, \mathcal{F}, \mathbb{P})$. 3–6, 8–10, 12–20, 33, 36–41, 43, 45–47, 49

$O$  The noisy observations $O_{t_k} := X_{t_k} + \epsilon_k$ for $0 \leq k \leq n$, where $\epsilon_k$ is the noise with known mean (see Section 3.1). For the majority of the paper we assume that the noise has zero mean (whenever we use (the MC-approximation of) the loss (6) and (7)), except Section 3.3 where we assume any known mean $\beta_i(\tilde{O}^{\leq \tau(t)}) := \mathbb{E}[\epsilon_i|\mathcal{A}_{t_i-}]$ and thus use (the MC-approximation of) the loss (10). We define $O_{t_i-} := X_{t_i-} + \epsilon_i$ and therefore also have that $O_{t_i} = O_{t_i-}$ almost surely. In Section 3, $\epsilon_k$ are i.i.d.

---

**Algorithm 1** Forward pass of the PD-NJ-ODE. A small step size $\Delta t$ is fixed and we denote $t_{n+1} := T$. ODESolve$(f, x, (a, b))$ numerically solves the 1st-order ODE defined by $f$, taking inputs $x$, for $t \in (a, b)$.

---

**Input:** Data points with timestamps and masks $\{(X_i, t_i, M_i)\}_{i=0\ldots n}$
**Output:** Prediction process $Y$
set $H_{0-} = 0$
**for** $i = 0$ **to** $n$ **do**
  construct $\tilde{X}^{\leq t_i}$ from data
  $S_i = \pi_m(\tilde{X}^{\leq t_i} - X_0)$                          $\triangleright$ compute truncated signature
  $H_{t_i} = \rho_{\theta_2}(H_{t_i-}, t_i, S_i, X_0)$         $\triangleright$ Update hidden state given next observation $x_i$
  $Y_{t_i} = \tilde{g}_{\tilde{\theta}_3}(H_{t_i})$                                $\triangleright$ compute output
  $s \leftarrow t_i$
  **while** $s + \Delta t < t_{i+1}$ **do**
    $H_{s+\Delta t} = \text{ODESolve}(f_{\theta_1}, (H_s, s, t_i, S_i, X_0), (s, s+\Delta t))$     $\triangleright$ get next hidden state
    $Y_{s+\Delta t} = \tilde{g}_{\tilde{\theta}_3}(H_{s+\Delta t})$                         $\triangleright$ compute output
    $s \leftarrow s + \Delta t$
  **end while**
  $H_{t_{i+1}-} = \text{ODESolve}(f_{\theta_1}, (H_{s-}, s, t_i, S_i, X_0), (s, t_{i+1}))$
  $Y_{t_{i+1}-} = \tilde{g}_{\tilde{\theta}_3}(H_{(s+\Delta t)-})$
**end for**

---

**Algorithm 2** Computation of the standard loss for PD-NJ-ODE. A small $\epsilon > 0$ is used for numerical stability.

---

**Input:** $N$ samples of data points with timestamps and masks $\{(X_i^{(j)}, t_i^{(j)}, M_i^{(j)})\}_{i=0\ldots n^{(j)}}$ and the respective prediction process $Y^{(j)}$ of PD-NJ-ODE for $1 \leq j \leq N$
**Output:** Loss $L$
$$L = \frac{1}{N}\sum_{j=1}^{N}\frac{1}{n^{(j)}}\sum_{i=1}^{n^{(j)}}\left(\left|M_i^{(j)} \odot \left(X_i^{(j)} - Y_{t_i}^{(j)}\right) + \epsilon\right|_2 + \left|M_i^{(j)} \odot \left(X_i^{(j)} - Y_{t_i-}^{(j)}\right) + \epsilon\right|_2\right)^2$$

---

**Algorithm 3** Computation of the noisy observation loss for PD-NJ-ODE.

---

**Input:** $N$ samples of noisy data points with timestamps, masks and conditional means of the noise $\{(O_i^{(j)}, t_i^{(j)}, M_i^{(j)}, \beta_i^{(j)})\}_{i=0\ldots n^{(j)}}$ and the respective prediction process $Y^{(j)}$ of PD-NJ-ODE for $1 \leq j \leq N$
**Output:** Loss $L$
$$L = \frac{1}{N}\sum_{j=1}^{N}\frac{1}{n^{(j)}}\sum_{i=1}^{n^{(j)}}\left|M_i^{(j)} \odot \left(\left(O_i^{(j)} - \beta_i^{(j)}\right) - Y_{t_i-}^{(j)}\right)\right|_2^2$$

---

**Algorithm 4** Training of PD-NJ-ODE.

---

**Input:** step size $\alpha$; number of epochs $E$; $N$ samples of noisy data points with timestamps, masks and conditional means of the noise $\{(O_i^{(j)}, t_i^{(j)}, M_i^{(j)}, \beta_i^{(j)})\}_{i=0\ldots n^{(j)}}$ for $1 \leq j \leq N$
**Output:** trained network parameters $\theta = (\theta_1, \theta_2, \tilde{\theta}_3)$
randomly initialize $\theta$
**for** $e = 0$ **to** $E$ **do**
  split the dataset into random batches $B$
  **for** $b$ **in** $B$ **do**
    **for** each sample $j$ **in** $b$ **do**
      compute prediction $Y^{(j)}$ with Algorithm 1
    **end for**
    compute loss $L$ over all samples in the batch $b$ with Algorithm 2 or 3
    update NN parameters $\theta \leftarrow \theta - \alpha \nabla_\theta L$
  **end for**
**end for**

---

random variables on $(\tilde{\Omega}, \tilde{\mathcal{F}}, \tilde{\mathbb{P}})$, but in Section 4 (and all sections thereafter including the appendix), $\epsilon_k$ are i.i.d. random variables on $(\Omega, \mathcal{F}, \mathbb{P})$. 7–12, 25, 33, 34, 36–41, 45–48

$T$  The largest time $T \in \mathbb{R}_{>0}$ we consider (see Section 2). 3–5, 8–10, 17–19, 21–24, 36, 38–42, 44–49

$X^\star$  The running maximum process of $X$, i.e., $X_t^\star(\omega) := \sup_{s \in [0,t]} |X_s(\omega)|_1 \ \forall t \in [0,T] \ \forall \omega \in \Omega$ with $X^\star : (\Omega, \mathcal{F}, \mathbb{F}, \mathbb{P}) \to \mathbb{R}_{\geq 0}{}^{[0,T]}$ (see Section 2). 3–5, 8–10, 17, 36, 39–41

$X$  The stochastic process $X := (X_t)_{t \in [0,T]}$ we want to study (which is an adapted càdlàg stochastic process $X : (\Omega, \mathcal{F}, \mathbb{F}, \mathbb{P}) \to \mathbb{R}^{d_X}{}^{[0,T]}$). In our notation $X_{t,k}^{(j)}$ refers to the $k$-th coordinate of the $j$-th path at time $t$ for $1 \leq k \leq d_X$, $1 \leq j \leq N$ and $t \in [0,T]$ (see Section 2). E.g., for a medical data-set, the index $j$ could correspond to a patient and the first coordinate could correspond to the body-temperature and the second coordinate could correspond to the blood-pressure, then $X_{t,1}^{(7)}(\omega) = 36.9$ would mean that the $7^{\text{th}}$ patient had a body temperature of 36.9°C at time (or age) $t$. 2–27, 33–37, 43–49

$Y$  The output $Y^\theta(\tilde{X}^{\leq \tau(\cdot)})$ of our PD-NJ-ODE (2) which should approximate $\hat{X}$ for properly trained parameters $\theta$ (see Definition 2.6). We use the short notation $Y^{\theta,j} := Y^\theta(\tilde{X}^{\leq \tau(\cdot),(j)})$. 6–10, 14, 17, 21–24, 33–35, 37, 39–45

$\hat{O}$  The conditional expectation of $O$, which is its $L^2$-optimal prediction (Krach et al., 2022, Proposition 2.5) given the currently available information, is defined as $\hat{O}_{t_i-} := \mathbb{E}_{\mathbb{P} \times \tilde{\mathbb{P}}}[O_{t_i-} | \mathcal{A}_{t_i-}] = \hat{X}_{t_i-}$ (see the proof of Lemma 3.4). Only directly at observation times, $\hat{O}_{t_i}$ and $\hat{X}_{t_i}$ deviate from each other; i.e., in general $\hat{X}_{t_i,k} = \mathbb{E}[X_{t_i,k} | \mathcal{A}_{t_i}] \neq O_{t_i,k} = \mathbb{E}[O_{t_i,k} | \mathcal{A}_{t_i}] = \hat{O}_{t_i,k}$ if $M_{i,k} = 1$ (see Section 3). In Section 3, the expectation is taken with respect to $\mathbb{P} \times \tilde{\mathbb{P}}$, but in Section 4 (and all sections thereafter including the appendix), the expectation is taken with respect to $\mathbb{P}$. 9, 37

$\hat{X} = (\hat{X}_t)_{0 \leq t \leq T}$  The conditional expectation process of $X$, which is its $L^2$-optimal prediction (Krach et al., 2022, Proposition 2.5) given the currently available information, is defined as $\hat{X} = (\hat{X}_t)_{0 \leq t \leq T}$, with $\hat{X}_t := \mathbb{E}_{\mathbb{P} \times \tilde{\mathbb{P}}}[X_t | \mathcal{A}_t]$ (see Section 2). In Sections 2 and 3, the expectation is taken with respect to $\mathbb{P} \times \tilde{\mathbb{P}}$, but in Section 4 (and all sections thereafter including the appendix), the expectation is taken with respect to $\mathbb{P}$. 3, 4, 6–11, 14–17, 20–24, 34, 36–40, 42, 44, 46

$\hat{\Phi}_N$  The objective function $\hat{\Phi}_N : \Theta \to \mathbb{R}, \theta \mapsto \hat{\Phi}_N(\theta) := \Psi(Y^\theta(X))$ on the parameter-space $\Theta$ for a finite number of $N \in \mathbb{N}$ training paths (i.e., the Monte Carlo approximation of $\Phi$). In Sections 2, 4 and 6.2, $\hat{\Phi}_N$ is defined in Equation (5) (this old objective function only leads to the correct results for noiseless observations, but in Section 3 and in the appendix, $\hat{\Phi}_N$ is defined analogously via (6) and (7) (this new objective function is different from the old one and even leads to the right result for noisy observations). In Sections 6.1 and 6.3 we compare both losses against each other. Note that in Section 3.3 we use another modification of the loss (10) for the case of noise with known non-zero mean. Note that these 3 variants of $\hat{\Phi}_N$ are the loss functions on which we actually train our parameters in practice with finitely many training data. 6–8, 34, 36, 37, 41, 42, 47, 48

$(\Omega, \mathcal{F}, \mathbb{F}, \mathbb{P})$  The filtered probability space $(\Omega, \mathcal{F}, \mathbb{F} := (\mathcal{F}_t)_{0 \leq t \leq T}, \mathbb{P})$ on which $X$ is defined (see Section 2). In Section 4 (and all sections thereafter including the appendix) all stochastic objects $X, n, K, t_i, \tau, M, \mathcal{A}_t$ are defined on this probability space. However, in Sections 2 and 3 only $X$ is defined on this space, while the observation framework $n, K, t_i, \tau, M$ is defined on a different filtered probability space $(\tilde{\Omega}, \tilde{\mathcal{F}}, \tilde{\mathbb{F}}, \tilde{\mathbb{P}})$. 3–10, 13–16, 19, 21–23, 36, 37, 40–44, 46–49

$(\tilde{\Omega}, \tilde{\mathcal{F}}, \tilde{\mathbb{F}}, \tilde{\mathbb{P}})$  The filtered probability space $(\tilde{\Omega}, \tilde{\mathcal{F}}, \tilde{\mathbb{F}} := (\tilde{\mathcal{F}}_t)_{0 \leq t \leq T}, \tilde{\mathbb{P}})$ on which $n, K, t_i, \tau, M$ are defined describing the observation framework, but $X$ is not defined on this space (see Section 2). In Section 4 (and all sections thereafter including the appendix) this space is not needed since all stochastic objects $X, n, K, t_i, \tau, M, \mathcal{A}_t$ are defined on $(\Omega, \mathcal{F}, \mathbb{F}, \mathbb{P})$. However, in Sections 2 and 3 this space is used for $n, K, t_i, \tau, M$ related to the events of observations. 3–10, 13, 44, 46–49

$\mathbb{A} := (\mathcal{A}_t)_{t \in [0,T]}$  The filtration of the currently available information $\mathbb{A} := (\mathcal{A}_t)_{t \in [0,T]}$ defined by

$$\mathcal{A}_t := \boldsymbol{\sigma}\left(X_{t_i,j}, t_i, M_{t_i} \mid t_i \leq t, \, j \in \{1 \leq l \leq d_X | M_{t_i,l} = 1\}\right),$$

in all sections, where we have noiseless observations $X_{t_i}$ and analogously defined as

$$\mathcal{A}_t := \boldsymbol{\sigma}\left(O_{t_i,j}, t_i, M_{t_i} \,|\, t_i \leq t,\, j \in \{1 \leq l \leq d_X | M_{t_i,l} = 1\}\right),$$

in all other sections, where we have noisy observations $O_{t_i}$. In both cases this corresponds to the information obtained from seeing the observations until time $t$ where $\boldsymbol{\sigma}(\cdot)$ denotes the generated $\sigma$-algebra (see Sections 2 and 3.1) We use the notion of stopped $\boldsymbol{\sigma}$-algebras

$$\mathcal{A}_{t_k} := \boldsymbol{\sigma}\left(X_{t_i,j}, t_i, M_{t_i} \,|\, i \leq k,\, j \in \{1 \leq l \leq d_X | M_{t_i,l} = 1\}\right),$$

and pre-stopped $\boldsymbol{\sigma}$-algebras

$$\mathcal{A}_{t_k-} := \boldsymbol{\sigma}\left(X_{t_i,j}, t_i, M_{t_i}, t_k \,|\, i < k,\, j \in \{1 \leq l \leq d_X | M_{t_i,l} = 1\}\right)$$

from Karandikar & Rao (2018, Definitions 2.37 and 8.1). In the sections with noise, we replace $X$ by $O$ . 3–6, 8–23, 33, 36, 37, 43, 44, 46, 47, 49

$\mathbb{D}$  The set of all càdlàg $\mathbb{R}^{d_X}$-valued $\mathbb{A}$-adapted processes (see Section 2). In Sections 2 and 3, the stochastic processes in $\mathbb{D}$ live on $(\Omega \times \tilde{\Omega}, \mathcal{F} \otimes \tilde{\mathcal{F}}, \mathbb{F} \otimes \tilde{\mathbb{F}}, \mathbb{P} \times \tilde{\mathbb{P}})$, but in Section 4 (and all sections thereafter including the appendix), they live on $(\Omega, \mathcal{F}, \mathbb{F}, \mathbb{P})$. 6–10, 16, 17, 37–40, 47

$\mathcal{J}$  The random set of the jump times $\mathcal{J} : (\Omega, \mathcal{F}, \mathbb{P}) \to \mathcal{P}(([0, T]))$ of $X$ (see Section 2). 3, 4, 36, 37

$\mathcal{N}$  The set $\mathcal{N}$ of bounded output neural networks consists of all feed-forward neural networks that have bounded outputs. In particular we assume that the final activation function applied to the output of the neural network is the bounded output activation function $\Gamma_\gamma = (\cdot) \min\left(1, \frac{\gamma}{|(\cdot)|_2}\right)$ (see Section 2). Throughout the paper we assume that the functions $f_{\theta_1}, \rho_{\theta_2} \in \mathcal{N}$ in the PD-NJ-ODE (2) are bounded output feedforward neural networks. Note that this assumption is not really important in practice but facilitates our theoretical proof. 5, 6, 39, 48, 49

$\mathcal{N}(\mu, \sigma^2)$  The normal distribution $\mathcal{N}(\mu, \sigma^2)$ (also known as Gaussian distribution) with mean $\mu$ and standard deviation $\sigma$. 12, 19, 20, 26, 43

$\mathcal{P}$  The powerset $\mathcal{P}(S)$ of a set $S$ is the set of all subsets of $S$. 47

$\Phi$  The objective function $\Phi : \Theta \to \mathbb{R}, \theta \mapsto \Phi(\theta) := \Psi(Y^\theta(X))$ on the parameter-space $\Theta$ in expectation (i.e., "for an infinite amount of training paths"). In Sections 2, 4 and 6.2, the old objective function given in (3) and (4) that only works for noiseless observations is used, but in Section 3 and in the appendix, the new objective function given in (6) and (7) that also works for noisy observations is used. In Sections 6.1 and 6.3 we compare both losses against each other. Note that in Section 3.3 we use another modification of the loss (10) for the case of noise with known non-zero mean. $\Phi(\theta)$ is the expected value of $\hat{\Phi}_N(\theta)$, where $\hat{\Phi}_N$ is the loss we actually train on based on $N$ training paths. 6–8, 10, 16, 17, 24, 39–42, 46, 48

$\Psi$  The objective function $\Psi : \mathbb{D} \to \mathbb{R}$ (cf. *equivalent objective function* from Remark 4.7 & Appendix A.1.4 of Krach et al. (2022)) on the path-space $\mathbb{D}$ in expectation (i.e., "for an infinite amount of training paths"). In Sections 2, 4 and 6.2, the old objective function (3) that only works for noiseless observations is used, but in Section 3 and in the appendix, the new objective function (6) that also works for noisy observations is used. In Sections 6.1 and 6.3 we compare both losses against each other. Note that in Section 3.3 we use another modification of the loss (10) for the case of noise with known non-zero mean. 6–10, 16, 17, 20, 24, 34, 36–40, 42, 46, 47

$\Theta_m$  The compact subset $\Theta_m \subset \Theta$ consists of all $\theta = (\theta_1, \theta_2, \tilde{\theta}_3) = ((\tilde{\theta}_1, \gamma_1), (\tilde{\theta}_2, \gamma_2), \tilde{\theta}_3) \in \Theta$ that correspond to (bounded output) neural networks with widths and depths that are at most $m$ and such that the truncated signature of level $m$ or smaller is used and such that the norms of the weights $\tilde{\theta}_i$ and the bounds $\gamma_i$ are bounded by $m$. I.e., $\Theta_m := \{\theta = ((\tilde{\theta}_1, \gamma_1), (\tilde{\theta}_2, \gamma_2), \tilde{\theta}_3) \in \hat{\Theta}_m \,|\, |\tilde{\theta}_i|_2 \leq m, \gamma_i \leq m\} \subset \Theta_m \subset \Theta$, where $\hat{\Theta}_m \subset \Theta$ is defined as the set of possible parameters for the 3 (bounded output) neural networks, such that their widths and depths are at most $m$ and such that the truncated signature

of level $m$ or smaller is used (see Section 2). We use the notation[24] $\Theta^i_m := \left\{ \theta_i \mid (\theta_1, \theta_2, \tilde{\theta}_3) \in \Theta_m \right\}$ and $\tilde{\Theta}^i_m := \left\{ \tilde{\theta}_i \mid ((\tilde{\theta}_1, \gamma_1), (\tilde{\theta}_2, \gamma_2), \tilde{\theta}_3) \in \Theta_m \right\}$ for the projections of the sets on the weights $\theta_i$ and $\tilde{\theta}_i$ respectively. 6, 7, 10, 35, 37–41, 48

$\Theta$ The set of all possible trainable parameters $\theta = (\theta_1, \theta_2, \tilde{\theta}_3) \in \Theta$ for our our PD-NJ-ODE (2) (see Definition 2.6). 6–8, 24, 34, 35, 46–49

$\gamma$ The parameters $\gamma$ are contained in $\theta = ((\tilde{\theta}_1, \gamma_1), (\tilde{\theta}_2, \gamma_2), \tilde{\theta}_3)$ and bound the outputs of bounded output neural networks. I.e., for every bounded output neural network $f_{\theta_1}, \rho_{\theta_2} \in \mathcal{N}$ it holds that $|f_{\theta_1}(x)|_2 \leq \gamma_1$ and $|\rho_{\theta_2}(x)|_2 \leq \gamma_2$, because of the bounded output activation function $\Gamma_{\gamma_i} : \mathbb{R}^d \to \mathbb{R}^d, x \mapsto \Gamma_{\gamma_i}(x) = x \cdot \min\left(1, \frac{\gamma_i}{|x|_2}\right)$ (see Section 2). I.e., $f_{\theta_1}(x) = f_{(\tilde{\theta}_1, \gamma_1)}(x) = \Gamma_{\gamma_1}\left(\tilde{f}_{\tilde{\theta}_1}(x)\right)$ and $\rho_{\theta_2}(x) = \rho_{(\tilde{\theta}_2, \gamma_2)}(x) = \Gamma_{\gamma_2}\left(\tilde{\rho}_{\tilde{\theta}_2}(x)\right)$. 5–7, 34, 39, 47–49

$\tau$ The last observation time $\tau(t)$ before a certain time $t$, i.e., $\tau : [0, T] \times \tilde{\Omega} \to [0, T]$, $(t, \tilde{\omega}) \mapsto \tau(t, \tilde{\omega}) := \max\{t_i(\tilde{\omega}) | 0 \leq i \leq n(\tilde{\omega}), t_i(\tilde{\omega}) \leq t\}$ (see Section 2). In Sections 2 and 3, $\tau$ is defined on $(\tilde{\Omega}, \tilde{\mathcal{F}}, \tilde{\mathbb{P}})$, but in Section 4 (and all sections thereafter including the appendix), $\tau$ is defined on $(\Omega, \mathcal{F}, \mathbb{P})$, i.e., $\tau : [0, T] \times \Omega \to [0, T]$. 3, 4, 6–8, 10, 12, 13, 20, 23, 34, 36–39, 43, 44, 46, 48

$\theta^{\min}_m \in \Theta^{\min}_m$ The set of all minimizers $\theta^{\min}_m \in \Theta^{\min}_m := \operatorname{argmin}_{\theta \in \Theta_m}\{\Phi(\theta)\}$ of the objective function $\Phi$ (corresponding to infinitely many training data) under the constraints of $\Theta_m$ for any given $m \in \mathbb{N}$ (see Theorem 2.7). 7, 9, 14, 16, 17, 21–23, 37, 39–43

$\theta^{\min}_{m,N} \in \Theta^{\min}_{m,N}$ The set of all minimizers $\theta^{\min}_{m,N} \in \Theta^{\min}_{m,N} := \operatorname{argmin}_{\theta \in \Theta_m}\{\hat{\Phi}_N(\theta)\}$ of the objective function $\hat{\Phi}_N$ (corresponding to $N$ training paths) under the constraints of $\Theta_m$ for any given $m, N \in \mathbb{N}$ (see Theorem 2.7). 7, 21–23, 33, 35, 41, 42, 44

$\theta$ The trainable parameters $\theta = (\theta_1, \theta_2, \tilde{\theta}_3) \in \Theta$ contain all trainable parameters. I.e., $\theta$ contains $\theta_i = (\tilde{\theta}_i, \gamma_i)$ for $i \in \{1, 2\}$ which parameterise the bounded output feedforward neural networks $f_{\theta_1}, \rho_{\theta_2} \in \mathcal{N}$ and $\tilde{\theta}_3$ parameterize the feedforward neural network $\tilde{g}_{\tilde{\theta}_3} \in \tilde{\mathcal{N}}$. Thus, $\theta = (\theta_1, \theta_2, \tilde{\theta}_3) \in \Theta$ parameterize all 3 parameterized functions $f_{\theta_1}, \rho_{\theta_2}, g_{\tilde{\theta}_3}$ in our PD-NJ-ODE (2) (see Definition 2.6). 6–8, 10, 16, 17, 24, 33–37, 39–42, 44–49

$\pi_m$ The truncated signature $\pi_m(\mathbf{X})$ of order $m \in \mathbb{N}$ of a continuous path with finite variation $\mathbf{X}$ is defined in Definition 2.5. In simple words, $\pi_m(\mathbf{X})$ is a finite dimensional feature-vector representing a continuous path $\mathbf{X}$. For every finite truncation level $m$, $\pi_m(\mathbf{X})$ does not capture all the information about the infinite dimensional object $\mathbf{X}$, but in our proof we use that there always exists a $m \in \mathbb{N}$ such that $\pi_m(\mathbf{X})$ describes $\mathbf{X}$ sufficiently well. 5, 6, 34, 38–40, 45

$\tilde{O}^{\leq t}$ The interpolated observation process $\tilde{O}^{\leq t}$ continuously interpolates the noisy observations $O_{t_i}$ that where observed before time $t$ (see Section 3.1). To be precise the first $d_X$ coordinates of $\tilde{O}^{\leq t}$ interpolate the noisy observations $O_{t_i}$, while the next $d_X$ coordinates of $\tilde{O}^{\leq t}$ captures explicit information on when which coordinate was observed and the last coordinate is just the time. At time $s \in [0, T]$, the interpolated observation process $\tilde{O}^{\leq t}_s(\omega) \in \mathbb{R}^{2d_X+1}$ is defined analogously to $\tilde{X}^{\leq t}_s(\omega) \in \mathbb{R}^{2d_X+1}$ from Section 2, by replacing $X$ by $O$. In Section 3, $\tilde{O}^{\leq t}$ is an adapted stochastic process on $(\Omega \times \tilde{\Omega}, \mathcal{F} \otimes \tilde{\mathcal{F}}, \mathbb{F} \otimes \tilde{\mathbb{F}}, \mathbb{P} \times \tilde{\mathbb{P}})$, but in Section 4 (and all sections thereafter including the appendix), $\tilde{O}^{\leq t}$ is an adapted stochastic process on $(\Omega, \mathcal{F}, \mathbb{F}, \mathbb{P})$. 8, 10, 12, 36–40, 44

$\tilde{X}^{\leq t}$ The interpolated observation process $\tilde{X}^{\leq t}$ continuously interpolates[25] the observations of $X$ that where observed before time $t$. To be precise the first $d_X$ coordinates of $\tilde{X}^{\leq t}$ interpolate the observations of $X$, while the next $d_X$ coordinates of $\tilde{X}^{\leq t}$ captures explicit information on when which coordinate was observed and the last coordinate is just the time. At time $s \in [0, T]$, the $j$-th coordinate $\tilde{X}^{\leq t}_{s,j}(\omega)$ of $\tilde{X}^{\leq t}_s(\omega) \in \mathbb{R}^{2d_X+1}$ is defined in Section 2. In Sections 2 and 3, $\tilde{X}^{\leq t}$ is an adapted stochastic process

---

[24] The definition is less ambiguous, if we write more precisely: $\Theta^i_m := \left\{ \theta_i \mid \exists \left(\theta'_1, \theta'_2, \tilde{\theta}'_3\right) \in \Theta_m : \theta_i = \theta'_i \right\}$.

[25] Interpolation also includes extrapolation within this paper. $\tilde{X}^{\leq t}$ interpolates the observations before time $t$ without leaking any information from observations after time $t$. Furthermore, its time-consistency allows for efficient online updates of its signature instead of recomputing its signature for the whole path at every new observation time $t_k$.

on $(\Omega \times \tilde{\Omega}, \mathcal{F} \otimes \tilde{\mathcal{F}}, \mathbb{F} \otimes \tilde{\mathbb{F}}, \mathbb{P} \times \tilde{\mathbb{P}})$, but in Section 4 (and all sections thereafter including the appendix), $\tilde{X}^{\leq t}$ is an adapted stochastic process on $(\Omega, \mathcal{F}, \mathbb{F}, \mathbb{P})$. 4, 6, 8, 21–24, 33, 34, 37, 42–46

$\tilde{\mathcal{N}}$  The set $\tilde{\mathcal{N}}$ of feedforward neural networks consists of all (classical) feed-forward neural networks. We assume that $\tilde{\mathcal{N}}$ is a set of standard feedforward neural networks with id $\in \tilde{\mathcal{N}}$ that satisfies the standard universal approximation theorem with respect to the supremum-norm on compact sets, see for example Hornik (1991, Theorem 2) Throughout the paper we assume that the functions $\tilde{g}_{\tilde{\theta}_1} \in \tilde{\mathcal{N}}$ in the PD-NJ-ODE (2) is a feedforward neural networks. 5, 6, 38, 48, 49

$\tilde{\theta}$  The trainable parameters $\tilde{\theta} = (\tilde{\theta}_1, \tilde{\theta}_2, \tilde{\theta}_3)$ contain all trainable weights and biases of $\theta = (\theta_1, \theta_2, \tilde{\theta}_3) \in \Theta$ but not the bounds $\gamma_1$ and $\gamma_2$. I.e., the classical feedforward neural network $\tilde{g}_{\tilde{\theta}_3} \in \tilde{\mathcal{N}}$ is fully parametrized by $\tilde{\theta}_3$, while the bounded output feedforward neural networks $f_{\theta_1}, \rho_{\theta_2} \in \mathcal{N}$ are paremetrized by $\theta_i = (\tilde{\theta}_i, \gamma_i)$ for $i \in \{1, 2\}$ which also includes $\gamma_1$ and $\gamma_2$ additionally to $\tilde{\theta}_1$ and $\tilde{\theta}_2$ (see Definition 2.6). 5–7, 34, 35, 37–39, 45, 47–49

$\tilde{u}$  The jump process $\tilde{u}_{t,j} := \sum_{k=0}^{K} M_{k,j} \mathbb{1}_{t_k \leq t}$ counts the coordinate-wise observations (see Section 2). 4, 42

$d_X$  The dimension $d_X \in \mathbb{N}$ of $X$ (i.e., $X_t(\omega) \in \mathbb{R}^{d_X}$) (see Section 2). 3, 4, 6, 8, 10, 11, 13–16, 18, 20, 34, 36–39, 42–44, 46–48

$d_k$  A pseudo metric between two càdlàg $\mathbb{A}$-adapted processes defined in Definition 2.4 in Section 2 to measure the distance between processes. 5, 7, 9, 10, 14, 16, 17, 20, 21, 24, 37, 38, 40, 42

$n$  The random number of observations $n : (\tilde{\Omega}, \tilde{\mathcal{F}}, \tilde{\mathbb{P}}) \to \mathbb{N}_{\geq 0}$ up to time $T$ (see Section 2). Every observation time $t_i \in [0, T]$ counts as 1 observation for this count (also for incomplete observations). In Sections 2 and 3, $n$ is defined on $(\tilde{\Omega}, \tilde{\mathcal{F}}, \tilde{\mathbb{P}})$, but in Section 4 (and all sections thereafter including the appendix), $n$ is defined on $(\Omega, \mathcal{F}, \mathbb{P})$. In our medical example, $n^{(j)}$ denotes the number of observation times for the $j$-th patient. 3–6, 8–10, 12–23, 36–40, 44–46, 48, 49

$t_i$  The random observation times $t_i : (\tilde{\Omega}, \tilde{\mathcal{F}}, \tilde{\mathbb{F}}, \tilde{\mathbb{P}}) \to [0, T] \cup \{\infty\}$ for $0 \leq i \leq K$ are sorted stopping times, with $t_i(\tilde{\omega}) := \infty$ if $n(\tilde{\omega}) < i$ (see Section 2). In our practical implementation we replace "$\infty$" by "$T$" (see Algorithm 1), since we are not interested in times after $T$ anyway. In Sections 2 and 3, $t_i$ are defined on $(\tilde{\Omega}, \tilde{\mathcal{F}}, \tilde{\mathbb{F}}, \tilde{\mathbb{P}})$, but in Section 4 (and all sections thereafter including the appendix), $t_i$ are defined on $(\Omega, \mathcal{F}, \mathbb{F}, \mathbb{P})$. 3–26, 33, 34, 36–41, 43–49

$u$  The jump process $u_t := \sum_{k=1}^{K} \mathbb{1}_{t_k \leq t}$ counts the observations without considering which coordinates where observed (see Section 2). 6

