# OpenReview forum: "Extending Path-Dependent NJ-ODEs to Noisy Observations and a Dependent Observation Framework"
_TMLR — Accepted by TMLR_

### Review · Reviewer_Zmm1 · 2023-10-31

**Summary Of Contributions:**

This paper extends the  PD-NJ-ODE (Path Dependant Neural Jump ODE) model to two extensions on the previous model.

Firstly, to settings with observation noise. The observation noise is assumed to be additive, and have a known expectation given previous observations. This includes cases of biased noise. This case is motivated by the very natural setting of noisy observations e.g. from sensor noise.

Secondly, to the setting where there is a dependence between the observation framework and the process. This setting is readily motivated by the healthcare application, where certain observation would induce more observations if a serious condition was observed.

These settings are defined rigorously mathematically and proved in detail.

Two simple experiments validate the methodology in 1D settings.

**Audience:**

Yes

**Claims And Evidence:**

Yes

**Requested Changes:**

I have no specific changes I require, I think the paper in is acceptable shape as-is. Addressing some of the comments in the Strengths and Weaknesses section however would improve the paper and its readability I believe.

**Strengths And Weaknesses:**

The two topics are clearly motivated and worthwhile extensions of the framework.

For a paper dealing with a technically challenging topic, the mathematics are explained well and with reasonable clarity. There are moments however where this becomes very dense, and reading notation becomes tricky as there is a lot to memorise. The authors have done a good job however with the challenge. Perhaps one suggestion would be to provide a notation appendix with a short definition of each term. Since the authors have done a good job at not reusing symbols, this should be reasonably easy.

To the best of my ability, I believe the mathematics to be correct. However this area is not my expertise.

For the less technical reader interested in implementing this framework, I think it would be quite challenging to interpret and code up. I appreciate there is a code base, but these can sometimes be just as challenging to try and decipher. An appendix with plain-terms / pseudo code explanations of key parts of the computation would be beneficial. For example, computing $\pi_m$ the truncated signature transformation.

On a similar practical note, while the experiments do show that the methods work on simple synthetic settings, a lack of further experimentation on real data makes it difficult to assess whether this framework is useful practically. The motivation on the healthcare setting is clear and a set of experiments in the setting on real data for example would be compelling.

---

> ### Author Response · Authors · 2023-12-15
> **Answer to review**
>
> We are grateful for your very positive feedback on our paper and the suggestions to further improve it.
> Below, we separately answer each point in detail. Additionally, we highlighted the substantial changes in the paper in colour.
>
>
> - As suggested by the reviewer, we added a notation glossary section at the end of the appendix. In the paper we added hyperlinks that point to the corresponding entries in the notation glossary.
>
> - We also added a pseudo code section to the appendix, where we provide pseudo code for the forward pass through the PD-NJ-ODE method, the computation of the loss and the training of the method. We note however, that we didn't provide pseudo code for the computation of the signature, since there are a handful of specialized python packages for this purpose. We listed 3 of them in the paper now, including the one we use in our implementation. We added a mathematical definition of the truncated signature $\pi_m$ into the paper.
>
> - We agree with the reviewer that experiments on real data are needed to show the methods practical usefulness. We want to note however, that this only applies for the extension with noisy observations, since our theoretical results of this paper show that the framework used in the predecessor paper \citep{krach2022optimal} can deal with dependence between the observed process and the observation framework.
>     Moreover, there does not exist a version of PD-NJ-ODEs that \emph{cannot} deal with this dependence.
>     Hence, the results on real world data that were shown in \cite{krach2022optimal} are representative here as well. The usefulness of the dependence result is purely theoretical, by showing that the existing method is also theoretically sound under more general settings.
>
>     On the other hand, the extension to noisy observations actually uses a different loss function and therefore allows for a comparison between the previous method (which could not deal with observation noise) and the new method on real world data.
>
>     Therefore, we added the new Section 3.5 to the paper, where we speak about the practical relevance of the noise-adapted loss function depending on the problem setting. Moreover, we added experiments on real world data in Section 6.3, where the practical importance of the noise-adapted loss function becomes visible, when comparing it to the original loss function.

---

### Review · Reviewer_oEbU · 2023-11-06

**Summary Of Contributions:**

The authors extend prior work [1] on incompletely observed path dependent jump ODEs to the setting where observations are observed with noise and to where the masking is dependent on the underlying process. In particular the authors show the conditional expectation of the underlying process is estimated by the L2 minimizer of the observation process and provide assumptions to support this, such as the noise being zero-mean.

[1] Krach et al 2022 Optimal Estimation of generic dynamics by path dependent neural jump odes

**Audience:**

Yes

**Broader Impact Concerns:**

No concerns

**Claims And Evidence:**

Yes

**Requested Changes:**

- Introduce a higher dimensional experiment to see if this method scales or some less toyish experiments
- Compare to particle filtering approaches, if applicable, or explain more why this is not suitable

**Strengths And Weaknesses:**

Strengths

- The work is well motivated
- The work is thorough, providing detailed assumptions, proofs and treated rigorously.
- The results appear correct

Weaknesses
- The zero-mean noise seems quite restrictive. And makes the results appears not so surprising given prior work -i.e. the L2 minimizer with zero-mean noise is also the L2 minimizer without noise. It is not clear the significance of the extension.
- The work is highly dependent on [1], indeed many proofs are extensions. Some concepts are not detailed such as "truncated signature of level m or smaller is used" where signatures are not defined or discussed. I imagine properly defined in [1]. It would be beneficial to the reader if the work was more self contained.
- Experimental results are 1-dimensional and appear quite simple. It is not clear if this method would work in more challenging larger scale / dimensional settings.
- I understand one would still need to take gradients through time in order to evaluate the ODE which may make things less scalable than recent ODE/ SDE frameworks (diffusion models / flow matching). Is this why there are only 1-d experiments?
- Does this really need neural jump ODEs? Would a particle filter and a state-space model suffice here? Indeed such particle filter + state-space models can be made differentiable and hence trainable. It may be beneficial to compare to such methods given they are highly related se e.g [2-5]. I am not familiar with any particular reference but I imagine PFs are applicable for jump diffusion processes with masked state and noisy observations. Is the idea that the assumptions are less restrictive and you do not need to know the observation model density under this framework?

---------------

In summary I believe the main contribution of this work is to relax the assumptions of [1] for more realistic applications. Unfortunately there are no real world examples / experiments which showcase the utility of the extension, nor any baselines other than prior work in [1].

[1] Krach et al 2022 Optimal Estimation of generic dynamics by path dependent neural jump odes
[2] Filtering Variational Objectives, Maddison et al 2017
[3] Auto-Encoding Sequential Monte Carlo, Le et al 2018
[4] Variational Marginal Particle Filters, Lai et al 2021
[5] Differentiable Particle Filtering via Entropy-Regularized Transport, Corenflos et al 2021

---

> ### Author Response · Authors · 2023-12-15
> **Answer to review**
>
> We thank you for your detailed feedback and your great suggestions to improve the paper. In the following, we answer each point in detail. Additionally, we highlighted the substantial changes in the paper in colour.
>
> ## Weaknesses
>
> - Even though our setting in Section 3.1 focuses on centered (zero-mean) noise, we extend this setting in Section 3.3 to more general noise structure, including deterministically biased noise or even bias functions depending on the previous observations.
> Additionally, we want to highlight that even in the centered case, the results are not immediate. You are correct that the $L^2$ minimizer with centred noise is also the $L^2$ minimizer without noise. But the important subtlety is that this is true everywhere except at observation times. At an observation time we have in general that $ \hat X_{t_i} = E[ X_{t_{i}}  |  O_{t_0}, ... , O_{t_{i}} ] \neq \hat O_{t_i} = E[O_{t_{i}} \, | \, O_{t_0}, ... , O_{t_{i}}]=O_{t_{i}} $.
>
>     By the definition of the original loss function of the  PD-NJ-ODE, it learns $\hat{O}_t$, and therefore the wrong behaviour at observations. In contrast to this, our noise-adapted version of it provably learns to predict $\hat{X}_t$. The different behaviours can be observed in the comparison in Figure 1.
>     To summarize: We agree for classical regression you don't need a different loss for zero-mean noise (the classical L2-loss obviously works for both settings - noise-less and zero-mean noise). However in our paper we point out the surprising fact, that for PD-NJ-ODEs one can not use the same loss for zero-mean noise; instead we had to propose a new loss that can deal with noisy observations, while the old loss used by Krach et al. does not work for zero-mean noise (see Figure 1).
>
>     Our contribution with respect to noise is not only that we provide a loss and a theory for noisy settings with known mean. Our second contribution with respect to noise is that we explain that even in the case of zero-mean noise the classical L2-loss proposed in Krach et al. does not lead to the correct result. We think this result is particularly relevant, since some people might be mislead by statements such as ``the L2 minimizer with zero-mean noise is also the L2 minimizer without noise'' to believe that they can use the same classical L2-loss proposed in Krach et al. for zero-mean noise settings. But we show in Figure 1 that this is not the case.
>     We agree that mathematically the proof is not extremely advanced, but in practice, in the case of zero-mean noise, it can have a big impact if people use our loss rather than the one proposed by Krach et al. (see Figure 1).
>
>     Further note that zero-mean noise is very common in practice (for example in the case of measurement-noise). This is a very common assumption in various Machine-learning regression settings. If one only observes noisy versions $O$ of $X$ and never any noise free version of $X$ during training it is impossible for any method to get rid of the bias of the noise. For example if the weight-scale at a hospital always measures 1 kg too much of body weight. It is impossible to see only from this data that the actual weight of all patients was always 1kg lower than measured. The setting we are studying is learning how to forecasting an unknown dynamic from data, where the training data only consists of noisy observations $O$. Filtering is a different problem, where one observes $X$ and $Y$ during training (or the dynamics are even known a priori) and then during inference one only observes $Y$ to forecast $X$. As mentioned in Section 1.2, we can easily extend our setting to these filtering tasks, where $Y$ and $X$ can have a very complicated relationship as already explained in Krach et al. (We tried to additionally clarify this in the Related Work section of the updated version of the paper.) One has to be careful not to confuse the role of $Y$ in the filtering problem with the role of $O$ in the forecasting problem.
>
> - Thank you for noting that the work was not self contained. We added the definition of the (truncated) signature to the paper and also made the paper more self-contained at other points in accordance with the suggestion of the other reviewer.

---

> ### Author Response · Authors · 2023-12-15
> **continuation of answer**
>
> - You are right that the examples we give are rather simple. However, their main purpose is to illustrate the main theoretical contributions of our work experimentally. We note that our theoretical results prove that our method should work in nearly any higher-dimensional or more complex setting, which is also supported by the experimental results on real world datasets presented in Krach et al. Especially for the purely theoretical extension to dependence between the underlying process and the observation framework, these experiments of Krach et al.  are representative, since the method itself didn't change.
> In particular, it is very likely that there actually is such a dependence in the Physionet dataset, as discussed in the Introduction and in Section 4.1. Hence, our paper provides the theoretical foundation for those empirical results of Krach et al. and, vice versa, those results show that the method discussed in our paper is applicable to complex high-dimensional settings.
> We also added a note on this to the paper.
> On the other hand, our method *did* change for the case of zero-mean (or known mean) noise. Therefore, we now added experiments in higher dimensions based on real word-data to show the benefits of our new method in the presence of zero-mean noise.
>
>
> - You are right that we need to take gradients through time during training (not when evaluating the PD-NJ-ODE during inference). However, this is not a big issue and the method scales to higher dimensional settings as was shown in the experiments on real world datasets in Krach et al. Note that our method is differentiable w.r.t. the parameters $\theta$ in contrast to particle filters. The reason why we only have 1-d experiments is that we consider this work as an extension of the previous work, where the practical applicability of the method in higher-dimensional and more complex settings was already shown experimentally. In this work we mainly wanted to illustrate the (theoretical) extensions with simply synthetic datasets. However, we added high-dimensional experiments that show the advantages of our new loss in the presence of zero-mean noise. The training on the real-world dataset takes ca. 1 day  on a CPU (would be faster on a GPU), but inference (i.e., making forecasts based on a new path) takes less than a second.

---

> ### Author Response · Authors · 2023-12-15
> **continuation 2 of answer**
>
> - Thank you for the references to particle filters. Even though particle filters are used for similar problems as the one studied in our paper, there are several (theoretical and practical) differences such that they cannot be applied to our problem setting.
>     Particle filtering methods are applied in the context of state-space models (SSM), which are characterized by a discrete latent Markov process  $(X_{t})_{ t  \geq 1 }$  and a discrete observation process
>
>     $ (Y_{t})_{t \geq 1} $  defined on a fixed time-grid.
>
>     Particle filters are used to approximate e.g. the conditional distribution of $X_t$ given the observations $(Y_s)_{1\leq s\leq t}$, or the joint distribution of
>
>     $(X_s, Y_s)_{1 \leq s \leq t}$, for any $t\geq 1$, using weighted sequential Monte Carlo samples.
>     In our work, we allow for a much more general setting than the SSM. In particular, we allow for a continuous-time (instead of discrete-time), non-Markovian stochastic process. Since our setting allows for jumps of the process, this also includes the SSM case of a discrete-time Markov process.
>     Moreover, in our setting, the underlying process can be observed at random, irregularly sampled, discrete observation times and the framework allows for incomplete observations, where some coordinates might not be observed. This irregular observation times are not fixed, but they can vary substantially from one path to another.
>     Also the primary goal of the neural jump ODE method is different, trying to make optimal forecasts for $X_t$ given all observations of $X$ prior to time $t$. As a special case (since the framework can deal with incomplete observations), this includes the filtering problem, however, in a more general setting, allowing for example to predict $X_t$ while only having (discrete) observations of $Y$ at randomly sampled observation times until time $s < t$. This was explained in Krach et al.
>     In principle, the filtering approach could also be used to deal with noisy observations. However, this would require some type of knowledge of the underlying distribution and the noise distribution (or access to noise-free observations of $X$ during training). In contrast to this, our extension for noisy observation, doesn't need such knowledge, i.e., we do not need any assumption what the underlying distribution of $X$ is, but it is enough to observe the noisy data samples.
>     Another big part of our work are the proofs showing that we can learn the dynamics only from observing finitely many noisy observations $O$ from sufficiently many paths (without any noise-free observations of $X$). We do not find such results in any of the papers on particle filters and SSMs you cited.
>
>     For these reasons, particle filters are not an alternative to the neural jump ODE method in these general settings.
>
>     As far as we understood, all the particle filter and SSM papers you cited consider a fixed time grid, while the main focus of our work lies on irregular random observation times.
>
>
> ## Requested changes
>
> - As requested, we added experiments on a higher-dimensional real world dataset in Section 5.3.
>
> - As requested, some explanation about particle filters and why they are not suitable as baseline was added to the related work.

---

> > ### Author Response · Authors · 2023-12-15
> > **sorry for the line breaks ...**
> >
> > ... somehow without them the equations do not get rendered correctly. We hope it is still understandable.

---

### Review · Reviewer_g67F · 2023-12-09

**Summary Of Contributions:**

This paper extends path dependent neural jump ODEs by attempting at removing two independence assumptions on coordinate-wise observation times and observation noise.

**Audience:**

Yes

**Claims And Evidence:**

Yes

**Requested Changes:**

1) Abstract: This has to be updated clearly. It is not clear at all how the authors propose to resolve the limitations of previous work. It is most helpful to specify here, exactly how this is achieved.

2) Section 2 title could be made less informal, perhaps "Technical Background" or something similar.

3) It is not clearly explained why $\tilde{X}\_{s,j}^{\leq t}$ is needed, why it takes the specific form it does, and what is the intuition behind this process. The formula looks sufficiently complex and it reads as a big pile of equations, whose role is not explained -- it is just noted that "Since $\tilde{X}_{s,j}^{\leq t}$ carries all information", please clarify. Why does the Doob-Dynkin Lemma apply?

4) Before Assumption 2.1, you mention "PD-NJ-ODE model" which is not yet introduced.

5) Several things in Definition 2.3 are not clarified: $\pi_m$, the meaning of $H_t$ and $Y_t$, $\tau(t)$. Please give intuition of this model as it is the backbone of the paper. It looks like $(H\_t)\_{t\geq 0}$ takes the role of a latent process that depends on $\tilde{X}_{s,j}^{\leq t}$, and $Y_t$ is the observation map that is noiseless.

6) Give more explanation as to why existence and uniqueness of the solution is justified. Does it not require some assumptions at least on observations? What are they? Are they already assumed?

7) Define the meaning of the notation $X_t$ vs. $X_{t-}$ without assuming that the reader will be an expert on stochastic analysis. Similarly, in the loss (3), explain why you need two terms and why it makes sense, and especially, what the difference and contributions of these two terms are.

8) Signature kernels are not introduced or explained - yet are used in Theorem 2.4 as the theorem as $m \to \infty$ in terms of signature kernel levels, from this reading, I actually do not know what they are. This is again readable for a very specific expert in my opinion.

9) What is $n^{(j)}$? So in essence is it assumed that, at every time step, multiple observations are available? And these number of observations are random? Please clarify this, while introducing the concept.

10) Also can you please explain Theorem 2.4 _after_ the theorem? It feels like it is proved that the loss function minima converges but that does not necessarily mean much as finding the minimiser is the real issue. In what sense this is a convergence guarantee of NJODE?

11) It looks like noisy model introduced in Sec. 3 is rather simple with just additive noise. Is it at least easy to generalize this to linear observation models $O_{t_k} = A X_{t_k} + \epsilon_k$?

12) I understand the logic of numbering in Assumption 3.1 but this can be confusing. You can assume 1, 2, 3, and 5 of Assumption 2.1 and number these as 1, 2, 3 anyway to easily refer back e.g. 1 of Assumption 3.1. I think I cleaner way is to number (i), (ii), e.g., and refer to them as Assumption 2.1(i) etc.

13) Before Assumption 3.1, it is mentioned that $F_j$ exists again, without much clarification. Also how does the existence result of $F_j$ interact with the assumptions introduced in Assumption 3.1? Are they related?

14) Remark 3.2: Remarks like these are not helpful. More context is helpful, e.g., what was the main points/conclusions of the discussion in Krauch et al. (2022)?

15) I am not sure why the proof of Theorem 2.4 is given in Section 3, by noting "for noisy observations" so therefore this is *not* the proof of Theorem 2.4 but a potentially proof of a new theorem. Can you clearly rewrite the new theorem, even though it might be repetition, but I thought this is the main contribution of the paper.

16) Again, to make the point above, I'm not sure in what sense this can be said to be a convergence proof, as it looks like it is just the convergence of the loss function to the original loss, which is evident from the fact that the loss is a Monte Carlo approximation.

17) I am not sure how realistic it is to assume $\beta_i$ is a known function. This is probably not a practical case, so not sure what conclusions can be drawn from Sec. 3.3. Please clarify.

18) Before moving to Sec. 4, I think the real interesting case in Section 3 would be, as mentioned above, a linear observation process. And potentially one where $O_{t_j}$ and $X_{t_j}$ are allowed to be in different dimensions, e.g., $A$ as defined above is not square. Is this extension doable? I suggest authors to investigate and clarify this.

19) Section 4 discussion about dependence is *very* helpful. My questions above apply to this section (Section 4) as well: Could you please improve readability in general by discussing the intuition of the proof in Sec. 4.3 before starting to give technical intermediate results like Prop. 4.3. What is the idea of the proof or main technical challenge? Why such a splitting result would be necessary?

20) Again "Sketch of Proof of 2.4 with dependence" is quite obscure. This is a proof of another result, which needs to be written as a new Theorem.

21) Is it possible to get any convergence results for the trained PD-NJ-ODE (let us say, a PD-NJ-ODE with parameter $\theta^\star$ which is $\varepsilon$-close to minima in function value)? If you use trained PD-NJ-ODE (or NJ-ODE) with a good parameter, then is the path produced by the NJODE close to "true conditional expectation"? I suspect this might not be hard to prove in terms of parameter error?

**Strengths And Weaknesses:**

Strenghts: The problem this paper tries to solve in practice is an important one - with irregularly observed noisy data points. I do believe that a framework like this could be very useful.

Weaknesses: There are several weaknesses of this submission. I write bullet points below, which are going to be expanded in Requested Changes section:

- Sometimes informal tone of the paper
- The way method is introduced obscures a lot of the points and makes the reading difficult for a reader not specifically an expert in previous NJODE literature. Various improvements are needed for readability (see below).

---

> ### Author Response · Authors · 2023-12-15
> **Answer to reviewer**
>
> We thank you for your detailed feedback and the numerous suggestions to improve the paper. In particular, we thank you for the numerous remarks to make the paper more self-contained (which is definitely a great benefit for readers who are not familiar with Krach et al.) as well as your suggestions to explain additional insights. We tried to address all these remarks, while not repeating too much of the content of Krach et al.
> In the following, we answer each point in detail. Additionally, we highlighted substantial changes to the paper in colour.
>
> 1. We added more details to the abstract on how we propose to resolve the prior limitations.
> 2. We kept the main title of Section 2 as ''Recall: the PD-NJ-ODE'' (since recalling the previous framework is the main purpose of the section) but called the new Subsection 2.2 ''Technical Background and Mathematical Notation'' as you suggested.
> 3. Since Section 2 is meant to recall (the most important parts of) the PD-NJ-ODE framework from Krach et al., we didn't include additional information like this, because we did not want to be too repetitive wrt. Krach et al.
>     However, we agree with you that the definition of $\tilde{X}^{\leq t}$ is rather complex and therefore added the additional information you asked for, also including more details why the Doob-Dynkin Lemma applies. For an explanation of the Doob-Dynkin lemma, see the answer to point 13.
> 4. Since Section 2 is recalling the setup of the PD-NJ-ODE (as noted in the very beginning of it), we do not think that there is a problem with mentioning PD-NJ-ODE before its formal definition in Definition~2.5. In particular, we speak about the framework defined in the cited work Krach et al., which is, in that sense, already introduced in the abstract and again in the introduction.
> 5. We added the new Definition 2.4, which formally introduces the signature and the truncated signature $\pi_m$. $\tau(t)$ is introduced in the list in the beginning of Section 2.2 (where leaving away $\tilde{\omega}$ is the standard notation for random variables).
>     You are right that $H_{t}$ is a latent process depending on the past observations through the signature of $\tilde{X}$. $Y_{t}$ is the output of the PD-NJ-ODE model (as mentioned in the paper). $Y_{t}$ is not the observed process (we do not use the notation of filtering here). $Y_{t}$ is an estimator for the conditional expectation
> $ \hat X_{t} = E [ X_{t} | \mathcal{A}_{t-} ] $.
>
>     In simple words, at time $s$, $\lim_{m\to\infty}Y_t^{\theta_{m,N_m}^\text{min}}(\tilde{X}^{\leq s})$ is the optimal forecast/prediction for the future value $X_t$ given all the (noisy) observations of $X$ observed up to time $s$. To repeat, we use the notation of forecasting, where we have (noisy) observations $O_{t_k}$ of $X$ and we want to predict the expected value of future values $X_t$. We do not use the notation of filtering, i.e., we do not observe any other process $Y$ in order to predict $X$. If one would apply our method to filtering than one would consider some coordinates of $X$ to be the observed process and other coordinates of $X$ to be the coordinates we want to predict and use the observations masks $M$; but we never use the variable $Y$ to denote the observed process (except for section 1.1 where we discuss related filtering work).
>     We added a short explanation for the intuition of the model and its single parts, while trying not to repeat too much of the previous papers.
>
> 6. We added more details for the existence and uniqueness of the PD-NJ-ODE model. In particular, we made it more precise that Lipschitz continuous activation functions are needed in the neural networks and cited the more detailed existence results of Cohen et al. Since we integrate with respect to finite variation processes, integrability is trivially satisfied in a path-wise manner. No additional assumptions on the observations are needed, because the neural network structure together with the finite variation integrators are regular enough. We added a footnote citing all relevant definitions giving short explanations why they are satisfied.
>     Note, that we are only stating that given fixed deterministic parameters $\theta$, given fixed deterministic observation data, the fixed deterministic ODE has an unique solution. (Given any finite number of training paths $N$ and a large model complexity $m$, there are multiple $\theta$ that minimize the loss, that result in different $Y^{\theta}$.)

---

> ### Author Response · Authors · 2023-12-15
> **continuation of answer**
>
> 7. We added a short explanation for the left limits after Definition 2.4, where this notation was first used. Additionally, we added a short explanation of the two terms in the loss function (3) as you requested. Moreover, in the new Section 3.5 we added a discussion on the practical difference between the original loss function (3) and the noise-adapted loss function (10).
>
> 8. We added Definition~2.4 where the signature and the truncated signature of a path are defined.
>
> 9. As explained in the beginning of Section 2.2, $n$ is a random variable defining the number of observations. In particular, for any fixed $(\omega, \tilde{\omega})$, $n(\tilde{\omega})$ is the number of observations along the path of $X(\omega)$. At any time $t$, there can either be an observation (if $t=t_k(\tilde{\omega})$ for some $k$) or not (the number of coordinates that are observed at an observation time $t_k$ can be inferred from $M_k$).
> As is standard when working with data, we then introduce $N$ i.i.d. random variables and processes $(X^{(j)}, M^{(j)}, n^{(j)}, t_1^{(j)},  \dotsc, t^{(j)}_{n^{(j)}})$ with the same distribution as $X$ and the observation framework $(M, n, t_1,  \dotsc, t_n)$, which correspond to the realizations of the training paths. With this we can define the Monte Carlo approximation (5) of the (theoretical) loss function (3). This is important, since clearly, only the Monte Carlo approximation (5) can actually be computed with data. We tried to make this a bit clearer in the paper.
>
>     Since we consider multiple i.i.d. realizations $X^{(j)}$ and $( M^{(j)}, n^{(j)}, t_1^{(j)},  \dotsc, t^{(j)}_{n^{(j)}})$
>
>     in the Monte Carlo approximation, it can of course happen, that some of them are observed at the same time (i.e., that in one state of the world $\tilde{\omega}$, there exist $j_1, j_2$ and $k_1, k_2$ s.t. $t_{k_1}^{(j_1)}(\tilde{\omega}) = t_{k_2}^{(j_2)}(\tilde{\omega})$). However, this is not relevant for our analysis.
>     We have added a sentence explaining $n^{(j)}$ in the notation glossary of $n$, which should (in combination with the new intuitive Section 2.1) answer your question. In simple words, for different paths (e.g., corresponding to different patients) $j$ in our training data-set, we can have different numbers of observations $n^{(j)}$.
>
> 10. Our results show that, given we find the minimiser of the loss function, the output $Y$ of the PD-NJ-ODE model converges to the optimal prediction, i.e., to the conditional expectation $\hat{X}$ (which is proven to be the unique optimizer of the loss function). In particular, the result shows that learning the optimal prediction can be done by finding the optimizer to our loss function (i.e., by finding the optimal weights $\theta$ for the neural network), which is not trivially clear. Without this result, one wouldn't know whether optimizing the loss function yields a model output that is helpful for our problem (i.e., close to the optimal prediction). Moreover, one would not even know that the model framework can in principle approximate the conditional expectation arbitrarily well (i.e., that the minimizer of (4) is also a minimizer of (3), since the minimum in (4) is taken over a much smaller set than in (3)). A less flexible model might not be able to do this. Furthermore, we show the corresponding convergence results for the minimizers of the empirical loss function (5).
> You are correct that we do not consider the problem of finding the minimizer of the loss function. The task of finding global minimizers of loss functions with neural network methods is a quite large field on its own, which is somewhat orthogonal to our research. Orthogonal in the sense that any result there can be used in combination with our results to show convergence with respect to the respective optimisation scheme. This was discussed in more detail in the first work Herrera et al. We added a citation to this discussion as well as a short explanation of the theorem as requested. In our experiments adam stochastic gradient descent works sufficiently well.

---

> ### Author Response · Authors · 2023-12-15
> **continuation2 of answer**
>
> 11. Even though the noise model is rather simple, we believe it is the one which is most relevant in practice, since measurement noise is often of this form. We also want to highlight again that noise which has a more complex structure, can also be dealt with by the stochastic filtering approach described in Section 6 of Krach et al. However, this requires additional assumptions, i.e., some type of knowledge of the underlying distribution and the noise distribution (or access to noisy *and* noise-free observations of $X$ during training).
>     In contrast to this, our extension for noisy observation doesn't need such knowledge, i.e., it is enough to observe the noisy data samples (without ever observing noise-free samples) to learn to optimally predict the noise-free process $X$.
>
>     We added Remark 3.7 which explains the case of a linear observation model $O = AX+\epsilon$, where the ultimate goal is to predict $X$ from observations of $O$.
>     Moreover, we added Remark 3.8, which discusses more general observation model.
>
> 12. We changed the numbering of the items in the assumptions as suggested. However, for a better visual detection which items changed or were added and which stayed the same, we do not repeat those items that didn't change in Assumptions 3.1 and 4.1.
>
> 13. Existence of $F_j$ is a consequence of the Doob-Dynkin lemma, which basically states that if a random variable $A$ is measurable with respect to the sigma-algebra $\sigma(B)$ generated by another random variable $B$, then there exists a measurable function $g$ such that $A = g(B)$. This is a very basic result in probability theory, which is often implicitly used without noticing.
>     Before Assumption 2.1, we added the intermediate steps that were used before applying the Doob-Dynkin lemma. In Section~3 this works identically, therefore, we think it is not needed to repeat the arguments.
>
>     The existence of $F_j$ can therefore be regarded as a basic fact, hence, this does not further interact with the assumptions (except that we know that the function for which we make these assumptions exists in general).
>     The assumptions we then make on $F_j$ are obviously much stronger, ensuring a certain regularity/structure of this function, which is then used by our model.
>
> 14. Since in some applications the described relaxation might be needed, we still think that this remark can be helpful just to point out that this is possible. However, the details of it are not very insightful, but rather technical; therefore, we do not think it would help to speak more about it.
>
> 15. As requested, we added the new Theorem 3.3, which is proven afterwards. Moreover, we added a short explanation what the main differences are in the proof, before starting it.
>
> 16. Short answer: When show that the prediction $Y$ of our model converges to the true conditional expectation $\hat{X}$ as the model size $m$ and the number of observed training paths $N$ tend to infinity. (Note again that $Y$ is the prediction of our model and not the observation process.) Please see the answer in point 10 for more details.
>
> 17. One practically relevant case where $\beta_i$ are known is given in Remark 3.6, where we show how to learn the conditional expectation of higher moments of $X$. This is an important example if one is interested not only in the prediction of the conditional mean, but rather in the conditional distribution.
>
> 18. Please see the answer for point 11 above.
>
> 19. As suggested, we added a discussion of the main technical challenges in the proof and some remarks about the helper lemmas directly after stating Theorem 4.3.
>
> 20. We added the new Theorem 4.3 as requested.
>
> 21. It is easy to see from equation (28) in the full proof in the appendix that for $\theta^\star$ satisfying $\Phi(\theta^\star) \leq \inf_{\theta\in\Theta}\Phi(\theta) + \epsilon = \Psi(\hat{X}) + \epsilon$, we have $d_k(\hat{X}, Y^{\theta^\star}) \leq C \, \sqrt{\epsilon}$ for every $k$ and for some constant $C > 0$. In particular, a loss close to the minimal loss value implies that $Y^{\theta^\star}$ is close to the true conditional expectation in terms of our pseudo-metrics. In the newly added Section 5 (and particularly in Section 5.2) we further discuss what can be inferred from closeness in the pseudo-metrics about the distance of the path produced by PD-NJ-ODE to the path of the true conditional expectation. Intuitively speaking, Proposition 5.4 tells us that the paths must be close at all $t$ which are in the support of one of the observation times $t_k$. Under further assumptions on the distribution of the $t_k$, we get more tractable ``closeness statements'', as shown in Examples 5.5 and 5.7.
>
>     Based on your question, we now additionally added Section 5.3 where this is discussed.

---

### Author Response · Authors · 2023-12-15
**Outline of additional changes (not requested by reviewers) since last version**

Due to internal discussions between the authors based on the reviews, we made some additional changes to the paper since first submitting it, which were not explicitly requested by the reviewers. Below we outline them.

- We did some reformulation in the beginning of Sections 3 and 4. Moreover, we fixed some typos and made some statements more precise.

- We added Section 2.1, which introduces the intuitive example of patient health data that is used in Sections 3 and 4.

- We added Remark 2.2, which clarifies that our results still hold if we allow for ``pseudo observation times''.

- We added Remark 2.3, which explains that Assumption 2.1 (i) is always satisfied and therefore not needed.

- In Section 4.3 after the proof of Theorem 4.3 we added Proposition 4.8 and Remark 4.9 which explain that the conditional independence of $X_{t_i-}$ and $n$  given $\mathcal{A_{t_i-}}$ can be dropped when changing the weighting in the loss function from an equal weighting of all observations to a weighting depending on the time difference since the last observation. This weighting gives more importance to observations after a long time of no observation. However, in turn, we need an additional assumption on the integrability of the time steps size.

- In Section 4.4.1 we replaced the usage of a dummy variable (that is always observed) by pseudo observation times (see item 3.), where no coordinate is observed. Even though it practically doesn't change anything, we believe it is a bit cleaner and easier to follow for readers not familiar with the topic.

- We added the new Section 5 where we discuss the practical implications of the convergence results that we proved.
    First, in Section 5.1, we study the practically relevant version of the conditional expectation that we would like to predict. We give an intuitive counterexample, where our model does not converge to it, and then give a sufficient condition (with proof) for this to hold.
    Secondly, in Section 5.2, we discuss the implications of convergence in the (pseudo) metrics $d_k$ (which ensures a good approximation at left-limits of observation times) for general times $t$. In particular, we study two practically relevant examples for the conditional distribution of the observation times and show that in these cases, with high probability, our model approximates the conditional expectation well on the entire support of the observation times.

- For the proof in Section 5.1, we added the new Proposition A.9 to Appendix A.

- In Remark 5.6 (also in the new Section 5.2), we provide convergence guarantees for the evaluation metric in one of the practically relevant examples. The evaluation metric was used in \citet{krach2022optimal} and in our paper for the experiments on synthetic data as a quality measure of the model approximation.

- We added more details about the usage of Dynkin's $\pi$-$\lambda$ theorem in the proof of Proposition A.1.

- We added Appendix B that discusses the inductive bias of our PD-NJ-ODE in detail for a better understanding how our model generalizes to out-of-sample data.

---

> ### Author Response · Authors · 2023-12-20
> **Small update**
>
> - We have merged the assumption of Section 5.1 into section 4 which improves clarity and avoids confuision.
>
> - (We also gave more detailed explainations in the proof of Theorem C.3 - part 2)

---

### Decision · Action_Editor_JvAe · 2024-01-29

**Recommendation:** Accept as is

**Comment:**

The paper has been reviewed by three reviewers: one recommends leaning accept (oEbU), one recommends accept (Zmm1) while the last one recommends leaning reject (g67F). The main criticism of g67F is that the paper contributions are fairly incremental compared to a previous paper by the same authors (Krach et al., 2022).
The proposed model is indeed an extension of the model presented in (Krach et al., 2022) to the setting where one has noisy observations and the masking can be dependent of the underlying model. While it is at the high level incremental, the authors have made in my opinion enough original contributions for the paper to be published in TMLR. The work is well motivated, results are thorough and clearly presented and illustrated. The revision has also improved the paper significantly.

I thus recommend acceptance of the manuscript as it is.

**Audience:**

Neural ODEs are popular models in machine learning so the proposed extension will find an audience.

**Claims And Evidence:**

The claims made in the submission are supported by accurate, convincing and clear evidence. There are both solid theoretical results as well as convincing experiments.

---

> ### Author Response · Authors · 2024-01-31
> **Thank you for the positive feedback**
>
> We would like to thank the Action Editor and all the Reviewers for their great effort throughout the reviewing process.
>
> The Authors